# CAPRI enables comparison of evolutionarily conserved RNA interacting regions

Amol Panhale [1], Florian M. Richter[1], Fidel Ramírez [1], Maria Shvedunova [1], Thomas Manke [1], Gerhard Mittler[1] & Asifa Akhtar [1]

RNA-protein complexes play essential regulatory roles at nearly all levels of gene expression. Using in vivo crosslinking and RNA capture, we report a comprehensive RNA-protein interactome in a metazoan at four levels of resolution: single amino acids, domains, proteins and multisubunit complexes. We devise CAPRI, a method to map RNA-binding domains (RBDs) by simultaneous identification of RNA interacting crosslinked peptides and peptides adjacent to such crosslinked sites. CAPRI identifies more than 3000 RNA proximal peptides in *Drosophila* and human proteins with more than 45% of them forming new interaction interfaces. The comparison of orthologous proteins enables the identification of evolutionary conserved RBDs in globular domains and intrinsically disordered regions (IDRs). By comparing the sequences of IDRs through evolution, we classify them based on the type of motif, accumulation of tandem repeats, conservation of amino acid composition and high sequence divergence.

[1] Max Planck Institute of Immunobiology and Epigenetics, 79108 Freiburg im Breisgau, Germany. Correspondence and requests for materials should be addressed to G.M. (email: mittler@ie-freiburg.mpg.de) or to A.A. (email: akhtar@ie-freiburg.mpg.de)

RNA–protein complexes are key components in the life cycle of RNA[1,2]. RNA–protein interactions can be studied on a high-throughput level from two angles[3]. The protein-centric angle relies on isolating proteins and sequencing the RNAs bound to them[4–7]. On the other hand, the RNA-centric approach detects proteins bound to RNA using mass spectrometry[8,9]. An example of an RNA-centric method is fixing RNA–protein interactions in vivo using UV irradiation and isolating polyA-containing (polyA+) RNAs using oligodT-coupled beads and highly stringent washes[10–12]. This approach was used to identify around 1800 mammalian direct RNA-binding proteins (RBPs), including hundreds of novel RBPs, many of which are involved in non-canonical pathways like energy metabolism and DNA repair[2,13–16]. However, proteins which interact with polyA+ RNA indirectly have not been probed on a systematic level until now.

Most of the new RBPs discovered by the RNA-centric RBP capture methods do not possess known RNA-binding domains (RBDs). Previous RBD mapping approaches have applied mass spectrometric techniques to identify peptides covalently bound to RNA[17]. The throughput of these techniques is limited due to the heterogeneity in peptide-ribonucleotide conjugates and difficulty of identifying the conjugated peptide tandem MS spectra by standard database search engines. Even with a specialised database search algorithm and homogeneous thio-uridine-mediated UV crosslinking, the polyA+ RNA interactome capture method could only identify 133 unique crosslinking sites in 57 S. cerevisiae proteins[18,19]. Alternative approaches rely on predicting crosslinked peptides by detecting peptides adjacent to the site of crosslink[20,21] or by comparing peptide intensities between crosslinked and non-crosslinked samples[22].

Here, we introduce CAPRI (Crosslinked and Adjacent Peptides-based RNA-binding domain Identification), a technique which enhances RBD discovery by simultaneously identifying both crosslinked peptides (in the immediate vicinity of the RNA–protein interface) and adjacent peptides (next to the crosslinked peptide) from the same sample. We apply CAPRI to Drosophila and human cells, uncovering 234 new Pfam RBDs in the two species. We identify 29 conserved RBDs with a globular structure. Furthermore, we uncover hundreds of IDRs mapped by 870 CAPRI-peptides. Based on manual screening, we find 40 pairs of IDRs whose RNA-binding function and location is conserved between Drosophila and human orthologs. We also produce exhaustive coverage of the Drosophila RNA-binding proteome (RBPome). Our study compares the RBPs captured by formaldehyde (FA) and ultraviolet (UV) crosslinking at a high-throughput level, and shows that FA crosslinking can retrieve not only direct RNA binders but also the secondary layer of RNA-interacting proteins. Our integration of RBD discovery by CAPRI with FA- and UV-mediated interactome capture provides the basis for a more complete understanding of the metazoan RNA–protein interaction network.

## Results

### The Drosophila RBPome at four levels of resolution.
Our study presents comprehensive coverage of the Drosophila RBPome at four levels of resolution (refer to Fig. 1 for the workflows used in the study). We first identified the RNA–protein complexes in the cell by combining formaldehyde crosslinking and polyA+ pull-down (Fig. 1a, (1)). Next, we scored direct RBPs using UV crosslinking (Fig. 1a, (2)). Following this, we identify RBDs through a combination of adjacent and crosslinked peptides using a methodology we developed: CAPRI. The CAPRI technique provided the final two levels of resolution of our study. Owing to

its use of UV crosslinking and a parallel workflow, the CAPRI pipeline mapped both the peptides in proximity to RNA (Fig. 1b, (3)) and the precise amino acids contacting RNA (Fig. 1b, (4)). In a separate analysis, we also applied FA crosslinking to RBD capture (Fig. 1c) and showed it to be a viable complementary method to the CAPRI workflow. We additionally applied CAPRI interactome capture to human cells in order to analyse the evolutionary conservation of newly identified RBDs between the two species. These interactome capture techniques and the datasets acquired using them are discussed in detail in the following sections.

### UV- and FA-mediated capture of direct and indirect RBPs.
Although Drosophila melanogaster constitutes an important metazoan model organism, RBPome description in Drosophila (~800 proteins) is limited compared to mammals (~1800 proteins)[2]. Previous attempts to characterise the fly RBPome have relied exclusively on UV crosslinking[14,23], which only retrieves proteins in direct contact with RNA nucleobases[11,12,18,24]. In contrast, FA can create inter-protein and RNA–protein cross-links, which are ideal for capturing both direct and indirect RNA–protein interactions (Supplementary Note 1)[8,25]. Therefore, we adapted the polyA+ interactome capture to use FA cross-linking in addition to UV crosslinking in Drosophila Schneider (S2) cells. We optimised the FA interactome capture such that comparable amounts of both proteins and RNA were captured by FA and UV crosslinking (Fig. 2a and Supplementary Fig. 1a, b, c). The specificity of interactome capture was validated (Supplementary Fig. 1b–h) by silver staining (Fig. 2a) and western blot against the known Drosophila RBPs Mle, Glorund, Squid and Rump, with histone H3 and actin serving as negative controls (Fig. 2b).

We performed the UV- and FA-based interactome captures in three biological replicates. We observed high consistency between the replicates as assessed by RNA profiles (Supplementary Fig. 1d, e), exclusion of genomic DNA (Supplementary Fig. 1d, f), recovery of mRNA (Supplementary Fig. 1g) and protein analysis (silver staining; Supplementary Fig. 1h). Next, the samples were subjected to GeLC-MS analysis[26]. We used label-free quantification and a moderated t-test to select proteins enriched in the crosslinked samples at a false discovery rate of 1% and further applied a filter for eightfold increase compared to the non-crosslinked control samples. This led to the determination of two Drosophila RNA-interacting protein datasets: the UV-crosslinked RNA-binding proteome (UV-RBPome) comprising 1512 proteins and the FA-crosslinked RNA-binding proteome (FA-RBPome) encompassing 2327 proteins (Fig. 2d, Supplementary Fig. 2a–f and Supplementary Data 1).

The Drosophila RBPomes obtained using FA and UV cross-linking showed significant overlap both with each other and with previously published studies. As anticipated, >85% of the proteins in the UV-RBPome were also represented in the FA-RBPome (Supplementary Fig. 2c, d). Furthermore, the protein intensities detected in both UV- and FA-RBPomes were found to be highly correlated (Pearson correlation > 0.8) (Fig. 2c). We successfully captured >70% of previously published UV-based Drosophila embryo RBPomes[14,23] (Fig. 2d). It is important to note that both previous studies were performed on early embryonic stages (0–2 h[14,23] and 4.5–5.5 h[14], whereas the S2 cells employed in our work were originally isolated from late embryonic stages 20–24 h[27]. We observed > 50% overlap between our interactomes and predicted fly orthologs of RBPs described in published RNA interactomes from several species[2] (Fig. 2d). Our UV- and FA-RBPomes together revealed 1021 candidate RNA-associated proteins (Fig. 2d).

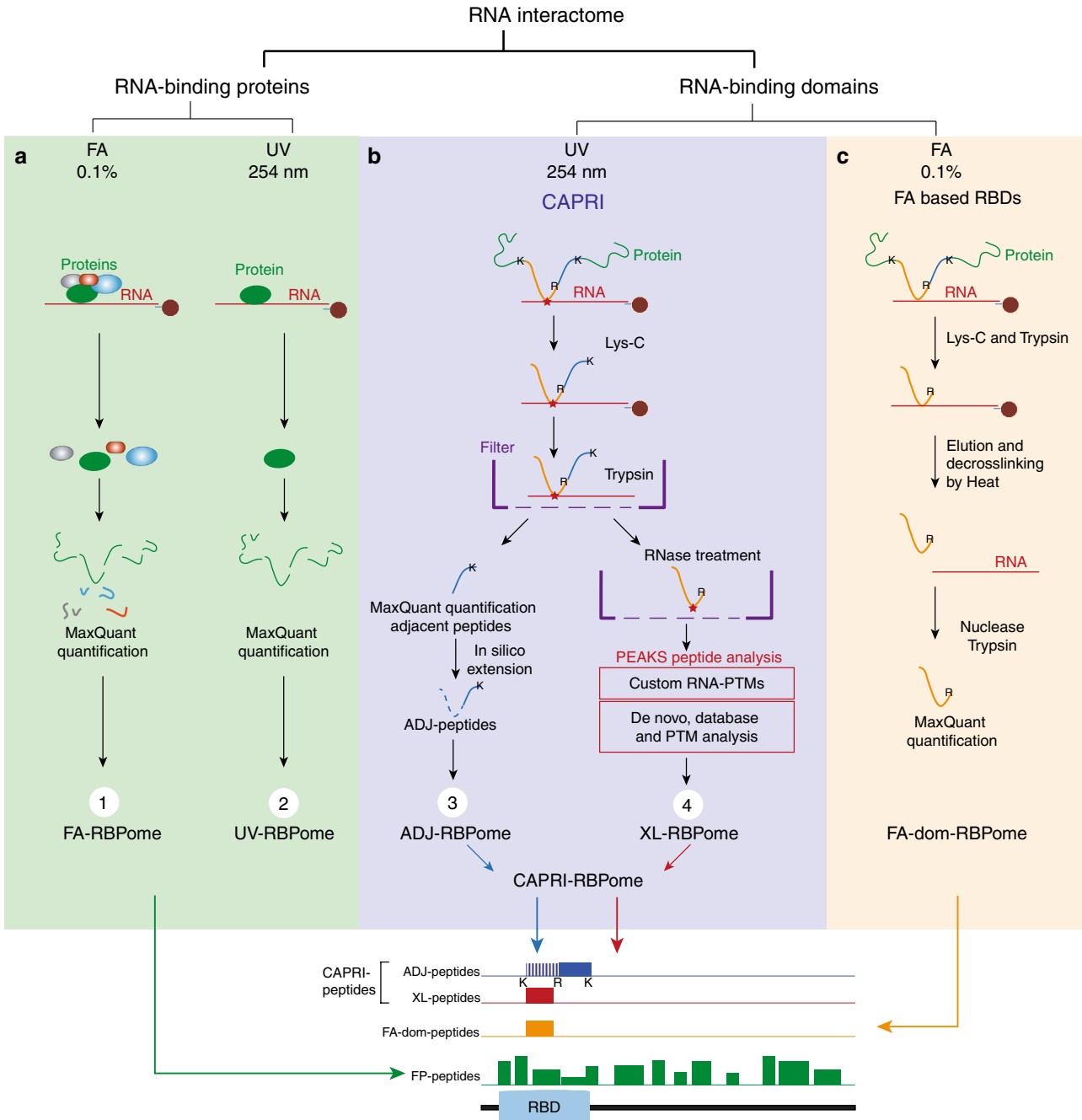

**Fig. 1** Characterisation of RNA–protein complexes, RBPs and RBDs. We characterised the Drosophila RBPome at four levels of resolution. **a** (green box) On the first two levels of resolution we focus on comprehensive identification of RBPs and RNA-associated protein complexes in *Drosophila* cells. First we used FA crosslinking to compile a dataset containing both direct and indirect RNA binders (FA-RBPome (1)). At the next level we used UV crosslinking to generate a dataset composed of direct RNA binders (UV-RBPome (2)). Both FA- and UV-RBPome datasets exhibit peptide coverage across the entire length of RBPs, which is illustrated by the green FP-peptides in the schematic in the lower panel. **b** (blue box) The next two levels of resolution focused on discovery of RNA–protein contact sites. We devised a new approach, termed CAPRI, which integrates two analysis pipelines to facilitate large-scale identification of RBDs. Briefly, UV-crosslinked RBPs are isolated using oligo-dT beads and digested with endoprotease Lys-C to release non-crosslinked peptides. The RNA-peptide moieties are eluted from the beads, transferred to a 30 kD filter and washed with 8 M urea to remove nonspecific background peptides. The crosslinked peptides are further digested with trypsin (cutting C-terminal to K/R) to release the adjacent peptides (coloured blue) as flowthrough. The RNA-peptide conjugates (orange-red) retained on the filter were released after degrading the RNA using a cocktail of nucleases. The MS/MS data from all crosslinked replicates are submitted to PEAKS for a combined search with custom defined monoisotopic masses of RNA adducts to identify XL-peptides. The non-heterconjugate peptides from all MS/MS data are analysed by MaxQuant. The statistically enriched peptides in the UV irradiated samples (adjacent-peptides) are reconstructed in silico to the original endoproteinase Lys-C-digested full peptide sequence (ADJ-peptides: blue). The ADJ-RBPome (3) and XL-RBPome (4) together give the CAPRI-RBPome. **c** (orange box) The FA-dom-RBPome provides complementary data on RBDs using FA as a crosslinker. Briefly, 0.1% formaldehyde-crosslinked RBPs are captured using oligo-dT beads and are further digested using Lys-C and trypsin. Following elution the RNA-peptide crosslinks are reversed by heating and later the nucleic acids are degraded by nucleases. The peptides are cleaned-up using SP3 purification and analysed using LC/MS

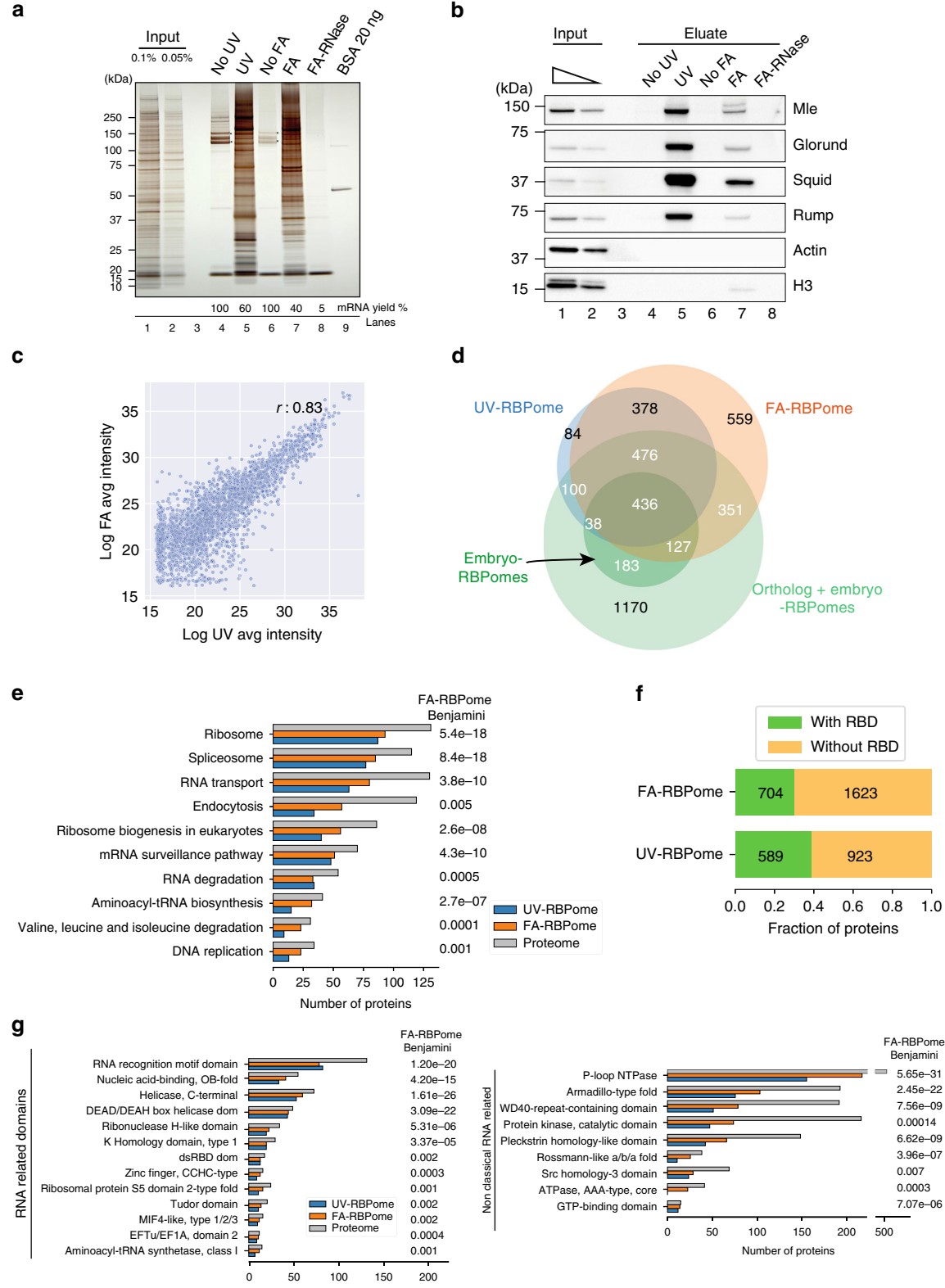

Despite their overall similarity, the FA-RBPome revealed interesting differences compared to the UV-RBPome, particularly with respect to the classes of proteins represented. Gene ontology (GO) enrichment analysis of molecular functions and biological processes uncovered that the most enriched terms were related to RNA functions for both RBPomes (Supplementary Fig. 3a, b).

Similarly, KEGG pathway enrichment analysis revealed pathways like mRNA transport, RNA surveillance and ribosome biogenesis in both UV- and FA-RBPomes (Fig. 2e and Supplementary Fig. 3c, d, e). However, the overall coverage of RNA-related proteins was higher in the FA-RBPome compared to UV-RBPome. Additionally, novel pathways like endocytosis, valine,

**Fig. 2** RNA interactome capture in *Drosophila melanogaster*. **a** Protein profiles from interactome capture visualised on a polyacrylamide gel by silver staining. Sample loading: Input (lanes 1–2); Interactomes captured from UV control (noUV), UV crosslinked (UV), formaldehyde control (noFA), formaldehyde-crosslinked (FA), and formaldehyde-crosslinked with RNase treatment (FA-RNase) (lanes 4–8); and finally BSA (20 ng) (lane 9) were utilised to validate the interactome capture protocol. **b** Validation of the interactome capture by western blot of selected proteins. Antibodies against known *Drosophila* RBPs (Mle, Glorund, Squid, Rump) were used as positive controls, whereas antibodies against H3 and actin served as negative controls. Source data are provided Supplementary Fig. 19. **c** Scatter plot of average intensity values of proteins detected in both the UV-RBPome and FA-RBPome. Pearson correlation (*r*) of 0.83 was observed between the two sets. **d** Venn diagram of the UV-RBPome (blue), FA-RBPome (orange), previously reported embryo interactomes[14,23] (green) and previously reported UV-RBPomes in all species, including the *Drosophila* embryo RBPomes (light green)[2,14,23]. **e** KEGG pathways enriched in the FA-RBPome (Benjamini–Hochberg correction 1% FDR). Numbers of proteins from the full proteome, FA-RBPome and UV-RBPome present in each of the pathways are presented. **f** Fraction of proteins with known RNA-binding domains (RBDs) in the UV- and FA-RBPomes. **g** Enriched InterPro domains in the FA-RBPome (Benjamini–Hochberg correction 1% FDR). Number of proteins with enriched InterPro domains along with proteome count (Benjamini–Hochberg correction 1% FDR). Representative RBDs (left panel) and domains related to RNA function (right panel) are shown

---

leucine, isoleucine degradation and DNA replication were more enriched in the FA-RBPome (Fig. 2e and Supplementary Fig. 4a, b). Although some of the metabolic enzymes are also present in the UV-RBPome, as seen previously[28], their number is significantly higher in the FA-RBPome (Supplementary Fig. 4c, d). Thus, we predict that interaction of metabolic proteins with the mRNA machinery occurs either via indirect binding to bona fide RBPs or through direct interactions that cannot be recovered by UV crosslinking.

We next asked whether the RBPs we identified carried known RBDs. As expected, classical RBDs like RRM, KH and Helicase domains were enriched in both UV- and FA-RBPomes (Fig. 2g and Supplementary Fig. 4e). We also found high enrichment for InterPro domains such as P-loop containing nucleoside triphosphate hydrolase, WD40 and Armadillo. However, >60% of the identified *Drosophila* proteins in both UV- and FA-RBPomes did not harbour any of the classical RBDs (see Methods and Fig. 2f). Thus, it seems likely that these proteins possess new classes of RBDs that have not yet been catalogued[29,30].

**CAPRI as a workflow for comprehensive identification of RBDs**. We were interested in discovering RBDs in the hundreds of RBPs identified in our RBPomes. The current standard methodology employed for RBD mapping is detection of UV-crosslinked RNA-peptide heteroconjugates by mass spectrometry[18]. Although this analysis delivers single amino acid resolution, the high sample heterogeneity observed with UV-crosslinked peptides severely hampers its coverage and throughput (Supplementary Note 2). We overcame these limitations by isolating peptides adjacent (N- or C-terminal) to the peptides carrying the crosslinked amino acid. These adjacent peptides have better abundance and homogeneity compared to crosslinked peptides (Supplementary Fig. 5a). Furthermore, they can then be analysed using standard proteomics pipelines. The RBDmap method used the same principle to successfully identify adjacent peptides in humans[20]. However, no methodology until now has combined analysis of both crosslinked and adjacent peptides from the same samples. Keeping this in mind, we developed a new workflow: (simultaneous) Crosslinked and Adjacent Peptide-based RNA-binding domain Identification (CAPRI).

The first step of the CAPRI workflow involves isolation of bona fide RBPs from UV-irradiated cells on beads (Fig. 1b). This is followed by the digestion of bound proteins using Lys-C (cleaves C-terminal to K) to release any peptides not covalently attached to RNA (Fig. 3a and Supplementary Fig. 5b–d). After several washes, the RNA-peptide conjugates were eluted. We devised a new strategy (RNA-FASP) for enrichment of crosslinked peptides under stringent conditions (8 M urea) based on the retention of RNA on 30 kD MWCO centrifugal filters (FASP filters)[31] (Supplementary Fig. 5e, f). The RNA-peptide conjugates retained

on the filter were further digested with Lys-C to ensure complete digestion. The released peptides were collected as a flowthrough to estimate the proportion of Lys-C missed cleavage sites (Supplementary Fig. 5h). The peptides remaining on the filter were subsequently digested with trypsin (cleaves C-terminal to K and R) to release the peptides adjacent to the crosslink. The crosslinked RNA-peptide heteroconjugates were then collected after digesting the RNA completely (Fig. 1b and Supplementary Fig. 5g).

The CAPRI analysis approach is bipartite. The first branch detects peptides adjacent to crosslink sites and is thereby capable of detecting a higher number of RBDs (Fig. 1b, ADJ-RBPome). The second branch delivers an extremely high-confidence but smaller set of direct crosslinked peptides (Fig. 1b, XL-RBPome). The peptides were analysed by two different software pipelines, one for adjacent peptides and another for crosslinked peptides (described in the following sections), and then combined together to identify RBDs at two levels of resolution: peptide and amino acid level.

**CAPRI adjacent peptide analysis**. In the first branch of the CAPRI workflow, the adjacent peptides were analysed by conventional label-free quantification using the MaxQuant software[32,33]. A total of 2388 adjacent peptides were obtained after stringent statistical evaluation (Fig. 3b and Supplementary Data 2). The selected tryptic adjacent peptides were extended in silico to the nearest Lys-C digestion sites to encompass the neighbouring crosslink site (Fig. 1b, See Supplementary Note 3). Multiple adjacent peptides arising from the same Lys-C-cleaved peptide were then merged together to give a final dataset of 1510 extended RNA-adjacent peptides (termed ADJ-peptides) mapping to 574 proteins (termed ADJ-RBPome) in *Drosophila* (Supplementary Data 2).

At the outset, the ADJ-RBPome was highly enriched in RNA-associated GO molecular functions and biological processes (Fig. 3c and Supplementary Fig. 6b). The ADJ-peptides were validated as mapping to RBDs by checking for the enrichment of classical RBDs relative to background peptides identified from full proteins (FP-peptides; Fig. 3d, e). Using the fly protein Larp as an example, we demonstrate the identification of RNA-interacting regions based on ADJ-peptides (Supplementary Fig. 6a). We additionally performed CAPRI ADJ-peptide analysis on human HEK293 cells (Fig. 3f and Supplementary Fig. 6c–e). We identified a total of 1,609 RNA ADJ-peptides in 543 human proteins (Supplementary Data 3). Similar to our findings in *Drosophila*, the human proteins were highly enriched for RNA-related functions and RBDs (Supplementary Fig. 6f–h). Thus, by using the ADJ-peptide analysis branch of the CAPRI workflow, we recapitulated canonical RBDs in both *Drosophila* and human cells.

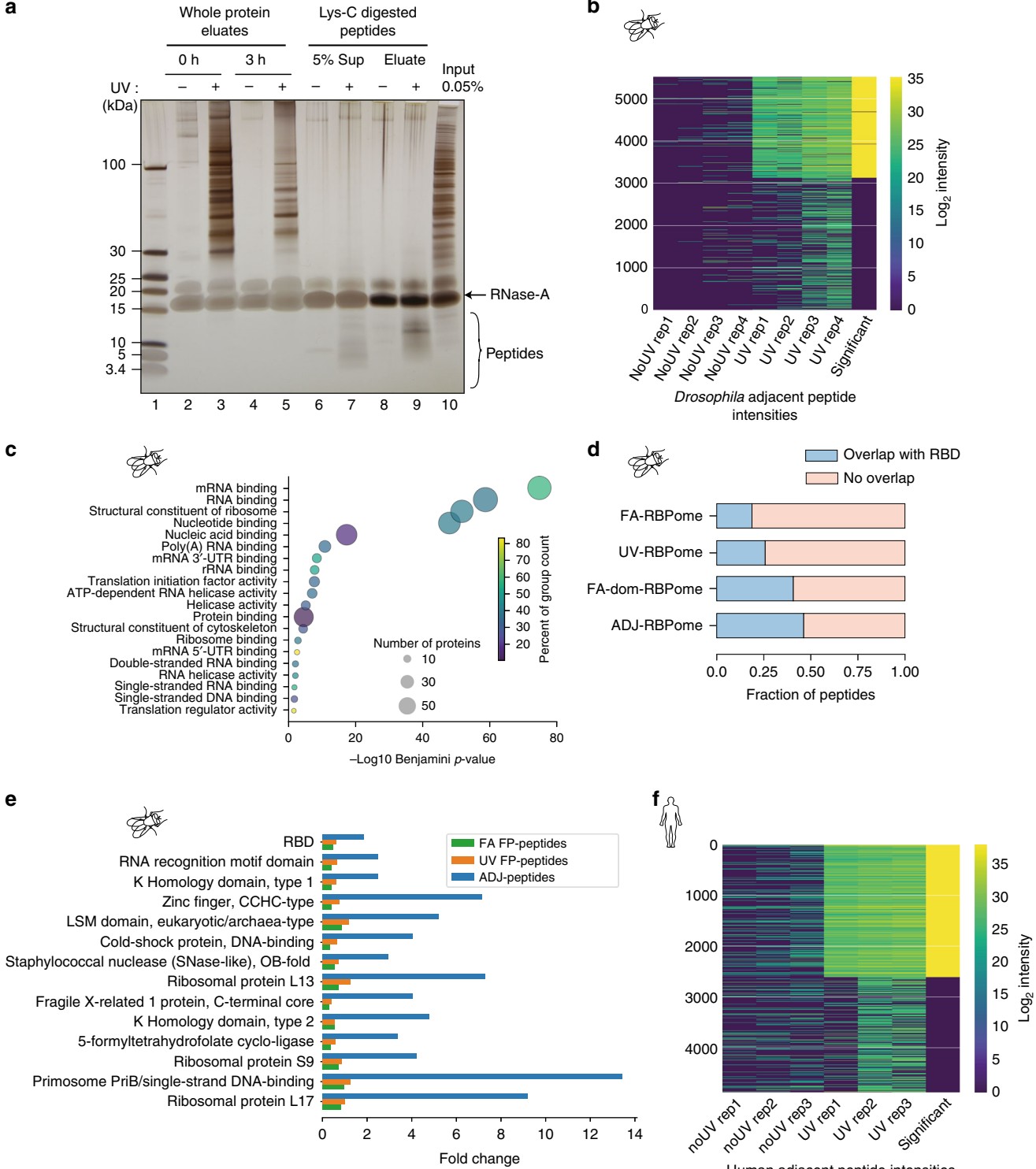

**Fig. 3** Identification of RBDs by CAPRI. **a** On-bead digestion of proteins with Lys-C was verified after RNase-A digestion by silver staining. The lanes are as follows: whole-protein eluates at 0 h and 3 h incubation (lanes 2–5); the released peptides following digestion with Lys-C for 3 h (5% of supernatant) and peptides eluted from beads (Elu) for non-crosslinked ( - ) and UV crosslinked ( + ) samples (lanes 6 to 9). **b** Clustered heatmap of raw intensities (log₂) of adjacent peptides detected in the four biological replicates of non-crosslinked and UV-crosslinked samples in *Drosophila*. Adjacent peptides in UV crosslinked samples were selected by applying a moderated *t*-test with a Benjamini–Hochberg correction (FDR 5%) and additional filtering for an eightfold intensity increase relative to the non-crosslinked samples. Adjacent peptides selected after statistical analysis are depicted in the last lane with maximum intensity. **c** GO terms (Molecular function) enriched in the *Drosophila* ADJ-RBPome. **d** The fraction of ADJ-peptides mapping to canonical RBDs is plotted along with the fraction of full-protein peptides (FP-peptides) mapping to the same domains from the UV- and FA-RBPomes. **e** Domains enriched in ADJ-peptides compared to FP-peptides from UV- and FA-RBPomes. **f** Same as **b** but for three biological replicates in humans

**CAPRI crosslinked peptide analysis**. In the second branch of CAPRI, we analysed crosslinked peptides using a newly developed pipeline. The CAPRI protocol permits enrichment of crosslinked peptides over non-crosslinked peptides from the same sample (Fig. 1b). This makes it possible for crosslinked peptides, present in lower stoichiometric abundance, to be detected in the data-dependent acquisition mode of MS. As the masses of conjugated RNA-peptide molecules are observed as the sum of their individual components[34], the RNA adducts can be treated as post translational modifications (PTMs) by the MS analysis software (Supplementary Note 4). Since the number of PTMs which need to be defined for this purpose are in the hundreds, only specialised database search-based strategies have been used in analysing crosslinked peptides in a high-throughput manner thus far[18,19,35]. We designed a workflow that not only detects crosslinked peptides by de novo search but also reduces the expert spectrum annotation work to a minimum by analysing all MS/MS spectra a priori. The de novo sequencing of the peptide moiety was performed by the PEAKS algorithm[36] followed by database identification in the PEAKS software[37] (see Methods for details).

Briefly, we designed custom RNA-PTMs taking into consideration the fragmentation pathways observed in previously published data[34]. The HCD/CID fragmentation of crosslinked peptides in MS/MS preferentially cleaves the glycosidic and phosphodiester bonds but not the covalent link between the base and the amino acid side chain. Thus, we treated the four standard ribonucleotides (A, G, C, U) as if each was composed of two PTMs: the ribonucleobase (A', G', C', U') and the rest of the ribonucleotide (ribose and phosphate) (Supplementary Data 4 and Supplementary Fig. 7a, b). The position of the base PTM on the peptide enabled identification of the crosslinked amino acid. Our design made it possible to cover all RNA sequence combinations of up to 3 nucleotides in only 34 RNA-PTMs. These reduced numbers of RNA-PTMs could be used in the de novo step (peptide sequence tag generation) of spectra identification (SS3 workflow in Supplementary Fig. 7b). The crosslinked spectra containing non-canonical combinations of nucleotide fragments were removed in silico (Supplementary Fig. 7c). Finally, the spectra were manually curated to produce a final set of crosslinked peptides, which we termed XL-peptides (Fig. 4a). We also manually annotated a few example spectra with marker ions, neutral losses and RNA-peptide conjugate (adduct) ions (Supplementary Data 5 and 6).

XL-peptide analysis uncovered hundreds of high-confidence RBDs in both *Drosophila* and humans (Supplementary Data 6–9). We identified a total of 829 unique XL-peptides mapping to 262 unique peptide sequences in 115 proteins in *Drosophila* (*Drosophila* XL-RBPome) (Fig. 4b). One-hundred seventy-seven (67.5%) of these *Drosophila* XL-peptide sites mapped to canonical RBDs (Fig. 4b). The remaining 85 peptides represent new sites of RNA interaction. Similarly, a total of 961 unique XL-peptides were identified in 280 unique peptide sequences mapping to 135 proteins in the human XL-RBPome. Two-hundred sixteen (77.14%) of the human XL-peptide sites overlapped with canonical RBDs. The remaining 64 XL-peptides represented new sites of RNA interaction (Fig. 4b).

We validated the XL-peptides by visualising their positions relative to RNA in seven available RNA-protein crystal structures (Structures and annotated spectra in Supplementary Data 6, Fig. 4c and Supplementary Fig. 7d-f). All mapped XL-peptides were located close to RNA in 3D space. Of the 17 amino acid crosslink sites that we mapped in these structures (Fig. 4c and Supplementary Data 6), 15 were in close proximity to RNA. We show here an annotated spectrum of a peptide from HNRNPC (Fig. 4c) crosslinked to AU RNA with a covalent bond between valine/phenylalanine and uridine (Fig. 4c inset). In another

example, an XL-peptide maps to the helix-turn-helix DNA binding domain (a non-classical RBD) in PA2G4 (Fig. 4d). The peptide is very close to a positively charged surface of the protein (Fig. 4d inset). Thus, XL-peptides identify both classical and novel RNA-interacting surfaces.

In-depth analysis of all XL-peptides revealed a clear preference for particular amino acids and nucleotides at the sites of crosslinks. We could identify the site of crosslink at single amino acid resolution in half of the XL-peptides. These amino acids had higher fractions of aromatic, hydrophobic and aliphatic residues compared to amino acids observed in close proximity to RNA in previously published RNA protein structures[38]. The F, Y and C residues are especially enriched at the sites of crosslinks in our data (Fig. 4e). Similarly, the nucleotide composition of the RNA sequences crosslinked to the peptides showed a clear bias for the uridine base (Fig. 4f), consistent with previous reports[18]. We also observed a new class of crosslinked peptides where only the base was present in the crosslinked RNA-peptide heteroconjugate (Supplementary Fig. 7g, h and Supplementary Data 5 Spectra 14,15). Such a crosslink could only be detected if the glycosidic bond between the nucleobase and ribose sugar is cleaved either during sample preparation or subsequent to ionisation as an MS in-source decay product.

**XL-peptides overlap with identified ADJ-peptides**. As we have simultaneously isolated ADJ- and XL-peptides from the same samples we expected a high degree of complementarity between XL-peptides with ADJ-peptides. We found that >60% of the XL-peptides overlapped with ADJ-peptides (Fig. 5b). Furthermore, when extended by the mid-range of tryptic peptide length (30 amino acids) on either side, >90% of XL-peptides overlapped with ADJ-peptides. XL-peptides and ADJ-peptides are therefore positionally complementary. In one example, the ADJ-peptides and XL-peptides mapping to one of the RRM domains in *Drosophila* pUf68 and its human ortholog PUF60 show clear complementarity (Fig. 5a). This strongly supports the reliability of using ADJ-peptides for RBD identification and gives us increasing confidence in those sites, which exhibit only ADJ-peptides (with no concomitant XL-peptide). We combined XL-peptides and ADJ-peptides into a single dataset, which we termed CAPRI-peptides.

**Classes of identified RNA interacting regions**. Using CAPRI we identified classical RBDs like RRM and KH-domains along with non-classical RBDs like R3H, Pumilio repeat, CSD, C2H2, FYVE-related and CCCH domains (Fig. 5a and Supplementary Fig. 9a–e). CAPRI analyses also discovered hundreds of previously uncharacterised RBDs in *Drosophila* and human cells (Supplementary Data 10). We identified and classified novel domains by clustering information on the overlap of the CAPRI-peptides with Pfam domains[39] and IDRs (Fig. 5c). We considered all peptides not overlapping with RNA-binding Pfam domains (see Methods) to be mapping to new RBDs and sub-categorised them into novel RBD Pfam domains, IDRs and unannotated structured domains (Fig. 5d). The XL-peptides were also seen to map across all domain subclasses (Fig. 5c). Some examples of high-confidence XL-peptides with annotated spectra mapping to new RBDs are shown in Supplementary Data 7.

Compared to other workflows, CAPRI supports maximum stringency for removal of non-specific RNA–peptide interactions and allows simultaneous identification of crosslinked and adjacent peptides. A comparison of CAPRI and existing methods[20,22,40] is outlined in Supplementary Fig. 8a. We found that the fraction of CAPRI-peptides mapping to known RBDs are similar to those observed in other recent publications

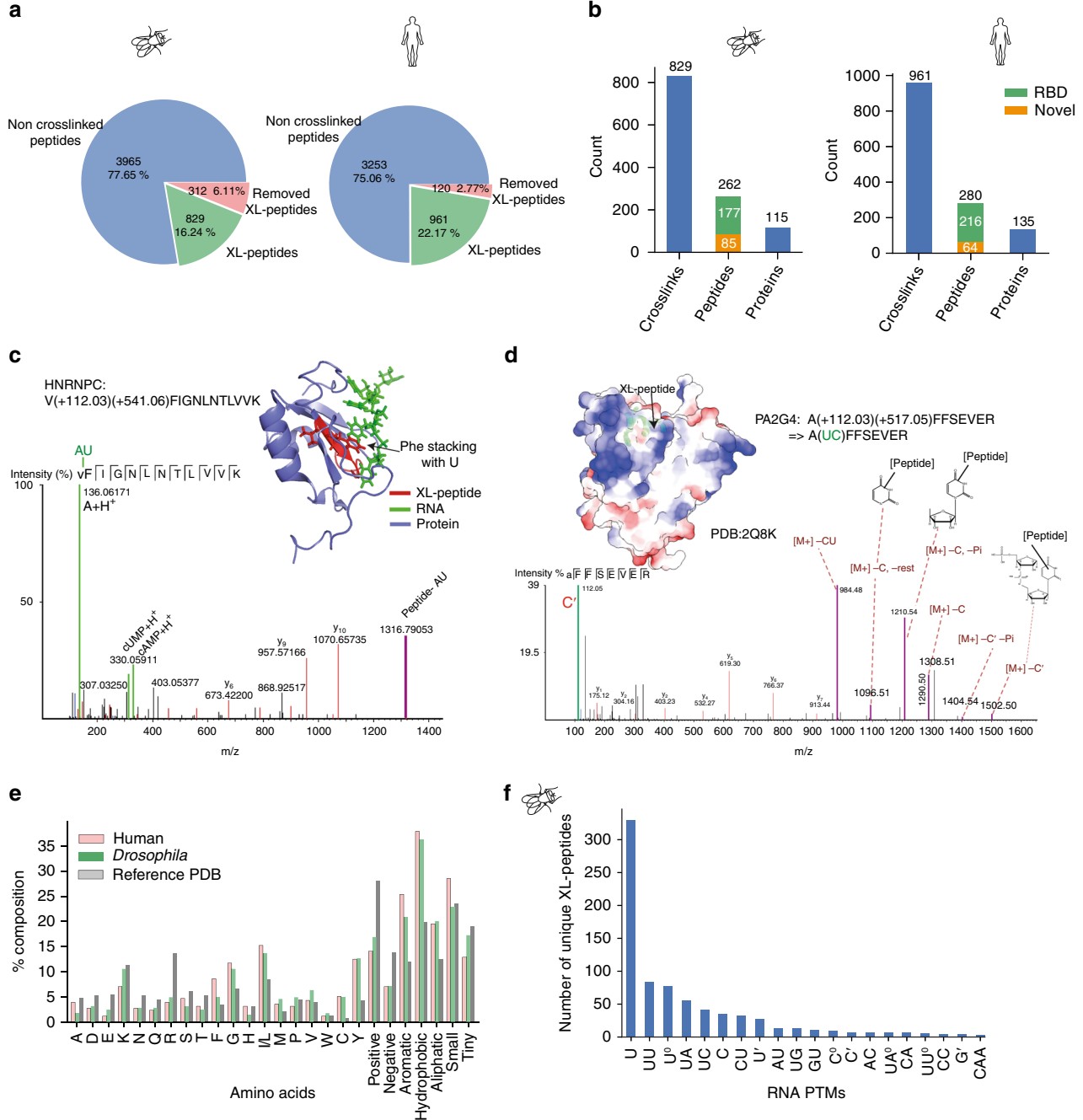

**Fig. 4** Identification of UV crosslinked XL-peptides using PEAKS PTM analysis. **a** Peptides detected in *Drosophila* (left) and human (right) PEAKS analysis: canonical non-crosslinked peptides (blue), filtered crosslinked peptides (red) and true XL-peptides (green). **b** Summary of all XL-peptides detected in *Drosophila* (left) and human (right). The crosslinks bar depicts the number of distinct RNA-peptide heteroconjugates. The peptides bar depicts the number of distinct peptide sequences. The proteins bar depicts the number of distinct proteins that are defined by the aforementioned peptides. **c** An example of an annotated XL-peptide spectrum from HNRNPC with a dinucleotide (AU) covalently attached to peptide VFIGNLNTLVVK. The y ions series (blue) and b ion series (red) were automatically annotated by PEAKS. The manually added RNA marker ions (A+, cUMP+, cAMP+) and RNA-peptide conjugates (peptide-AU) are shown in green and in magenta, respectively. Inset: The XL-peptide mapped to amino acids 18–28 in HNRNPC structure (PDB ID 2MXY). The protein is shown in blue, the RNA in green and the XL-peptide in red. Amino acids Phe and Val are shown in ball and stick models. The stacking interaction between Phe and proximal uridine base is pointed out (arrowhead). **d** Annotated spectrum of an XL-peptide peptide from PA2G4. The *m/z* ratios identified by PEAKS search are shown in red. The marker ion (C′) is shown in green. The peaks containing heteroconjugates with neutral losses are shown in purple. The nucleotide structures are shown for the purpose of visualising adducts present on the peptides and they may not represent the actual structure of the heteroconjugate. Inset: Position of the XL-peptide (green) in the crystal structure of the protein (PDB ID 2Q8K). The coulombic electrostatic potential of the protein surface is shown in blue for positive and red for negatively charged surfaces. **e** Composition analysis of amino acids identified as sites of crosslinks in *Drosophila* (green) and human (pink) compared to the composition of amino acids involved in direct interaction with RNA in known 3D structures (grey)[38]. **f** Bar chart displaying the distribution of ribonucleotide adducts observed in *Drosophila* XL-peptides

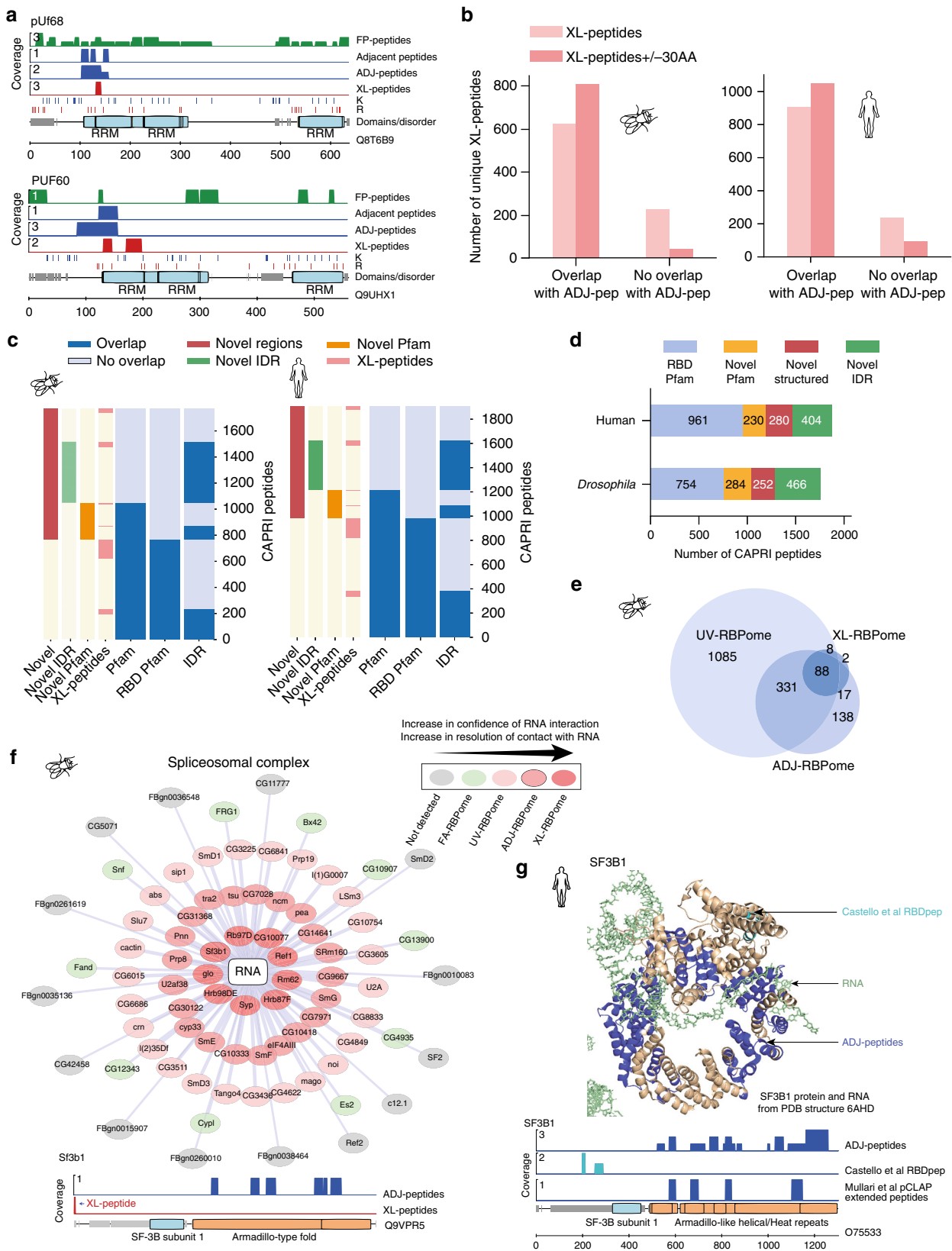

performed in mouse and human cells (Supplementary Fig. 8b). Additionally, a large proportion of the Pfam domains discovered by CAPRI-peptides in human cells were also identified in other mammalian RBD captures (Supplementary Fig. 8c).

**FA-based RBD identification**. FA has previously been used to identify RNA-interacting surfaces of single proteins[41]. However, its suitability for mapping RBDs at a high-throughput scale has not been investigated until now (Supplementary Note 5). We

**Fig. 5** Classification of CAPRI-peptides. **a** Visualisation of adjacent, ADJ- and XL-peptides detected in fly pUf68 and its human ortholog PUF60. The visualisation tracks from top to bottom for each protein: full-protein peptide (FP-peptide) (green), significant adjacent peptides (blue), extended adjacent peptides: ADJ-peptides (blue), XL-peptides (red), K and R amino acid tracks, disordered regions extracted from MobiDb (grey) along with labelled globular domains (blue boxes). **b** Summary of all XL-peptides overlapping with ADJ-peptides with (red) or without extending (light red) the XL-peptides by 30 amino acids in silico. **c** Clustered heat maps representing the distribution of XL- and ADJ-peptides identified in *Drosophila* (left) and humans (right) overlapping with Pfam, RBD Pfam and IDR. Left of each panel: XL-peptides and different classes of RBDs. **d** Distribution of CAPRI-peptides in RBD Pfam, novel Pfam, and novel structured domains as well as novel IDRs. **e** Overlap of *Drosophila* proteins identified in the UV-, ADJ- and XL-RBPomes. **f** Display of layered interactome representation for spliceosomal complex members after aggregating information from full-protein and domain interactome captures in *Drosophila*. The proteins are arranged in concentric circles where the confidence in and resolution of the site of RNA interaction increases with proximity to the RNA node (white) shown in the centre. The layers going from outside to inside: not-detected (grey), FA-RBPome (green), UV-RBPome (light red), ADJ-RBPome (red) and XL-RBPome (dark red). As an example of a novel surface interacting with RNA, Sf3b1 protein domain peptide coverage is shown below (N-term XL-peptide). **g** Identification of RNA-binding regions in human SF3B1. Top: The ADJ-peptides (blue) and Castello et al.[21] RBDpep peptides (cyan) mapped onto available sequence in the human SF3B1 protein structure (pale orange) (PDB ID 6AHD). Additional proteins in the structure are removed for better visualisation. The RNA is shown in pale green in stick model. Bottom: Peptide coverages of the ortholog human SF3B1. The visualisation tracks are as in **a**, except for the addition of tracks for Castello et al.[21] RBDpep extended peptides (cyan) and Mullari et al.[40] pCLAP extended peptides (blue)

performed on-bead proteolysis of oligo-dT-immobilised FA-crosslinked RBPs with Lys-C and trypsin (Fig. 1c). The RNA was eluted and RNA-peptide bonds were de-crosslinked by heat, after which the RNA was degraded. The peptides were further digested with trypsin and analysed by mass spectrometry (Supplementary Fig. 10b, see Methods for details). From three biological replicates a total of 554 peptides (FA-dom-peptides) mapping to 162 proteins (FA-dom-RBPome) were identified in *Drosophila* (see Methods and Supplementary Data 11, Supplementary Fig. 10c). These proteins were also enriched for classical RBPs (Supplementary Fig. 10d) and overlapped significantly with other *Drosophila* RBPomes (Supplementary Fig. 10e, f). Two-hundred twenty-five (40.6%) of the 554 FA-dom-peptides mapped to classical RBDs (Fig. 3d). The majority (57.4%) of FA-dom-peptides overlapped with ADJ-peptides and about a third (36.9%) of the XL-peptides overlapped with the FA-dom-peptides. Some examples of these overlaps can be seen in the peptide coverages shown in Supplementary Figs 6a and 9c–e. As a proof of principle, these overlaps clearly suggest that FA-dom-peptides can also be used to map RBDs. However, as the number of peptides uncovered by CAPRI was higher than in the FA-dom-peptides, we opted to focus on CAPRI-peptides for all subsequent analyses.

**Organisation of protein complexes based on proximity to RNA**. We found significant overlaps between the whole protein and CAPRI interactomes (Fig. 5e and Supplementary Fig. 11a). Our four datasets: FA-, UV-, ADJ- and XL-RBPomes in *Drosophila* define sets of proteins with increasing confidence for direct RNA interaction and with increasing precision in identifying the site of interaction. We hypothesised that this understanding could be used to produce a new way of visualising RNA-binding protein complex hierarchies based on the putative proximity of each protein component to RNA (see Methods). We implemented a visualisation based on concentric circles to show the predicted proximities of individual components of the 79-subunit spliceosomal complex to RNA (colour-coded in Fig. 5f). Spliceosome component Sf3b1 was placed in the innermost circle in our visualisation as it was recognised by a CAPRI XL-peptide in its N-terminus (Fig. 5f, darkest red and Supplementary Data 7 spectrum 26). Sf3b1 is known to be responsible for recognition of the splice site branch-point during RNA splicing[42–44]. In addition, several ADJ-peptides mapped to the Armadillo HEAT repeat region of both fly Sf3b1 (Fig. 5f, lower panel) and its human ortholog SF3B1 (Fig. 5g). The same region has previously been shown to be close to RNA by cryo-EM[42,43] (Fig. 5g).

We also applied this principle to visualise hierarchies in other complexes like the CCR4-NOT complex, which is involved in the

deadenylation of mRNA (Supplementary Fig. 11b). The HEAT repeat-containing and Tristetraprolin binding domains within the Not1 protein bind RNA (Supplementary Fig. 11c). In humans, the CNOT1 protein forms a scaffold on which the rest of the complex members assemble[45] and in yeast the Not1 C-terminal domain binds poly(U) RNA in vitro together with Not2 and Not5 (Not3 in *Drosophila*)[46]. We mapped the RNA-binding region of the fly Rga/NOT2 protein to its N-terminus, thus placing it in the innermost circle of our visualisation (Supplementary Fig. 11c). A similar analysis of large complexes like cytoplasmic ribosome and mitochondrial 55S ribosome along with smaller complexes such as TREX complex also revealed organisation of the proteins involved in RNA interaction (Supplementary Fig. 11d–f). Extending similar analysis to complexes or pathways without known RNA-related functions will permit us to gain insights into which of their components directly interact with RNA. For example, we identify RNA-associated proteins involved in oxidative phosphorylation (Supplementary Fig. 12a). We could also map the RNA interaction sites in fly ATPsyngamma, which is an extramembranous subunit (F1) of mitochondrial membrane ATP synthase (Supplementary Fig. 12b and Supplementary Data 7 Spectrum 1). We also identified the RNA bound peptides of fly mt:ND1 and COX1 proteins (Supplementary Fig. 12c, d). Thus, the combined analysis of CAPRI interactomes and RNA–protein interactomes provides insight into RNA-proximity-based protein complex hierarchy.

**Validation of newly identified RBPs**. We validated a selection of proteins representing diverse functional classes using two separate techniques in human cells. First, we independently developed a technique, which captures all large RNA ( > 200 nucleotides) in the cell by adapting commercially available silica solid phase extraction (similar to the recently published 2C technique[47]). Briefly, UV-crosslinked cells were resuspended in highly denaturing guanidine thiocyanate-containing buffer (Supplementary Fig. 13a, see Methods for details). After passing the solution through a genomic DNA elimination column, the RBPs were isolated on an RNA-binding column. Following stringent washes the RBPs were eluted from the column. These were validated by silver staining (Supplementary Fig. 13b, c) and western blots against positive controls (classical RBPs: ADAR, DHX9, HNRNPM1, HUR3A2, POLRMT; non-classical RBP: GAPDH) and histone H3 as a negative control (Supplementary Fig. 13d). We observed that several proteins involved in oxidative phosphorylation (OXPHOS Complex V member ATP51A and Complex II member SDHA), DNA repair (MSH6 and XRCC5)

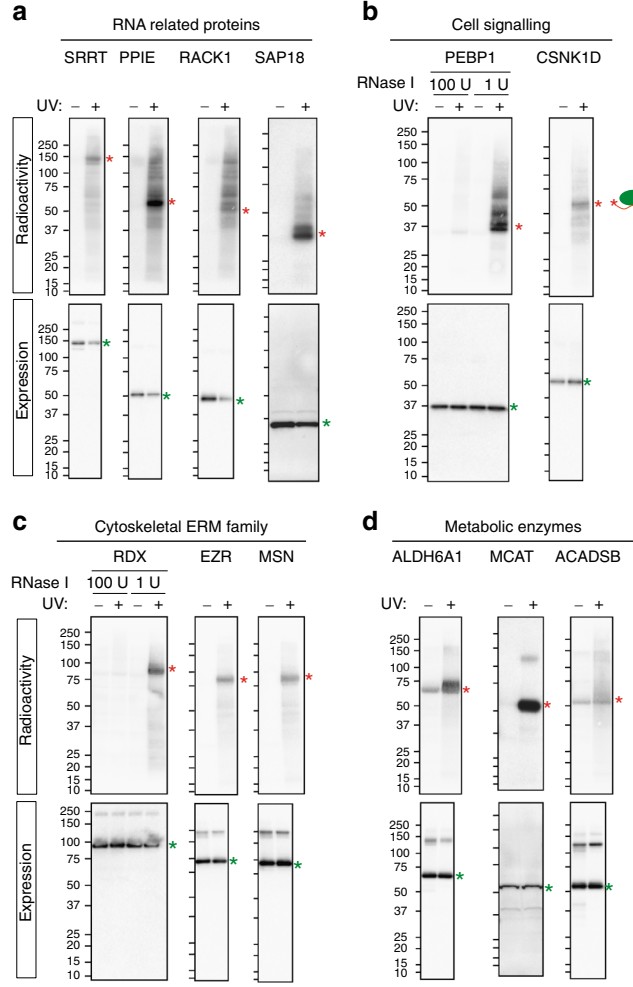

**Fig. 6** Validation of RBPs using PNK assay. Briefly, 3Flag-HBH-tagged proteins are stably expressed in Flp-In T-Rex 293 cells. The proteins are affinity purified from UV crosslinked and non-crosslinked cells under denaturing conditions. RNA bound to proteins is labelled with radioactive $^{32}$P using the PNK enzyme. The RNA–protein heteroconjugates are separated by size (SDS-PAGE) and transferred to a membrane on which the RNA binding capacity is detected by autoradiography (RNA band labelled with red asterisk). The proteins are detected by western blot (bottom) using anti-Flag antibody (protein band labelled with green asterisk). The assay was performed on the following class of proteins: **a** RNA-related proteins comprising the classical RRM domain-containing RBPs serrate RNA effector molecule (SRRT) and peptidylprolyl isomerase E (PPIE), as well as WD40 domain-containing protein receptor for activated C kinase 1 (RACK1) and splicing related protein Sin3A-associated protein 18 (SAP18). **b** Cell signalling proteins phosphatidylethanolamine binding protein 1 (PEBP1) and casein kinase 1 delta (CSNK1D). For PEBP1 an additional control with high amounts of RNase I treatment (100 units) is included alongside the standard (1 unit) treatment employed in all experiments. **c** Cytoskeleton ERM family proteins Radixin (RDX), Ezrin (EZR) and Moesin (MSN). An additional control with high amounts of RNase I is included for RDX. **d** Metabolic enzymes aldehyde dehydrogenase 6 family member A1 (ALDH6A1), malonyl-CoA-acyl carrier protein transacylase (MCAT) and short/branched chain acyl-CoA dehydrogenase (ACADSB)

[2,15] and mitochondrial tRNA/protein import (TOMM20) bind RNA (Supplementary Fig. 13e). Some of these proteins had been predicted (but not verified) to interact with RNA in previously published interactome capture experiments[2].

Second, we utilised PNK assays as an orthogonal approach[2,48] (see Methods). We first validated the assay using three known RBPs: SRRT, PPIE and RACK1 (Fig. 6a). Next, we applied the assay to SAP18, a splicing protein[49], which until now has not been confirmed to directly bind RNA (Fig. 6a). Third, we confirmed two proteins involved in cellular signalling as new RBPs, namely PEBP1/RKIP and CSNKD1. PEBP1 is a well-established tumour suppressor protein involved in a multitude of signalling pathways such as Raf/MEK/ERK, NFkB, PI3K/Akt/mTOR and Wnt[50]. The radioactive smear above PEBP1 is specific to radiolabelled RNA as it is sensitive to high amounts of RNase treatment (Fig. 6b). CSNKD1 represents another protein engaged in regulation of multiple pathways, including Wnt signalling, DNA repair and circadian rhythms[51]. We additionally validated the cytoskeletal ERM family proteins ezrin (EZR), radixin (RDX) and moesin (MSN), which had been predicted to interact with RNA in previous RBPomes[2,21] (Fig. 6c).

We were also able to verify RNA binding of three metabolic enzymes using PNK assay. Two of them, ALDH6A1 and MCAT, are newly identified RBPs with RNA-binding regions defined by our ADJ-RBPomes (and also predicted in previous RBPomes[2]) (Fig. 6d). ALDH6A1 plays a role in both the valine, leucine, isoleucine degradation (Supplementary Fig. 4b), while MCAT is involved in mitochondrial fatty acid beta oxidation. The short/branched chain acyl-CoA dehydrogenase CG3902 was predicted to bind RNA based on our FA-RBPome analyses. It is a member of both the fatty acid and valine, leucine, isoleucine degradation pathways (Supplementary Fig. 4b). We cloned its human ortholog ACADSB and validated its ability to directly bind RNA using the PNK assay (Fig. 6d). This finding demonstrates that the use of FA crosslinking can extend the discovery potential of comprehensive RNA interactome capture.

**Evolutionary conservation of globular RBDs.** We uncovered RBDs in 328 ortholog pairs in *Drosophila* and humans using CAPRI (Fig. 7a). The majority of the proteins which contained novel domains in *Drosophila* also contained novel domains in their human orthologs (Supplementary Fig. 14a and Supplementary Data 12). We also observed a considerable overlap in the Pfam domains recovered in both species (Fig. 7b). We identified 29 novel conserved Pfam RBDs and a further 46 novel InterPro RBDs (Fig. 7c, d). We classified the domains into metabolic enzymes, domains involved in the post translational modification of proteins and other structural domains. Over 60% of the novel Pfam RBDs from *Drosophila* were also identified in other mammalian studies[21,22,40,52] (Fig. 7e and Supplementary Fig. 14e and Data 13). We consider a few interesting examples of conserved RBDs below.

CAPRI successfully discovered additional conserved novel domains in proteins already known to carry a classical RBD. One such example is the human protein LARP1, in which CAPRI-peptides identified both a classical RBD (La-RRM) and the non-classical RBD DM15 (Fig. 8a top panel). CAPRI also mapped these same domains in the *Drosophila* ortholog larp (Fig. 8a bottom panel) and human paralog LARP1B (Supplementary Fig. 14b). We verified the RNA-binding activity of the human DM15 repeat by PNK assay (Fig. 8e). In support of our result, the DM15 repeat was recently independently shown to bind the mRNA cap[53]. We observed CAPRI-peptides identifying additional disordered regions (RG- and SR-rich regions preceding the La-RRM domain) in both flies and humans (Fig. 8a and Supplementary Fig. 14b). We also independently validated several other conserved globular RBDs: SPRY[54] and P-loop-containing nucleoside triphosphate hydrolase domains in HNRNPU (Fig. 8b, e) and its fly ortholog CG30122 (Supplementary Fig. 14c); as well as DZF domains in

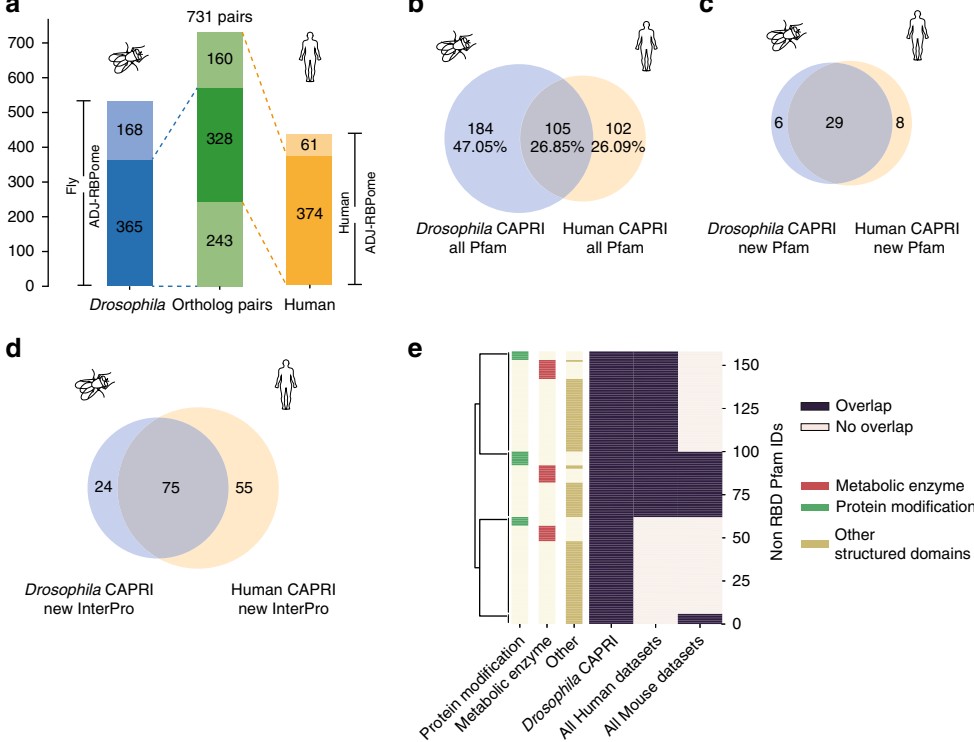

**Fig. 7** Conservation of RBDs between human and *Drosophila* proteins. **a** Mapping of proteins identified by ADJ-peptides in *Drosophila* (left) and humans (right) onto DOIPT ortholog pairs (centre). Proteins with no orthologs are noted by the lighter shade of colour in each of the species columns. **b** Overlap of all identified Pfam domains in *Drosophila* (blue) and humans (yellow). **c** Overlap of new Pfam domains in orthologous proteins identified in both species. **d** Overlap of new InterPro domains in orthologous proteins identified in both species. **e** Clustered heatmap representing all the non-RBD Pfam domains identified in *Drosophila* using the CAPRI protocol together with the identification of the same domains in all human (CAPRI, RBDpep[21] and pCLAP[40]) and mouse (RBDpep and RBR-ID[22]) protocols. The domains are manually annotated as belonging to three categories: metabolic enzymes, domains involved in protein modification and the remaining structured domains

human ILF3, ZFR, STRBP[21] and their fly ortholog Zn72D (Fig. 8c, e and Supplementary Fig. 14d). We further substantiated the reliability of novel RBD discovery by CAPRI by extrapolating a GTP-binding domain identified in *Drosophila* NON1 protein to its orthologous protein in humans, GTPBP4, and verifying it with PNK assay (Fig. 8d, e). We have further summarised interesting new conserved domains and the proteins harbouring them in Fig. 8f and Supplementary Fig. 15.

**CAPRI uncovers RNA-binding motifs in IDRs.** One of the strengths of the CAPRI methodology is its capacity to identify RNA-binding sequences in unstructured segments of protein, which are intractable by classical structural approaches. Indeed, >30% CAPRI-peptides mapped to IDRs. This is consistent with previous reports[21,22,29,55]. CAPRI sequences are rich in disordered amino acids and triplets of amino acids such as RGG, RSS, GFG and YGG, which are predicted to be RNA associated[29,55,56] (Supplementary Fig. 16 and Supplementary Note 6). We found many motifs using the MEME tool[57]. We classified the IDR-based motifs into four categories: charged arginine-containing motifs, aromatic motifs, single amino acid-repeat motifs and tandem repeats motifs.

The first category of motifs consists of positively charged RG[G] or RS motifs (Fig. 9a–c). These motifs can occur in tandem or in combination with other motifs in over a hundred sites in *Drosophila* and humans. Regions rich in RG[G]/RS repeats are positively charged and can bind the negative backbone of RNA[21,29,55,58]. We identified a new RG containing

motif with extended glycine repeats (Fig. 9b). It is important to note that the number of glycines in classical RG motifs (Fig. 9a) has been shown to influence binding to different sequences of RNAs[58]. Strikingly, we also found a novel class of motifs with alternating positively and negatively charged residues (RD/RE repeats) (Fig. 9c). All the above motifs contain R/S residues, which can undergo phospho/methyl/dimethyl modifications and such modifications can be used to control RNA binding of the disordered regions[29].

The second category of motifs we uncovered in IDR-based RBDs exhibit an aromatic amino acid-rich pattern with YG or NGF repeats (Fig. 9d, e). YG-rich repeats have been shown to undergo phase transitions and participate in RNA granule formation in vivo[59]. For example, we identify XL-peptides overlapping with the disordered YG-rich C-terminal region in human HNRNPA1 (Supplementary Data 7; spectra 27, 28). The spectra suggest a stacking interaction between the aromatic ring of tyrosine and RNA bases. The YG-rich sequence of HNRNPA1 also contains an amyloid-forming sequence, mutations in which have been associated with amyotrophic lateral sclerosis[60]. The third category of motifs is the tandem repeat motifs. In one example of such a motif, we found the DDDR sequence motif repeated 22 times in human EIF3A and once in its fly ortholog eIF3-S10 (Supplementary Fig. 17a, b). Similar repeats were observed in other human proteins like RBM12B and PPP1R10, along with *Drosophila* proteins like CG5787 (Supplementary Fig. 17c). Thus, use of short tandem repeats of amino acids in RBDs extends beyond the classical pentatricopeptide repeats.

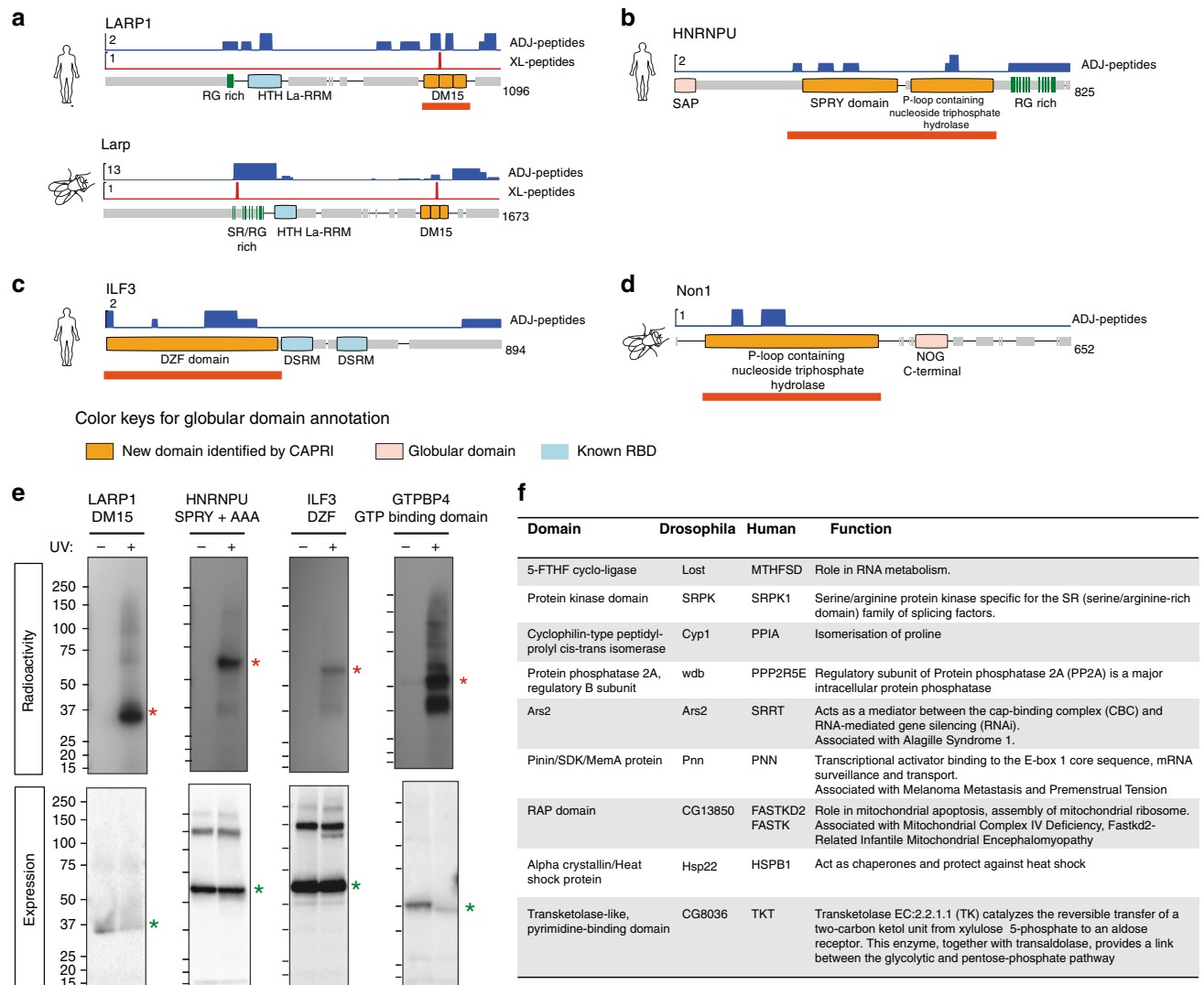

**Fig. 8** Validation of globular domains by PNK assay. **a** Conservation of non-classical RBDs between *Drosophila* and humans exemplified by the peptide coverage of *Drosophila* Larp protein. The tracks from top to bottom: ADJ-peptides (in blue), XL-peptides (in red), RG and RS rich region in the novel IDR (green bars), protein domains (classical RBDs in blue boxes, novel globular domains in orange boxes, other globular domains in faint pink boxes) and MobiDb disordered regions (grey bars) with protein length. **b** As in **a** for human RBP HNRNPU. **c** As in **a** for human RBP ILF3. **d** As in **a** for Non1 *Drosophila* protein (ortholog of human GTPBP4). **e** PNK assay performed to detect RNA-binding activity for the non-classical domains of the human proteins shown in **a–d**. Autoradiography (red asterisk) and western blot (green asterisk) using anti-Flag antibody after affinity purification of Flag-HBH-tagged domains in UV crosslinked ( + ) and non-crosslinked ( - ) HEK293 cells. From left to right: DM15 domain from LARP1, SPRY and AAA domains from HNRNPU, DZF domain from ILF3, and GTP-binding domain with Rossmann fold from GTPBP4. **f** Table describing a few notable conserved RBDs and their biological importance

One of the most important findings in our motif analysis was the presence of single amino acid-repeat motifs containing glutamine, alanine or histidine stretches (Fig. 9f–h). Consistent with this finding, the polyQ region of whi3 protein has been shown to be necessary for RNA-mediated granule formation in yeast[61]. PolyQ and polyA have also been implicated in protein aggregate formation in diseases like Huntington's disease and oculopharyngeal muscular dystrophy[62].

**Evolutionary conservation of RNA-binding IDRs.** The frequencies of RBD motif usage show species-specific bias (Fig. 9a–i). Among the subtypes of R-containing motifs, the G-rich subtype (Fig. 9b) is more frequently observed in *Drosophila*, whereas the RG repeat subtype is more abundant in humans (Fig. 9a). With regard to aromatic motifs, the NFG-containing motif is more frequently observed in *Drosophila* compared to humans (Fig. 9e). Among the single amino acid-repeat motifs, *Drosophila* CAPRI-

peptides are richer in polyQ, polyA and polyH relative to humans, whereas polyP is preferred in humans. This mirrors the fact that relative to other eukaryotes[63,] the *Drosophila* proteome exhibits higher levels of single amino acid repeats in general and polyQ repeats in particular[64]. Indeed, polyQ motifs are identified only in *Drosophila* CAPRI-peptides (Fig. 9g). This differential usage of various motifs in disordered regions of these organisms could be relevant for the functions of these proteins.

Comparison of CAPRI-peptides mapping to IDRs strongly suggests that these regions could exhibit conserved RNA-binding abilities in spite of significant sequence divergence. The IDRs are not constrained by structure, and hence their sequences can tolerate higher mutation rates compared to those of globular domains. The high sequence divergence makes it difficult to predict functions in silico across species. The CAPRI workflow enabled us to obtain a set of functionally conserved IDR regions from which we could draw salient

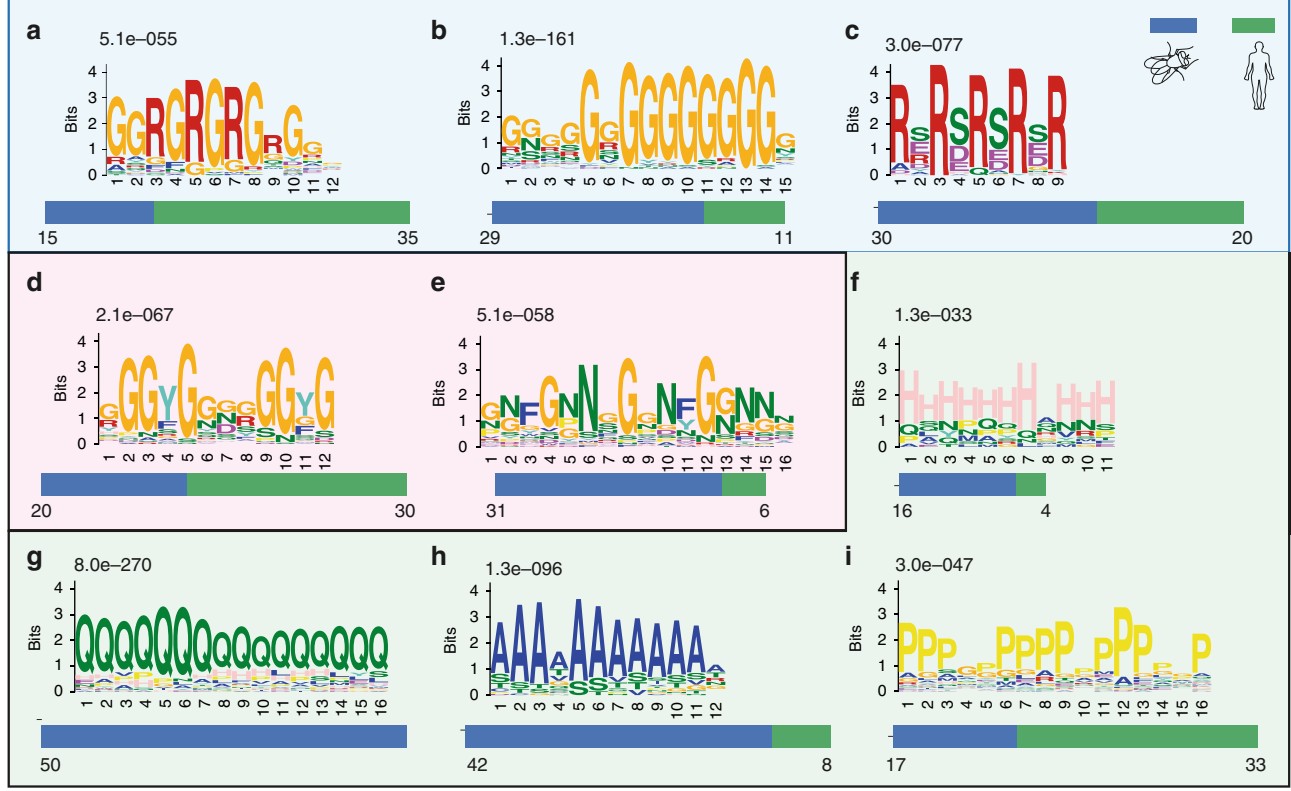

**Fig. 9** Motif usage in IDRs of CAPRI-peptides. **a–c** Illustration of arginine-based motifs discovered by MEME analysis[57] in disordered regions of *Drosophila* and human ADJ-peptides. *E*-values displayed on top of each motif are calculated by the MEME tool and number of sites (*n*) are shown for each of the motifs. The number of sites observed in each of the species are represented as bars below each motif. The maximum number of sites allowed for motif construction is 50 (see Methods for details). **d–e** As in **a** for aromatic amino acid (Y or F)-based motifs discovered in both *Drosophila* and humans. **f–i** As in **a** for repetitive single amino acid-based motifs discovered in both *Drosophila* and humans

features of conservation of RNA-binding function. We found a total of 142 ortholog pairs in which an IDR was detected in both species (Supplementary Fig. 17d). Of these, 40 pairs exhibited positional conservation of the RNA-interacting IDRs when manually evaluated. By studying 10 of these pairs we shed light on specific features that correlate with conservation of RNA-binding function in IDRs based on the changes observed at the sequence level (Fig. 10a). We grouped these features into five sets.

In the first set, we observed conservation of sequence with minor changes. We discovered an intrinsically disordered RG-rich region, which binds RNA in both human UBAP2L and its *Drosophila* ortholog Lig (Fig. 10b). On performing multiple sequence alignment, we note that the RG-rich region was conserved across a much broader range of species than just between *Drosophila* and humans (Supplementary Fig. 18a). We verified the RNA-binding capacity of human UBAP2L and its candidate RBD by PNK assay (Fig. 10c).

The second set contains proteins, which possess one of the motifs of the R-type (Fig. 9a–c), but with different subtypes used between species. We observe a tendency for RG-rich sequences to contain more spacer glycine repeats in *Drosophila* (Fig. 9b) compared to their orthologous counterparts in humans (Fig. 9a). This is exemplified by the C-terminal RG repeats in human apoptosis inhibitor protein API5 and the corresponding region in the fly ortholog Aac11 (Supplementary Fig. 18b). Such use of intervening spacer glycines has been reported to affect RNA sequence preference[58]. We found many examples of similar sequence changes in other protein pairs, which are summarised in Fig. 10a and illustrated in Supplementary Fig. 18c, d.

Third, we observed an expansion of DDDR repeats (Supplementary Fig. 17a) in EIF3A. There is positional conservation of the RNA interaction function in the C-terminal region of human EIF3A and *Drosophila* eIF3-S10 (Supplementary Fig. 17b). However, the DDDR repeat motif is repeated 22 times in human EIF3A and only once in its fly ortholog eIF3-S10. It has been hypothesised that the expansion of tripeptide repeats in IDRs of EIF3A orthologs (yeast to humans) represents an emergent property to enable increasingly diverse RNA interactions in complex interactomes[13].

The fourth set consisted of a single IDR region in the ortholog pair of YBX1 (human) and Yps (fly). Although we could not determine any clear motifs or extensive sequence conservation between the ortholog pair, the CAPRI-identified IDRs of the two proteins exhibited conserved amino acid composition (Supplementary Fig. 18e).

The IDRs which did not fit into any of the above groups formed the fifth set. For some ortholog pairs, we observed a complete change of IDR sequence in the CAPRI-peptides. For example, the arginine-rich sequence in YLPM1 contains a RE/RD/RG-rich stretch followed by a proline-rich motif. The corresponding region in its fly ortholog ZAP3 is rich in glycines with YG/GP repeats (Fig. 10d). The human YLPM1 RNA-interacting region was verified to bind RNA using the PNK assay (Fig. 10c, d). Another example of a change in a RG region is the ortholog pair of PRRC2A (human) and *Drosophila* nocte (Fig. 10e). PRRC2A contains closely spaced RG repeats, however Nocte contains a few dispersed RG sequences in the IDR region in the middle part of the protein. We verified the human domain to bind RNA by PNK assay (Fig. 10c).

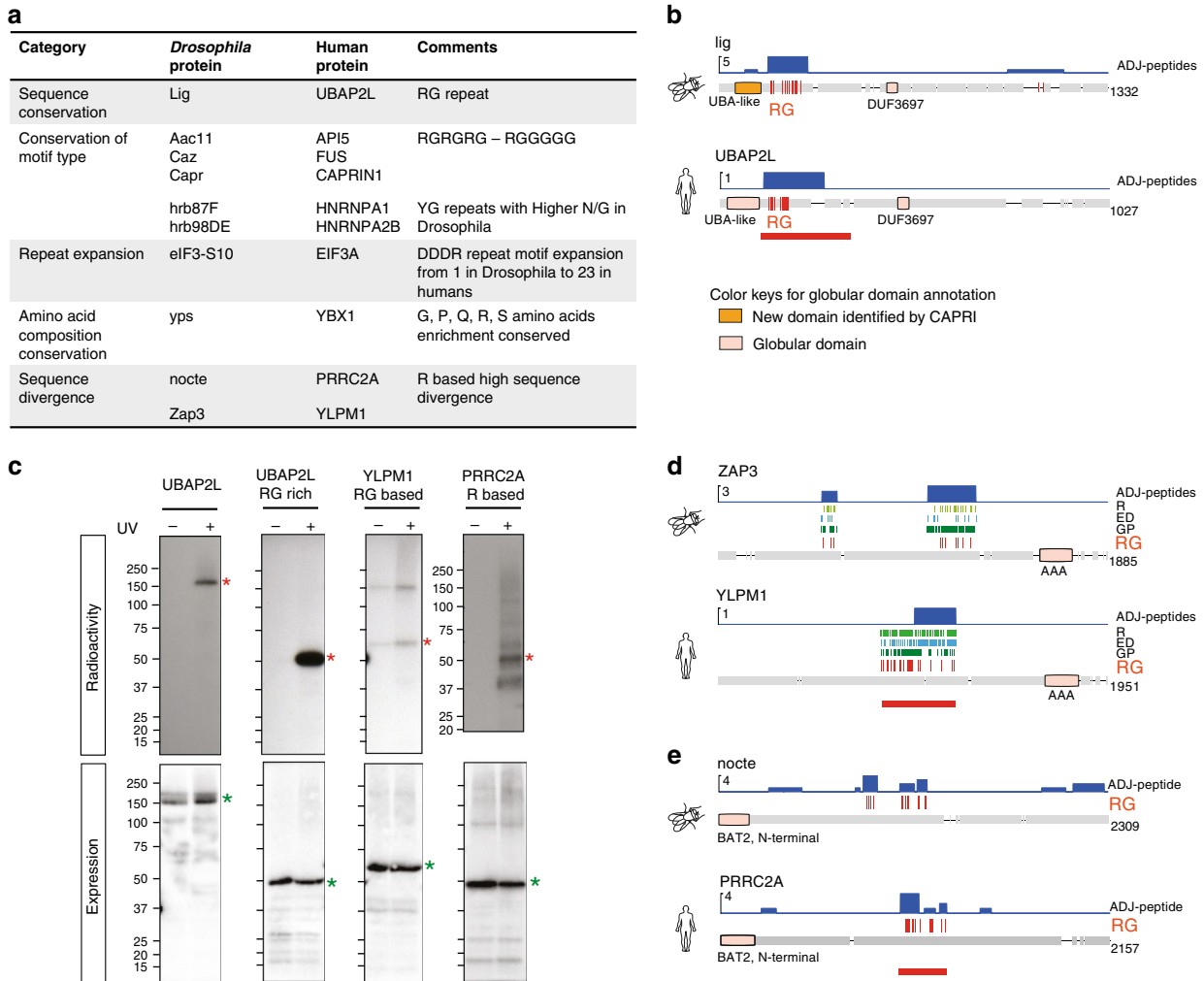

**Fig. 10** Conservation of disordered RNA-binding regions between human and *Drosophila* proteins. **a** Table showing examples of different modes of evolutionary conservation of RNA interacting function in disordered regions of orthologous proteins. **b** Peptide coverage for the conserved RNA-binding RG rich region in Lig2 protein (*Drosophila*) and its ortholog UBAP2L protein (human). Regular expression annotations (RG|GR in red) are restricted to RNA interacting regions. Globular domains in boxes (globular domains identified by CAPRI in orange, other globular domains in faint pink). Region cloned for verification is shown by a red bar. **c** PNK assay performed to detect RNA interaction of disordered regions in human proteins described in **b**, **d**, **e**. Autoradiography (red asterisk) and western blot (green asterisk) using anti-Flag antibody after affinity purification of 3Flag-HBH-tagged disordered regions in UV crosslinked ( + ) and non-crosslinked ( - ) HEK293 cells. From left to right: Human protein UBAP2L, RG rich regions of UBAP2L, RG-based region of YLPM1 and R-based PRRC2A. **d** Peptide coverage in Zap3 protein (*Drosophila*) and its human ortholog YLPM1 protein. Regular expression annotations (R, ED, GP, RG|GR) are restricted to RNA interacting regions in *Drosophila* and to the cloned region in YLPM1 (depicted by a red bar). **e** Peptide coverage in nocte (*Drosophila*) and PRRC2A (human) proteins. Regular expression annotations (RG|GR) are restricted to RNA interacting regions

## Discussion

In this study, we present a comprehensive analysis of the *Drosophila* RNA-protein interactome, identifying 2327 polyA+ RNA-associated proteins (FA-RBPome) and 1512 direct polyA+ RNA-binding proteins (UV-RBPome). Employing FA crosslinking enabled us to identify a secondary interaction layer comprising proteins involved in metabolism, endocytosis and DNA replication (Fig. 2e and Supplementary Figs 4 and 12). We independently verified that 14 of these proteins bind RNA using the orthogonal PNK assay and a newly developed silica-based assay (Fig. 6 and Supplementary Fig. 13). These results suggest widespread crosstalk between the post-transcriptional machinery and other central cellular pathways.

We established a new workflow, CAPRI, which enables comprehensive and accurate discovery of RBDs in the cell through simultaneous detection of RNA-protein crosslinked sites and peptides adjacent to the crosslinks. We successfully employed a

combined de novo and database search using PEAKS software to identify 1790 unique XL-peptides in Drosophila and human cells. In addition to XL-peptides, we also mapped 3119 ADJ-peptides in over a thousand *Drosophila* and human proteins. The simultaneous detection of ADJ-peptides and XL-peptides from the same sample is a technological advance, which permits CAPRI to extend its coverage and accuracy. Additionally, PEAKS' user-friendly interface and automated internal fragment annotation of crosslinked spectra, and of single nucleotide crosslinked spectra in particular, make CAPRI accessible to non-mass spectrometry experts. In future, the coverage could be extended by using new combinations of proteases, including Arg-C[21], GluC and chy-motrypsin[65]. CAPRI could also be easily adjusted to incorporate elements from recently published adjacent-only workflows such as isotope label-based quantification[21] and smaller experimental scale[40] to achieve even better results. The filter-based enrichment of RNA-crosslinked peptides used in CAPRI sets it apart from

other protocols as it allows for highly stringent washes along with both easy and economical enrichment of crosslinked peptides (Supplementary Fig 8a). CAPRI is also compatible with recently published methods developed for isolating total RNA interactomes[66–68].

In addition to capturing canonical RBDs, we could also identify 234 new structured RBDs. Comparing the human and *Drosophila* orthologs, we demonstrated the conservation of many globular domains (Figs. 7c–e and 8 and Supplementary Figs 14 and 15). We also discovered that > 30% of the CAPRI-peptides map to IDRs. We validated several of these regions individually using an independent in vivo assay. We not only identified classical RNA-associated motifs in IDR regions[21] but also found novel motifs (Fig. 9) with a clear bias for single amino acid motifs in *Drosophila* compared to humans. It is likely that these motifs are responsible for RNA-mediated phase transitions, especially in combination with other RBDs[60,61].

Since IDRs show high sequence divergence and in some cases accumulation of repeats[13] through evolution, it is surprising that we observe positional conservation of RNA interaction sites between orthologous IDR pairs[13]. It will be interesting to see if the IDR pairs we found would bind to similar or diverse RNA sequences. It is tempting to speculate that the use of small motifs and degenerate sequences in RNA-binding IDRs enables complex organisms to rapidly adapt to their increasingly diverse transcriptomes[69]. Application of CAPRI to a more evolutionarily distant species like *S. cerevisiae* would permit us to get an even better insight into the evolution of RNA-binding disordered regions.

## Methods

**Formaldehyde crosslinking of S2 cells**. Wild-type S2 DRSC cells (Stock #181 from Drosophila Genomics Resource Centre) were grown to exponential phase (6–8 million cells/ml) in Schneider (S2) medium in shake flasks at 70 rpm at 25 °C. The cells were crosslinked directly in the medium with 0.1% (v/v) FA (0.05–1% FA was used for optimisation) at room temperature for 10 min rotating at 8 rpm. The FA was quenched by adding glycine (final concentration 125 mM) and rotating for 5 min at 8 rpm. The cells were pelleted by centrifugation at 600 x $g$ for 5 min. These cells were resuspended and washed in cold PBS and pelleted again. The cell pellets were used further or flash frozen in liq. N₂ and stored at −80 °C for a maximum period of 1 month.

**UV crosslinking of S2 cells**. The cells were counted and plated out on cell culture dishes at 60% confluency overnight. The next day the medium was removed and cold PBS is added (4 ml for 15 cm dishes, 10 ml for 25 × 25 cm² plates). In case of cells growing in shake flask, cells were pelleted in the medium by spinning at 500 x $g$ for 5 min. For large-scale experiments, 450 × 10⁶ cells were resuspended in 10 ml ice cold PBS and spread on a 25 × 25 cm² plate. The plates were placed immediately on an ice-water mixture, 4 cm away from 254 nm wavelength UV lamps and irradiated with a total energy of 200 mJ/cm². The crosslinked cells were immediately scraped off the plates and pelleted by spinning at 500 x $g$ for 5 min at 4 °C. After discarding the supernatant the cell pellet was flash frozen in liq. N2 and stored at −80 °C for a maximum of 1 month.

**UV crosslinking of HEK293 cells**. HEK293 (Flp-In™ T-REx™-293, Thermo Fisher R78007) cells were grown in 10% FBS in DMEM medium to 90% confluency in 15 cm² plates. The plates were washed two times with PBS. Finally 4 ml cold PBS was added and the plates were placed 3 cm away from 254 nm wavelength UV lamps and irradiated with energy of 200 mJ/cm². The crosslinked cells were immediately scraped off the plates and pelleted by spinning at 350 x $g$ for 4 min at 4 °C. After discarding the supernatant the cell pellet was flash frozen in liq. N2 and stored at −80 °C for a maximum of 1 month.

**Whole-protein RNA interactome capture**. UV or FA crosslinked and non-crosslinked cells were used to isolate polyA+ RNA by oligo-dT magnetic beads from *Drosophila* S2 DRSC and HEK293 cells. The proteins covalently crosslinked to the RNA were isolated, purified and ultimately analysed by mass spectrometry. The following protocol was adopted from two previous publications[11,12]. It has been streamlined for the particular cell line (Schneider cells, HEK293 cells) by optimising crosslinking and, homogenisation conditions, cell number to lysis buffer ratio and washing steps.

Non-crosslinked and crosslinked (UV or FA crosslinked) frozen cell pellets were resuspended quickly in 10 ml of OLB (Oligo-dT Lysis Buffer − 50 mM Tris pH 7.5, 10 mM EDTA, 1% lithium dodecyl sulphate (LiDS), 0.5 M LiCl, 5 mM DTT, 1x cOmplete protease inhibitor from Roche) by using a pipette. The lysate was immediately homogenised to shear genomic DNA with a rotor stator homogeniser for 90 s (UV) or 60 s (FA), respectively at 2000 rpm. The extracts were diluted to 40 ml with OLB. The extract was spun at 10,000 x $g$ for 10 min at 4 °C to pellet aggregates if any. The extract was incubated with a suspension of 4 ml oligo-dT magnetic beads (NEB) at room temperature for 1 h on a rotating wheel (set at 10 rpm).

After incubation, the beads were collected on a 50 ml NEB magnetic stand for 25 min at room temperature (RT). The supernatant was removed and flash frozen or stored on ice for a second round of purification. The beads were washed at RT with 40 ml of OLB for 5 min (10 rpm, rotating wheel). All further wash steps were carried out at 4 °C. Briefly, the beads were serially washed once with 40 ml WB1 (oligo-dT wash buffer: 20 mM Tris-HCl, pH 7.5, 0.1% LiDS, 500 mM LiCl, 1 mM EDTA, 5 mM DTT), twice with 40 ml WB2 (oligo-dT wash buffer: 20 mM Tris-HCl pH 7.5, 0.05% LiDS, 200 mM LiCl, 1 mM EDTA, 5 mM DTT) and finally in a smaller 1.5 ml tube twice with 1 ml WB3 (oligo-dT wash buffer: 20 mM Tris-HCl pH 7.5, 200 mM LiCl, 1 mM EDTA, 5 mM DTT). The RNA was eluted in 400 μl of Elution buffer (20 mM Tris-HCl pH 7.5, 1 mM EDTA) by heating the mixture in a thermomixer at 56 °C for 4 min followed by removal of magnetic beads (magnetic stand). The beads were routinely reused for a second round of RNA isolation from the same cell extract. The RNA concentration and quality in each eluate were evaluated by employing Qubit assays (DNA-HS, Thermofisher Q32854; and RNA-BR, Thermofisher Q10210), agarose gel electrophoresis. To evaluate efficiency of mRNA capture equivalent fractions of input and eluted RNA were finally purified (subsequent to complete protein degradation) and quantified by quantitative reverse transcription PCR (RT-qPCR). To check for genomic DNA contamination by qPCR, quantification was performed on equal volumes of reverse transcribed eluate (cDNA) and non-reverse transcribed eluate (genomic DNA).

Three biological replicates of RBPome protein samples (UV crosslinked, no UV, FA crosslinked, no FA; 12 samples in total) were subjected to in-gel tryptic digestion by reconstituting them in 1x LDS sample buffer followed by separation on 4–12% Bis-Tris gels (NuPAGE) and colloidal Coomassie staining (Instant Blue, Expedeon). Entire gel lanes were sliced into six pieces and processed (Promega trypsin) by standard in-gel trypsin digestion[70]. Finally, tryptic peptide mixtures were desalted using C18 reversed-phase STAGE tips[71].

**Selection of RNA retaining filters for use in CAPRI**. The FASP filters were selected after screening for total RNA retention. We screened Amicon 30 kD MWCO and Microcon 30 kD and 10 kD MWCO filters by adding *Drosophila* total RNA on the filters and subjecting them to the standard FASP[31] washes with 200 μl of 8 M urea in 0.1 M Tris-HCl, pH 8.0 (UA buffer) by centrifugation (14,000 x $g$ for 20 min), 4 M Urea in 0.1 M Tris-HCl, pH 8.0 (UB buffer) by centrifugation (14,000 x $g$ for 20 min). Next, 50 μl of 4 M urea, 50 mM Tris-HCl, pH 7.9 was added to the filters and this solution was diluted to a final urea content of 1 M by addition of 150 μl 50 mM Tris-HCl, pH 7.9. The mixture was incubated with 0.5 μg of trypsin at 25 °C for 3 h. The RNA retained on the filter was collected by inverting the filter and centrifugation at 14,000 x $g$ for 20 min. After centrifugation, the RNA in all the filtrates and eluates was ethanol precipitated and analysed on a 1% agarose gel.

Filters were also screened for retention of small nucleic acid oligos (36mer ssDNA oligo, 21mer ssRNA oligo, 11mer ssRNA oligo, 6mer ssRNA oligo). Recovery of the oligonucleotides in the flowthrough was calculated by analysing input and flowthrough samples in a Nano-drop UV spectrometer.

**Isolation of peptides in CAPRI**. CAPRI combines on-bead digestion of covalently linked RNA–protein complexes with RNA-FASP in order to subsequently purify adjacent peptides and UV crosslinked ribonucleotide-peptide heteroconjugates.

In all, 4.4 × 10⁸ crosslinked (UV or FA) S2 cells or eight 15 cm² plates of UV crosslinked HEK293 cells were resuspended in oligo-dT lysis buffer and homogenised by using a rotor stator homogeniser. The extract was cleared by spinning at 10,000 x $g$ for 10 min and added to 4 ml suspension of oligo-dT magnetic beads. The mixture was incubated at room temperature for 1 h rotating at 10 rpm on a wheel. The beads were washed once with 40 ml OLB, once with 40 ml WB1 and twice with 40 ml WB2 at 4 °C. The beads were transferred to a 1.5 ml microcentrifuge tube and washed with 1 ml OBD buffer (50 mM Tris-Cl pH 7.8, 200 mM LiCl, 1 mM EDTA). The on-bead digestion was performed at 25 °C for 3.5 h in 400 μl of OBD buffer (rotating at 4 rpm) by addition of 1 μg Lys-C (Wako) per 40 μg of proteins. Next, the beads were separated (magnetic stand) and the supernatant containing released peptides was removed. The beads were transferred to a 5 ml protein low-binding tube and washed once with WB2 and three times with WB3 both at 4 °C. The RNA-polypeptide heteroconjugates were eluted from the beads in 400 μl of Elution buffer (10 mM Tris-Cl, pH 7.5) by constant agitation (1100 rpm, thermomixer) at 56 °C for 4 min.

Eluted RNA-polypeptide heteroconjugates were adjusted to a final concentration of 2 mM EDTA, 50 mM NaCl, and 5 mM DTT. The samples were

directly concentrated on 30 kD MWCO Microcon ultrafiltration devices (Millipore, forensic grade) by centrifugation at 16,000 x *g* for 30 min at 20 °C. In all subsequent steps, solutions on the filter were first mixed in a thermomixer at 600 rpm for 1 min (RT). The filter units were first washed once with 200 µl of 10 mM EDTA, 0.1 M Tris-HCl, pH 7.5, and then with 200 µl of 8 M Urea in 0.1 M Tris-HCl, pH 7.9 by centrifugation (14,000 x *g* for 20 min). For cysteine alkylation an aliquot of 5 mM Iodoacetamide (IAA) was added and the solution was incubated at 25 °C for 15 min in the dark. IAA was washed away by centrifugation (14,000 x *g* for 20 min). This was followed by a wash with 100 µl of UB buffer (50 mM Tris-HCl, pH 8.0, 4 M urea). Lys-C (0.5 µg) was added to the filter in 100 µl UB buffer for another round of digestion and incubated at 25 °C for 6 h for *Drosophila* S2 cells (10 h for HEK293 cells). After digestion the released peptides were washed away with a 100ul 0.5 M NaCl wash. To estimate the efficiency of Lys-C digestion, these peptides were cleaned-up using C18 STAGE tips[71] and analysed by mass spectrometry. The RNA-peptide heteroconjugates retained on the filter were further washed with 100 µl of 4 M Urea, 0.1 M Tris-HCl, pH 7.9. Next, 50 µl of 4 M urea, 50 mM Tris-HCl, pH 7.9 was added to the filters and this solution was diluted to a final urea content of 1 M by addition of 150 µl 50 mM Tris-HCl, pH 7.9. The RNA-peptide heteroconjugates were further digested overnight with 0.5 µg of trypsin at 25 °C. Next day, the released adjacent tryptic peptides (liberated from site of RNA crosslink) were collected by spinning (16,000 x *g*, 20 °C, 30 min). The filter units were further washed with 100 µl of 1 M urea, 50 mM Tris-HCl, pH 7.9. Both filtrates were combined and labelled as pool A peptides. The RNA-peptide complexes that have remained on the filter were again resuspended in 50 µl of 4 M urea, 50 mM Tris-HCl, pH 7.9 buffer. The urea was diluted to 1 M final concentration exactly as described above. To degrade RNA (and potential traces of genomic DNA), 1 µl benzonase (25 U, Novagen) was added and the mixture was incubated for 30 min at 37 °C. Subsequently an RNase cocktail consisting of 0.5 µl RNases A (Thermo Scientific, 10 µg/µl), 1.0 µl RNase-T1 (Ambion, 1 µg/µl, 1 U/µl) and 0.5 µl RNase I (Ambion, 100 U/µl) were added and incubated first for 60 min at 37 °C and second for 90 min at 52 °C with the aim of completely trimming the remaining RNA-peptide heteroconjugates to a length of one to maximally three ribonucleotides. Following a final proteolytic digestion step (0.1 µg trypsin, 3 h at 37 °C) the released peptide-ribonucleotide heteroconjugates were collected (16,000 x *g*, 30 min at 20 °C). The filter units were washed for the last time with 50 µl 0.5 M NaCl, 10 mM Tris-HCl, pH 7.4 and centrifuged as described above. Ultimately, the filtrates harbouring the ribonucleotide crosslinked peptides were combined and labelled as pool B peptides. All peptide samples were acidified with formic acid (0.1% final concentration) prior to C18 STAGE tip column clean-up[71]. Both pools (A and B) were analysed separately by LC/MS (described below). To identify adjacent peptide the.raw files from pool A and B were processed together by MaxQuant. To identify crosslinked peptides only the.raw files from pool B were used. TFA was omitted from all steps throughout this protocol.

**Isolation of FA-dom-peptides.** FA crosslinked RBPs were purified from $4.4 \times 10^6$ S2 DRSC cells with oligo-dT beads as described before at 4 °C. Once the RNA had been captured on beads, the proteins were digested. To pursue this the beads were transferred to a 1.5 ml microcentrifuge tube and washed with 1 ml OBD buffer (50 mM Tris-Cl pH 7.8, 200 mM LiCl, 1 mM EDTA). The on-bead digestion was performed at 25 °C for 1 h in 400 µl of OBD buffer (rotating at 4 rpm) by addition of 1 µg Lys-C and 1 µg of trypsin. After 1 h the supernatant was removed and replaced with 400 µl of OBD buffer for another round of digestion with 1 µg trypsin for 2 h. Next, the beads were separated and the supernatant containing released peptides was removed. The beads were transferred to a 5 ml protein low-binding tube and washed once with 4 ml OLB, once with 4 ml WB2, three times with 4 ml WB3 and finally with prechilled 1 ml 200 mM ammonium bicarbonate (ABC) buffer for 2 min each. The RNA was eluted with 200 µl of 50 mM ABC buffer at 56 °C for 3 min and again with 100 µl of 50 mM ABC buffer at 56 °C for 3 min. The eluates were cleared of remnant beads by using a magnet and spinning at 2000 x *g* for 5 min. The RNA peptide bonds were de-crosslinked by incubating the eluates at 65 °C, 1100 rpm for 90 min. We took an aliquot of RNA for Bioanalyzer capillary electrophoresis. The remaining RNA was degraded with a mixture of 0.5 µl of benzonase, 0.5 µl of RNase A and 0.5 µl of RNase I for 90 mins at 37 °C at 1100 rpm and for another 30 min at 52 °C at 1100 rpm. The peptides were further digested with 0.5 µg trypsin and cleaned-up for liquid chromatography–mass spectrometry (LC-MS) analysis by SP3 purification[72].

**Peptide LC/MS analysis.** General nanoLC-MS set-up was similar to one in Musa et al.[73]. Briefly, Q Exactive mass spectrometer (Thermo Fisher Scientific) interfaced with an Easy nLC1000 UHPLC system (Thermo Fisher Scientific) was used for all experiments. For chromatographic separation of peptides, peptides were analysed on in-house packed fused-silica emitter microcolumns ((75 µm ID, 8 µm tip, 250 µm length; (SilicaTip PicoTips; New Objective) packed with 1.9 µm reverse-phase ReproSil-Pur 120 C18-AQ beads (Dr. Maisch)). For RBPome samples, peptides were separated by a 4 h linear gradient of 5–80% (80% ACN, 0.1% formic acid) at a constant flow rate of 300 nl/min. For RBD samples, CAPRI-peptides were separated by a 1 h linear gradient of 5–80% (80% ACN, 0.1% formic acid) at a constant flow rate of 250 nl/min. For DDA acquisition the "fast" (RBPome) and "sensitive" (CAPRI-peptides) method from Kelstrup et al.[74] was adopted with the following alterations. The full scan was performed at 70,000

resolution (at *m/z* 200) with a scan window of 350–1650 *m/z*. The automatic gain control target for MS1 was set to 3e6 and for MS/MS scan it was set to 1e5. A top10 workflow at a MS/MS resolution of either 17,500 or 35,000 (depending on sample complexity) for selecting the most abundant precursor ions in positive mode for HCD fragmentation (NCE = 28) was employed. Precursor ion charge state screening was enabled, and all unassigned charge states, as well as singly charged ions, were rejected. The lowest fixed mass recorded in MS2 spectra was set to *m/z* 100 ensuring the detection of all RNA derived marker ions in ribonucleotide-peptide heteroconjugate spectra. Selected ions were excluded from repeated fragmentation in a time frame of 30 s (CAPRI-peptides) or 60 s (RBPome samples).

**MaxQuant analysis.** DDA MS raw files for RBPome, CAPRI adjacent peptides and FA-dom-peptides were analysed by MaxQuant[32] software (version 1.5.2.8) and peak lists were searched either against the *Drosophila* or human Uniprot FASTA database (version September 2015) concatenated with a common contaminants database by the Andromeda search engine embedded in MaxQuant. Cysteine carbamidomethylation was set as a fixed modification and N-terminal acetylation, deamidation (NQ) and methionine oxidation as variable modifications. FDR was set to 1% for proteins and peptides, respectively and was determined by searching a reverse database. Enzyme specificity was set to trypsin (enabling cleavage N-terminal to proline), and a maximum of two (RBPome) or three (CAPRI) missed cleavages were allowed in the database search. For deciphering Lys-C missed cleavages four missed cleavages were allowed and enzyme specificity was restricted to Lys-C. Peptide identification was performed with an allowed initial precursor mass deviation up to 4.5 ppm and an allowed fragment mass deviation of 25 ppm with a minimum required peptide length of 6 amino acids and a maximum peptide mass and charge of 4600 Da and 7 + , respectively. The "match between runs" (matching time window 0.7 min), and "second peptide" features were enabled. Label-free quantification was done using the maxLFQ algorithm. Protein groups were identified with at least two peptides, wherein one of them should be unique to this protein group.

**UV-RBPome and FA-RBPome data analysis.** The MaxQuant proteinGroups.txt output file was used to identify UV and FA RBPomes. Contaminant protein groups were removed. Missing raw intensity values were imputed with minimum intensity observed. Protein groups defined by the presence of more than two unique peptides were statistically analysed by a moderated two-sided *t*-test from the Limma package (FDR < 0.01) comparing crosslinked to the respective non-crosslinked samples and subsequently filtered by an average intensity cutoff ( > eightfold). A finalised unique list of genes was selected from the first protein of the majority protein ID column in MaxQuant output tables.

**Analysis of ADJ-peptides.** We used the peptides.txt output files from MaxQuant analysis to catalogue tryptic adjacent peptides that are significantly enriched in UV crosslinked samples. Peptides mapping to contaminants were removed. Missing intensity values were imputed with the minimum peptide intensity observed. Peptides were selected based on a moderated *t*-test (Benjamini–Hochberg FDR < 0.05) comparing crosslinked to the respective non-crosslinked samples and subsequently filtered by an average intensity cutoff ( > eightfold). Peptides mapping multiple times to the very same protein sequence (UniProt ID) were removed. An in-house python script was developed to assemble a database of adjacent peptides, uniprot sequences and Pfam domain identifications. The tryptic adjacent peptides were extended in silico to the next nearest (theoretical) Lys-C cleavage site (lysine) in both directions (N- and C-terminal extension) and were named ADJ-peptides. Finally, the longest protein isoform possessing the largest total number of mapped ADJ-peptides was chosen to represent each gene based on Ensembl gene ID. This protein list constitutes the ADJ-RBPome. Peptide coverage was calculated for each amino acid position in the protein sequence based on the number of tryptic adjacent peptides identified at the position. Comparison between full-protein peptides and ADJ-peptides was performed by treating the MaxQuant peptides.txt output file from interactome capture similar to the tryptic adjacent peptides.

Domain peptides from published studies were taken from their respective reports and integrated into the CAPRI database. Analysis of RBDpep (equivalent of adjacent peptide in CAPRI protocol)[21,52] and pCLAP[MS] (ref.[40]) peptides was performed similar to adjacent peptides to yield extended-RBDpeptides and extended-pCALP peptides (equivalent to ADJ-peptides). The RBDpeptides were extended to the next Arg-C or Lys-C digestion sites based on the enzyme used in domain enrichment. RBR-ID[22] peptides were mapped to the given proteins without any in silico extension.

**Analysis of XL-peptides.** In previous studies, the crosslinked peptides were analysed using specialised database-based searches[18,19,75,76] (for a brief summary see Supplementary Note 4). In CAPRI, we employ a benchmarked commercial software (PEAKS Studio, BSI, Canada) that is easy to use and that combines computational peptide de novo sequencing (to derive peptide sequence tags: PSTs) with conventional database searching[36,37]. Two decades ago, pioneering work by Mann and Wilm[77] suggested that PST assisted database searching is both error

tolerant and enables the identification of peptide sequences bearing unknown PTMs. Hence, instead of heuristic filtering applied prior to bioinformatic data analysis (RNP$^{XL}$) the concept of PSTs is harnessed to select interpretable MS2 spectra (exhibiting good partial peptide-spectrum matches: PSMs) from the complete raw dataset in order to reduce the search space for identifying the ribonucleotide adduct. Briefly, PEAKS identification of peptides is made of three steps: De novo spectrum analysis, database search (PEAKS-DB) and PTM search (PEAKS-PTM). The identification of PTMs is achieved by integrating the database searching with the initial de novo sequencing analysis (PST generation). Importantly, its algorithm maximises PTM identification with the possibility to make multiple custom modifications. The PEAKS de-novo algorithm is supposed to be more tolerant towards gaps because it does not use the graph theory model for PSM analysis but tries to select the best possible sequence of amino acids using local probabilities of amino acid identity. The database peptide mapping can take place with multiple PSTs predicted from the same MS2 spectrum. This allows complex PTMs to be matched in a spectrum[78]. For more detailed information a description of RNA adduct annotation is outlined below.

The HCD/CID fragmentation of crosslinked peptides preferentially cleaves the glycosidic and phosphodiester bonds but not the covalent link between the base and the amino acid side chain (Supplementary Fig. 7a). Thus, the ribonucleotide (A, G, C, U) crosslink reacts in the gas phase of the mass spectrometer as a single PTM composed of two parts: the ribonucleotide base (A', G', C', U') and the rest of the ribonucleotide (ribose + phosphate (PO4)). Importantly, this rest can fall off as a neutral loss during HCD/CID fragmentation. This behaviour (fractional PTM neutral loss) is different compared to classical PTM neutral losses (e.g., phosphoric acid loss for serine/threonine phosphorylation, carbohydrate loss for serine/threonine O-glycosylation) because in the latter cases the entire molecule that defines the PTM gets dissociated during HCD/CID.

In the PEAKS software version 8.0, it is not possible to define a custom PTM annotation, which would provide fractional neutral loss annotation of a particular PTM. To account for this, we formally split each of the ribonucleotide into two PTM sets; the nucleobase (A', G', C', U') and the rest of the nucleotide (ribose + PO$_4$). With these PTM definitions, peptides containing a single nucleotide (ribo-mononucleotide) crosslink will be annotated by PEAKS to contain one of the bases (e.g., U') at the site of modification (amino acid residue) and "rest" (the part of the PTM that dissociates during HCD/CID) annotated at one of the peptide termini. We also included an additional PTM of "cyclic-rest" to account for the possibility of cyclised ribose sugar phosphate left behind as a by-product of RNA degradation by RNase enzymes (definitions of RNA-PTMs in Supplementary Data 4).

For ribo-dinucleotides we simply define two sets again as a combination of crosslinked base (A', G', C', U') and the remaining modification as the sum of a ribo-mononucleotide and "rest" (ribose + phosphate) to end up with a complete (standard) neutral loss (of a PTM), instead of fractional neutral losses. For example, if there are two nucleotides UA crosslinked to a peptide via U, we expect the peptide to be annotated by two modifications (A + rest) and U' base. The U' base modification would be shown by PEAKS at the site of crosslinking (amino acid residue) and A + rest (that again dissociates upon HCD/CID) would again be formally annotated at the N- or the C-terminus of the peptide. In the same way, the ribo-trinucleotides were defined as a combination of crosslinked base and the remaining modification as a sum of the ribo-dinucleotide and "rest".

The raw data were searched with PEAKS 8.0. The PEAKS peptide identification is performed in three steps: de novo peptide analysis (PST generation), database search, and PTM analysis. In total, we allow 34 RNA (ribonucleotide adduct) specific PTM modifications. We chose three methods (Search Strategies—SS1, SS2 and SS3) (Supplementary Fig. 7b) to check for the speed of analysis of XL-peptides.

For all search strategies, and PEAKS algorithms (PEAKS de-novo, PEAKS-DB and PEAKS-PTM), the MMD for monoisotopic precursor and fragment ions was set to 10 ppm and 0.02 Da (Q Exactive data), respectively. The PEAKS search considered cysteine carbamidomethylation as a fixed modification and deamidation (N and Q), methionine oxidation, as well as protein N-terminal acetylation as variable post translational modifications (PTMs or Standard modifications) (Supplementary Fig. 7b). Enzyme specificity was set to trypsin and non-specific cleavage was disabled. The results were limited to peptide spectrum matches harbouring a maximum of two missed cleavages and three variable PTMs per peptide. Finally, the peptides were selected with a FDR cutoff set at 5%.

In the first iteration (SS1) the RNA-PTMs were used only in the last step of the PEAKS workflow (PEAKS-PTM search). This approach was fastest as none of the RNA PTMs are considered for de novo analysis. In SS2 we merely used the stably bound base modifications (A', G', C' U') for de novo and PEAKS-DB analysis and all 34 modifications were utilised at the PTM matching stage. SS3 is the computationally most demanding approach by accounting for all possible RNA modifications at each step of the PEAKS workflow (PEAKS de-novo, PEAKS-DB and PEAKS-PTM). The number of unique peptides and spectra identified by each of the search strategies is summarised for *Drosophila* XL-peptides data in Supplementary Data 4. The peptides identified in each of the search strategies were screened for those containing RNA-PTMs. Further, the

RNA-PTM combinations on each peptide were screened by a custom python script to select only those PTMs, which were composed of a single-nucleobase or complete single/di/trinucleotides (Supplementary Fig. 7c). Combinations of PTMs that represent biochemically impossible combinations (like "rest" only or "rest + A" and "rest + U") were removed. Subsequently, each of the above searches were also manually curated to remove low-quality spectra that might lead to identification of false positives. This removal was based on low spectral scores and incomplete spectral annotation. SS1 was the fastest strategy, however SS3 strategy identified the most spectra. Hence, we chose SS3 strategy for XL-peptide analysis.

In case of *Drosophila*, six biological replicates were utilised for crosslinked peptide analysis, whereas for human samples three biological replicates were considered. Search strategy 3 was employed for the final crosslinked peptide analysis and a peptide FDR cutoff of 5% was applied. A custom python script was used to map only the crosslinked peptides to annotated protein sequences. The spectra of peptides mapping known PDB structures were manually annotated to visualise the amino acid in proximity to RNA (Supplementary Data 6). Spectra of many peptides mapping to novel domains and IDRs were also manually annotated (Supplementary Data 7). Poorly characterised spectra identifying novel domains were removed.

In the manual verification, the first step of quality check is the coverage of amino acids annotated in the spectrum. The second step is confirmation of the proposed RNA-PTM combination. The chosen spectra can then be annotated. For example, in Supplementary Fig. 7f we show first a schematic and below it an example of tryptic peptide-crosslink harbouring a single ribo-mononucleotide adduct. The higher *m/z* range region contains the neutral loss area that is made up by molecular ions, which have lost a phosphate group, a ribose group or both (the ribose + phosphate moiety) from the peptide-ribonucleotide conjugate. In case of a crosslinked ribo-mononucleotide, this should result in the sequential loss of phosphate and ribose, respectively (Supplementary Fig. 7f). In general, no marker ions arising from the protonated RNA base are observed as the base remains covalently attached to the amino acid.

In the case of a crosslinked ribo-dinucleotide, we can observe neutral loss peaks derived by the loss of phosphate, ribose and the entire second ribo-mononucleotide moiety (that is not directly covalently bound to the amino acid residue). More importantly, singly charged (protonated) RNA nucleotides and bases can now also be detected in the lower *m/z* range of the spectrum (also called marker ions). We observed protonated ions of the nucleobases of A' (136), G' (152), C' (112) and U' (113). This feature is somehow analogous to the immonium ions observed in the lower *m/z* range for certain protonated amino acids (e.g., tyrosine, tryptophan immonium ions). The signals for nucleobase marker ions are in general the strongest in the spectrum of a peptide crosslinked to a ribo-dinucleotide (hence called the base peak of the spectrum). The marker ion for the adenine base (*m/z* 136) conflicts with the immonium ion for tyrosine, but the relative intensity is different. Owing to lower gas phase stability of ribo-dinucleotide adducts during CID/HCD, RNA marker ions will often constitute the base peak of the spectrum. Protonated mono-nucleotides from A (330/312 a.m.u) and C (306 a.m.u.) and U (307 a.m.u.) (Supplementary Data 5–7 and Fig. 4c) are frequently observed.

In the case of the rarely observed ribo-trinucleotide-peptide heteroconjugates (when using CAPRI), peaks similar to dinucleotide crosslinks are detected, however, it becomes more complicated to annotate the neutral loss region.

Advantages of using PEAKS software for crosslinked peptide identification are as follows: (i) No prefiltering of peak lists is required. (ii) The interactive environment of PEAKS enables easy visualisation of each of the unique crosslinks (Supplementary Fig. 7d). (iii) Annotation of the amino acid involved in the crosslink in a majority of spectra. (iv) Identification of peptides containing nucleobases only.

**Union of ADJ- and XL-peptides into CAPRI-peptides**. ADJ and XL-peptides tables were simply appended without merging overlapping peptides in order to maintain their distinct information. Custom Python and sqlite3 scripts were used to combine information about peptide coverage and domain information to summarise domain identification and plot protein coverage images.

**Analysis of FA-dom-peptides**. We used the peptides.txt output files from Max-Quant analysis to to identify the peptides enriched in FA crosslinked samples. First, peptides mapping to contaminants were removed and missing intensity values were imputed with the minimum peptide intensity observed. The peptides were selected based on a moderated *t*-test (Benjamini–Hochberg FDR < 0.15) comparing crosslinked to the respective non-crosslinked samples. In spite of a higher FDR cutoff the selected peptides were observed in at least two FA crosslinked replicates. The peptides were subsequently filtered by an average intensity cutoff ( > eightfold). An in-house python script was developed to analyse the overlaps of the FA-dom-peptides with Pfam domain identifications and disordered regions.

**GO term enrichment analysis**. The Gene ontology, InterPro domain and KEGG pathway enrichment analysis for *Drosophila* and human proteins was performed by DAVID on-line tool[79] using all the protein-coding genes as background and Benjamini–Hochberg correction (5% FDR) for multiple testing. The KEGG pathways were visualised via the DAVID tool and KEGG mapper tool[80].

**Analysis of RNA protein complexes based on proximity to RNA**. *Drosophila* protein complex information was extracted from the Compleat tool[81] FA-RBPome, UV-RBPome, ADJ-RBPome to XL-RBPome. A .sif file was generated with RNA as a new molecule interacting with each of the members in the complex with above scores used as the weight of interaction. This new complex network was visualised in Cytoscape 3.0 after applying force-directed layout based on the weight of interaction.

**Analysis of novel domains and orthologs**. The protein sequences were annotated using Pfam domains (release Jan 2016). Pfam definitions from InterPro database were used for defining the extent of domains in UniProt sequences. A list of classical RBDs was constructed by combining information described in the following. (1) Literature information[11], (2) a compiled list of domains possessing the keyword "ribosom", (3) those domains classified as "RNA binding" in their GO annotations and (4) a small manually curated list based on literature. CAPRI-peptides mapping outside this defined list of RBDs were classified to identify novel RBDs. Disordered regions were extracted from MobiDb[82] (by personal communication). DOIPT tool[83] was employed to obtain pairs of orthologs by mapping all *Drosophila* protein-coding genes to human genes by excluding low score hits (score > 1, unless only match score is 1). As a conservative measure, we utilised only those ADJ-peptides, which mapped to unique genes for ortholog analysis.

**Amino acid composition analysis**. Comparison of amino acid composition between two groups of sequences were tested by Fisher's exact test and the *p*-values were corrected for multiple hypothesis testing by Benjamini–Hochberg correction[21]. Comparison of amino acid composition for RNA conjugated amino acids was performed by simply calculating the amino acid percentage composition. Amino acid composition analysis for yps/YBX1 CAPRI-peptides was conducted by using the Composition Profiler online tool[84] considering the complete (Swiss-Prot and TrEMBL) UniProt Knowledgebase as background and using a Bonferroni correction for multiple testing.

**Motif discovery**. In order to avoid the bias which may result from a repeated overlap between XL-peptide and ADJ-peptides sequences, only ADJ-peptides were used for motif discovery. DREME (Discriminative Regular Expression Motif Elicitation) was used to discover short, ungapped motifs that are relatively enriched in the fraction of ADJ-peptides, which are exclusively mapping to disordered domains choosing all the UV-RBPome complete protein peptides (FP-peptides) as background. MEME (Multiple Em for Motif Elicitation) was performed on both unique human and *Drosophila* ADJ-peptides pooled together allowing a maximum motif length of 16 and a maximum of 50 sites per motif. The returned motif sequences and motif sites were screened to identify the *Drosophila* and/or human proteins (UniProt IDs) harbouring these motifs. MEME and DREME analysis was carried out using a local installation of the MEME suite[57].

**Evolutionary analysis of conserved of IDRs**. Ortholog pairs of proteins scaled to same length were visually compared. CAPRI-peptide positions were evaluated based on their relative position in respect to the protein N- and C-termini, as well as their relative distance towards other globular domains present in these proteins. Multiple sequence alignments generated in Clustal Omega/MAFFT were visualised in Jalview. A user defined colouring scheme was utilised to highlight R/G amino acids.

**Protein interactome of large RNA**. We have developed a large RNA (RNA length > 200 nucleotides) interactome capture protocol, which shares similarities to recently published 2C protocol[47]. The procedure uses silica-based purification of large RNA (RNA longer than 200 nucleotides). For large-scale experiments (25 million HEK293 cells equivalent to one 15 cm² plate) the same procedure was performed using (RNAeasy Midi kit Cat No. 75144) with some modifications. For the large-scale interactome capture 25 million UV crosslinked (254 nm, 200 mJ/cm²) and non-crosslinked HEK293 cells (a 15 cm confluent plate) were resuspended in 3.5 ml RLTplus Buffer (Qiagen Cat No. 1053393, probably contains ~5 M guanidine thiocyanate and proprietary detergents) with 35 µl of 14.3 M beta mercaptoethanol. The samples were homogenised with a rotor stator homogeniser at 2000 rpm for 30 s. The lysate was loaded on the midi genomic DNA elimination column (Enzymax EZC222) and spun through at 3000 x *g* for 5 min. An equal volume of 70% ethanol was added to the flowthrough and loaded onto a midi RNA column (RNAeasy Midi kit Cat No. 75144). The flowthrough was collected after centrifugation at 3000 x *g*, 3 min and stored at RT for a second iteration of RNA isolation. The column was washed once with 3.5 ml of RW1 wash buffer (Proprietary composition: contains guanidine salts) and twice with 2.5 ml RPE wash buffer from RNAeasy Midi kit with centrifugation at 3000 x *g*, 3 min. The columns were later dried by additional centrifugation at 3000 x *g* 2 min. The RNA along with the UV crosslinked RNA-protein complexes were eluted by adding 500 µl of pre-heated (75 °C) nuclease-free water to the columns and incubating for 1 min at room temperature. The eluate was collected as flowthrough after centrifugation at 3000 x *g* for 3 min. The column was reused to isolate RNA from the flowthrough fraction (kept aside beforehand). The eluted RNA and protein content was

quantified by using Qubit fluorometric quantification. RNA was also quality controlled by Bioanalyzer pico chip CE. For protein analysis, RNA was degraded by addition of 1/10th volume of 10x RNA digestion buffer (100 mM Tris-HCl pH 7.5, 1.5 M NaCl, 0.5% NP-40) and 1 µl of a cocktail of RNase A and benzonase (1 µl benzonase, 25 U, Novagen + 1 µl RNases A, Thermo Scientific, 10 µg/µl + 300 µl of 1x digestion buffer) and incubated at 37 °C for 12 h. Proteins were subsequently visualised by both silver staining and western blot analysis. For small scale capture experiments (1–10 million cells) the above procedure can be performed employing Qiagen AllPrep DNA/RNA mini kit 80204.

**Antibodies**. We utilised the following primary antibodies directed against *Drosophila* Mle (in-house), glorund (DSHB 5B7_C), Squid (DSHB 2G9-c), Rump (DSHB 5G4), Histone H3 (Active Motif 39763) and Beta Actin (Santa Cruz (I-19) sc-1616) as well as antibodies recognising human DHX9 (Abcam ab183731), HNRNPM1–4 (Santa Cruz sc-20002), EIF-4A1 (Abcam EPR14506/Ab185946), KHDRBS1 (Sigma S9575), BETA ACTIN (Santa Cruz (I-19) sc-1616), histone H3 (Active Motif 39763), histone H4 (Millipore 05–858), OXPHOS Rodent WB Antibody Cocktail (Abcam ab110413- to detect Complex I member NDUF88, Complex II member SDHB, Complex III member UQCRC2, Complex IV member MTCO1 and Complex V member ATP5A1), OXPHOS complex II member SDHA (Invitrogen 459200), TOMM20 (Santa Cruz sc-11415), XRCC5 (Invitrogen MA5–15873), MSH6 (Cell Signalling 5424P), GAPDH (Bethyl A300-641A), HUR (3A2) (Santa Cruz 5261) and FLAG HRP (Sigma A8592). All antibodies were used at a dilution of 1:1000 in 5% fat free milk powder dissolved in 0.3% Tween-20 phosphate buffered saline (Supplementary Figs 19–21).

**PNK assay**. Respective proteins and domains were cloned into pcDNA5-FRT-TO plasmids with addition of a C-terminal 15 kD 3Flag-HBH tag comprising a sequential arrangement of the following epitope-tag sequences: Flag, hexahistidine, in-vivo biotinylation signal peptide, hexahistidine that are derived from the HBH tag[85]. Affinity purification of the tagged proteins was adopted from a protocol described in Maticzka et al.[48]. Proteins in Fig. 6 (SRRT, MSN, EZR, RDX, PEBP1, SAP18, CSNK1D, ACADSB, ALDH6A1, MCAT, RACK1, PPIE) were stably integrated into a single FRT site Flp-In™ T-REx™-293 Cell Line using the product protocol. The proteins were expressed by inducing with 0.1 µg/ml doxycycline. For the rest of the proteins and domains the HEK293 cells were transiently transfected, induced with 0.1 µg/ml doxycycline and harvested in Lysis buffer (1% Triton-X, 0.1% Tween-20, 1x PBS, 0.3 M NaCl, 1x Complete protease inhibitors). The lysate was sonicated for 5 min (Bioruptor sonicator, Hi setting, 30 s ON and 30 s OFF) and further cleared by ultracentrifugation at 20,000xg for 15 min. The affinity purification of the tagged proteins was performed in two steps by first using TALON-Dynabeads (for His tag purification) followed by MyOneC1 Streptavidin Dynabeads (for Biotin tag purification). TALON bead purification starts by incubating them with extracts for 10 min at 4 °C. The beads are washed twice with Lysis buffer and proteins are eluted by using 250 mM imidazole in Lysis buffer. Subsequently, the eluates were incubated with MyOneC1 beads for 30 min and washed sequentially with iCLIP lysis buffer (50 mM Tris-HCl, pH 7.4, 140 mM NaCl, 1% Triton-X, 0.1% SDS, 0.1% DOC, 1 mM EDTA), Denaturing lysis buffer (20 mM Tris-HCl, pH 7.4, 0.5 M LiCl, 1% SDS, 1 mM EDTA), High-salt buffer (HSB) (50 mM Tris-HCl, pH 7.4, 1 M NaCl, 1% Triton-X, 0.1% SDS, 0.5% DOC, 1 mM EDTA) and NDB (50 mM Tris-HCl, pH 7.4, 0.1 M NaCl, 0.1% Tween-20). The beads were resuspended in NDB buffer and treated with 2 µl Turbo DNase I and 10 µl 1:1000 diluted RNase I for 3 min at 37 °C in a thermomixer at 1100 rpm. Next, the beads were washed twice with NDB buffer again. 10% of beads were separated and processed further without radioactive labelling in order to detect proteins by western blot. The crosslinked RNA on the remaining beads was radiolabeled with 0.5 µl of 10 µCi/µl gamma-[³²P]-ATP using T4 Polynucleotide Kinase in 20 µl PNK buffer (20 mM Tris-HCl, pH 7.4, 10 mM MgCl₂, 0.2% Tween-20) for 10 min at 37 °C in a thermomixer set at 900 rpm. The beads were washed twice with NDB buffer and proteins from both radiolabeled and non-labelled tubes were eluted in 1x NuPAGE LDS sample buffer at 90 °C for 5 min in a thermomixer at 1100 rpm. The unlabelled and unlabelled proteins were separated on a SDS gel and transferred separately to nitrocellulose membranes. The proteins were detected by using an anti-Flag-HRP antibody (1:1000 dilution) (Sigma A8592) and the labelled RNA-protein heteroconjugates were visualised by autoradiography.

**DNA oligos used**. Oligos used to clone protein domains:
LARP1(DM15 domain)
For primer: 5′ AGACTTAATTAAGCCACCATGCGTACTGCTTCCATC AGCTCCAG
Rev primer: 5′ AATAGGCGCGCCTTTGGGGTCAATGTCCAAATTTT
UBAP2L (IDR - RG rich)
For primer: 5′ AGACTTAATTAAGCCACCATGGATGGTGGCCAGACG GAATC
Rev primer: 5′ AATAGGCGCGCCCTGGGAGCCTGTAGTACTGCCG
HNRNPU (SPRY-AAA domains)

For primer: 5′ AGACTTAATTAAGCCACCATGGCCAAATCTCCTCAG
CCACCTG

Rev primer: 5′ AATAGGCGCGCCTTTTTGGGCTTCTTCCTTCTGAAGTT

ILF3 (DZF domain)

For primer: 5′ AGACTTAATTAAGCCACCATGATTTTTGTGAATGA
TGACCGCCAT

Rev primer: 5′ AATAGGCGCGCCCCCGTCCTCCTCCATTGGG

GTPBP4 (GTP-binding domain)

For primer: 5′ AGACTTAATTAAGCCACCATGAGAAAAGTCAAATTTAC
TCAACAGAATTACC

Rev primer: 5′ AATAGGCGCGCCAACTTTAATAACACCTTCCTCAG
TCAGG

YLPM1 (IDR)

For primer: 5′ AGACTTAATTAAGCCACCATGAGAGGCAACAGCTCA
TCTTACAGAG

Rev primer: 5′ AATAGGCGCGCCTCCTCTTTCTGGATACTCTCGAATC

PRRC2A (IDR)

For primer: 5′ AGACTTAATTAAGCCACCATGATAACCAAGGGGAAG
CTAGGGG

Rev primer: 5′ AATAGGCGCGCCTTTATCCTGCTGCTGAGCCCTCC

UBAP2L and proteins in Fig. 4e were subcloned from the human ORFeome
V5.1 collection (Open Biosystems).

qPCR oligos for detection of genomic DNA contamination

Drosophila roX2

For primer: 5' AGCTCGGATGGCCATCGA

Rev primer: 5' CGTTACTCTTGCTTGATTTTGC

Drosophila rpl22

For primer: 5' GGCTAGCCCGAAGTTTTCTT

Rev primer: 5' AGCTGATCCCTTCAGTGGAA

Drosophila hxk

For primer: 5' AGTGTGTACCGCTTCCATCC

Rev primer: 5' ATCAGATCGAAGGTGATGCC

Drosophila 18s rRNA

For primer: 5' CTGAGAAACGGCTACCACATC

Rev primer: 5' ACCAGACTTGCCCTCCAAT

Drosophila tubulin

For primer: 5' TGTCGCGTGTGAAACACTTC

Rev primer: 5' AGCAGGCGTTTCCAATCTG

Drosophila actin

For primer: 5' GCGTCGGTCAATTCAATCTT

Rev primer: 5' AAGCTGCAACCTCTTCGTCA

Drosophila Rpl32

For primer: 5' ATGCTAAGCTGTCGCACAAATG

Rev primer: 5' GTTCGATCCGTAACCGATGT

Human 18s rRNA

For primer: 5' CTCAACACGGGAAACCTCAC

Rev primer: 5' CGCTCCACCAACTAAGAACG

Human POLG

For primer: 5' ATCATAGTCGGGGTGCCTGA

Rev primer: 5' ATCATAGTCGGGGTGCCTGA

Human GAPDH

For primer: 5' TAGGGCCCGGCTACTAGCGGT

Rev primer: 5' CGCCAGGCTCAGCCAGTCCC

**Reporting summary**. Further information on research design is available in
the Nature Research Reporting Summary linked to this article.

## Data availability

The proteomics raw data and MaxQuant analysis output have been deposited to the
ProteomeXchange Consortium (http://proteomecentral.proteomexchange.org) via the
PRIDE partner repository with the dataset identifier PXD013338. Additional
experimental data including all PEAKS software analysis, MaxQuant outputs, all protein
profile images and motif analysis are available at https://owncloud.gwdg.de/index.php/s/
aZcFjXXtbQG0wjQ/authenticate; Password: Capri$2018. In source data file we have also
included the source data for nucleic acid analysis in Supplementary Figs 1d, f, g, 5d, 6e. A
reporting summary for this Article is available as a Supplementary Information file. All
other data supporting the findings of this study are available from the corresponding
authors upon reasonable request.

## Code availability

All custom code is available at https://owncloud.gwdg.de/index.php/s/
aZcFjXXtbQG0wjQ/authenticate; Password: Capri$2018.

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

## Acknowledgements

We thank Dr. Vivek Bhardwaj for help in bioinformatic analysis, Thiebault Lequeu, Herbert Holz, Dr. Yilong Zhou, Dr. Tugce Aktas Ilik and Dr. Ibrahim Ilik for help with experiments, A.A. and G.M. group members along with Dr. Joanna Chiang for helpful discussions. We thank PEAKS (Bioinformatics Solutions Inc.) for collaborating to make an alpha version of crosslinked peptide annotation tool. We thank BIOSS, University of Freiburg for providing human ORFeome clones. This study was supported by German Research Foundation (DFG) grants CRC992, CRC1140, CRC746 and Germany's Excellence Strategy (CIBSS–EXC-2189–Project ID 390939984) awarded to A.A.

## Author contributions

A.P., A.A. and G.M. designed the project. A.A. and G.M. supervised together. A.P., F. Richter and G.M. designed and executed CAPRI. G.M. performed Mass Spectrometry,

database searches using MaxQuant. F.Richter, A.P., G.M. planned the RNA-PTM and performed XL-peptide analysis. A.P., F.Ramírez performed in silico analysis with inputs from T.M., F.Richter, G.M. A.P. performed the evolutionary analysis, follow-up and validation experiments. A.P., M.S., G.M., A.A. and F.Richter wrote the manuscript together. All authors read and approved the manuscript

## Additional information

**Competing interests:** The authors declare no competing interests.

