## [Peer Review File · Nature Communications]

Reviewers' comments:

Reviewer #1 (Remarks to the Author):

In this manuscript, Panhale et al characterize RNA-protein interactions in human and Drosophila cells. They provide extensive data sets of global interactomes, identify direct interact sites, define multiple disordered regions as RNA-interaction domains, and validate several of them, and they have looked into evolutionary conservation. The work was performed at a high technical level, including all relevant (negative) controls. My only major comment is that the authors should do more justice to prior work, since not everything they show is as novel as they make it seem. In particular:

1. RNA interactome capture has been performed in Drosophila before – even in vivo (embryos), in contrast to the S2 cell line used here. Technically, the authors refer to the paper (Sysoev et al), however they do not spend a word on it. This should be better exposed in a revised manuscript.
2. Although CAPRI is presented as a novel method, it heavily leans on RBDmap that was developed by others recently. Also here, technically the paper is cited (ref #18), however without indicating the extent of similarity between both protocols. The only difference between the methods is the use of a filtration step (in CAPRI) instead of a second round of poly(A)-capture (in RBDmap). Again the authors should make this clear in the text (and probably methods section) to do justice to this previous work.

Minor comments:

3. In Fig 4d and e the authors visualize RNA-binding proteins within complexes using concentric circles. However it remains unclear what this actually indicates, in particular if it correctly reflects the 'true' configuration placing proteins in the core or the periphery. For instance, how can this be projected on known structures, and what novel insight does it provide? Probably more evidence should be demonstrated beyond just Sf3b1.
4. Fig S1e: Why is rRNA so abundant in a method that is directed at poly(A)-RNA?

Reviewer #2 (Remarks to the Author):

Panhale et al. describe a technique to map RNA-binding domains by identifying cross-linked and 'adjacent peptides' and in some cases protein-RNA interaction sites. The authors study the RNA-protein interactome in *Drosophila* and human representative for metazoan. The study is rather broad comparing and combining different techniques and organisms. The technique favoured here (CAPRI) was previously published by the Hentze laboratory. Unfortunately, the study is missing out in several points and, in its present form, does not meet the requirements of a high impact journal such as *Nature communications*. In fact, the only novelty of this work is the use of formaldehyde as a crosslinker which raises severe doubts for suitability. My points of concern are listed below.

Major concern:

- Most importantly, the analytical concept in the presented method/workflow is not new. The here-termed CAPRI workflow was already published by the Hentze laboratory in Castello et al 2016 *Mol Cell* and later presented in form of a protocol: Castello et al 2017 *Nat protoc*. Direct analysis of crosslinking sites by mass spectrometry has been introduced by the Urlaub laboratory (Kramer et al 2014 *Nat Methods*) and has already been applied in numerous publications of this group.
- The authors do, however, introduce FA crosslinking for the previously developed workflow. Even though FA crosslinking has been introduced in older studies for protein-RNA crosslinking this technique has not consolidated over the last decades. There are several reasons why FA crosslinking is not suited for these experiments. My next points will clarify this.
- FA crosslinking is rather unspecific. Not only primary amines of Lysines and hydroxyl groups (as stated in this manuscript) are reactive towards formaldehyde but also side chains of cysteine, tyrosine, tryptophan, asparagine, glutamine, arginine, histidine. The works by the Kast laboratory describe this in detail (see e.g. Sutherland et al 2008 *J Mass Spectrom* for a detailed review). Therefore, protein crosslinking is very likely and a high number of protein interactions will be captured (not only a 'second layer' but rather a complete interaction network). Consequently it is not surprising that a higher number of proteins will be identified from FA crosslinked sample. However, these results should be evaluated carefully as identified proteins might not directly interact with RNA but rather represent interactions partners in a network (such as a 'pull down').
- FA crosslinking is reversible by heat. See also the works by Kast and coworkers for details. The authors do not discuss this or consider this during their analysis. FA crosslinked samples are first heated at 56°C (which might be acceptable for FA crosslinking), then separated by NuPAGE gel electrophoresis (which usually involves heating at 70°C, details not given in the methods section) and then analysis using a Q exactive mass spectrometer (which requires high source temperatures of at least >200°C, details not given in the methods section). FA crosslinked peptides including protein-RNA interaction sites should therefore not be identified using these experimental conditions.
- The latter two points might explain the observation of 'novel pathways' which have not been reported to be involved in RNA-binding or metabolism before. Particularly concerning are

endocytosis and Valine-Leucine-Isoleucine degradation which should not be included in the RBPome and might represent false-positives.

- The CAPRI workflow includes pre-digestion with Lys-C. Followed by tryptic digestion so-called 'adjacent peptides' are generated. The authors use these peptides to locate RNA-binding domains. However, they do not discuss or consider incomplete digestion. When a protein is digested incompletely or in the case of missing Lys-C cleavage sites this localisation is misleading and most likely reveals false positives. Importantly, FA strongly reacts with and thereby modifies Lysine residues so that these can no longer be cleaved by Lys-C and/or trypsin. As stated above this is a misconception of the workflow and is prone to false positives.
- The identification of RNA-binding sites in the complexes of oxidative phosphorylation (electron transport chain complexes) is most unlikely. RNA-binding has never been discussed for any of these complexes and does not have any biological meaning. For instance, the gamma subunit of the ATP synthase is connecting the head and the base of the ATP synthase. Due to its position in the multi-subunit complex RNA-binding is sterically impossible (in addition to a biological meaning). The same is true for subunit I of Cytochrome c oxidase. This particular subunit is completely embedded in the membrane and surrounded by additional protein subunits.
- This study is missing independent (in vivo) experiments for the validation of the novel RNA-binding proteins and their domains
- The authors used the PEAKS software for assignment of crosslinked amino acids after sequencing of peptides that are still crosslinked to RNA. According to the supplementary data the database search took several days which is not worthwhile. Furthermore, the authors claim in the introduction that currently available software tools for identification of proteins and amino acids that are crosslinked to RNA require manual inspection of the mass spectra. However, the authors also manually evaluated their results when using PEAKS. The prolonged computation time and the fact that mass spectra needed to be manually validated make an advantage of the data evaluation strategy introduced here questionable when compared with existing software tools such as MS fragger or RNPxl.
- Upon manual annotation of their crosslink spectra the authors claim that they identified 'novel' crosslinks in the MS2 spectra where the N-glycosidic bond of the crosslinked RNA moiety has been cleaved. Similar spectra annotation has been performed and published by the Urlaub laboratory in previous studies.
- This study mostly targets Drosophila and in the second part (CAPRI method) compares the results with human HEK cells. The authors use these two species as representatives for metazoan. However, this conclusion is probably over-interpreted and additional species should be selected when discussing the 'metazoan RBPome'.
- The methods section does not allow to reproduce the results. There are too many sections and it is not clear which sections need to be performed for which experiment. For instance, the section 'Interactome capture' is completely unnecessary. Sections on data analysis (e.g. database search and sorting database search results) should be combined and not spread out over several sections. Large parts of the method sections are repetitive (e.g. on-bead digestion).

- The described workflow with different techniques including comparison is rather complicated and time-consuming. I doubt that this workflow in the described form is a technical improvement which will be used by the community.
- The figures are overloaded and not entirely clear. When printed at final size some of the text will not be readable. Figure 4d is not readable at current size and resolution. Furthermore, the authors fail to illustrate in any figure which crosslinks derived from UV and which from FA. This is misleading.
- I wonder whether the authors are aware of the current literature in the field of RNA-binding proteins. Particularly the parts on FA crosslinking and UV crosslinking are incompletely referenced.

General comments:

- The name CAPRI is not a good choice for a new technique as this abbreviation is already used in the structural biology community which is also addressed here. CAPRI stands for “Critical Assessment of PRediction of Interactions” and is a community-wide experiment on the evaluation of protein-protein docking for structure prediction. I suggest a different name which will not be confused in the structural biology community.
- The manuscript is written in a lousy manner. Some parts are very lengthy without providing sufficient information and clarity. The referencing is in large parts incomplete. Some of the results sections are rather a discussion. There are many abbreviations that are introduced, however, these are not used afterwards (even in the same paragraph). Some terms are spelled differently throughout the entire manuscript. See detailed comments below.

Minor comments:

- The term 'cross-linking, cross-linked, ...' is spelled differently throughout the manuscript (cross-link, crosslink, XL, ...).
- I suggest to use a different term than 'rest' for the specific cross-link PTM
- The abbreviation FA is several times introduced (e.g. twice on page 23) but not used and used in the text. This is just an example, there are several other abbreviations that are introduced and afterwards used or not used.
- There are many typos, missing punctuation and inconsistencies.
- What exactly is meant by 'trituration' (p. 24)? I suggest to use the term 'grinding'.
- The abbreviation MeCNI (e.g. p. 27) for ACN is rather unusual.
- The section 'background of crosslinked peptide analysis' in the methods section is entirely redundant. In addition, this section (as well as the introduction and some other sections) is poorly referenced...

In summary, the authors present a method that has been already published and extensively applied in others studies, mainly from the Hentze and Urlaub laboratories. The 'novelty' of introducing formaldehyde as RNA-protein crosslinker and applying the already described workflow from the Hentze group is in fact a clear misconception due to the comments listed above. Regrettably I have to recommend rejection of this manuscript.

Reviewer #3 (Remarks to the Author):

CAPRI enables comparison of evolutionarily conserved RNA interacting regions

by Panhale et al

Panhale et al. present a comprehensive strategy to identify RNA interacting proteins. In a first step, they perform an mRNA interactome capture using both UV and formaldehyde crosslinking. The latter allows them to identify components of RNPs that do not directly contact RNA which is an interesting extension to facilitate the characterisation of RNP complexes. Furthermore, they develop CAPRI which combines two previously known approaches to detect RNA-binding domains (RBDs). They use this to identify a large number of known and novel RBDs in human and *Drosophila* cell culture. One of their discoveries is that they observe a significant fraction of intrinsically disordered regions in contact with RNA, some of which are evolutionarily conserved between human and *Drosophila*.

The data of this study represents a valuable resource that will be useful for RNA biologists. The authors present their findings in a very well-written manuscript with clear figures.

Major comments:

- The authors claim that the FA-RBPome allows to discover a secondary interaction layer. However, they currently do not provide any independent evidence or experimental validations for these additional interactions. E.g. are these proteins from metabolism, endocytosis or DNA replication co-localising with any type of RNA granules?

Also, in order to evaluate the benefit of the FA-RBPome, it would be important to separately analyse the subset which is not present in the UV-RBPome.

- At present, it is not clear whether there really is a synergistic value of CAPRI. It seems that two already known technologies were run at the same time, but I am not sure about the added value of this combination, especially since the authors simply merge both datasets into a combined list of CAPRI peptides.

Minor comments:

- Since evolutionary conservation is only a minor point in this study, I would mention it in the abstract but not in the title.

- Figure 1d: The correlation of intensities used in this figure is not meaningful, as the intensities can be dominated in both cases from high background values. I assume that a comparison of FA-RBPome against total proteome could show a similar correlation. Therefore, enrichment should be used rather than intensities.

- page 6: "The overall coverage of RNA-related proteins was higher in the FA-RBPome compared to the UV-RBPome (Fig. 1f)." A comparison of absolute numbers is not meaningful, given that the detected FA-RBPome is substantially larger than the UV-RBPome. Using fractions instead would allow to evaluate the relative enrichment of each dataset in the different categories. It would also be interesting to show the same analysis separately for the UV-RBPome-overlapping and unique proteins from the FA-RBPome.

- page 7: "However, more than 65% of the identified *Drosophila* proteins in both UV- and FA-RBPomes did not harbour any of these RBDs (Fig. 1g)." This statement does not fit to the values shown in Figure 1g (i.e. 61% for UV-RBPome).

- It would be helpful to report the results with higher precision. E.g. on page 10, "In each of the structures, the amino acid at the site of crosslink was in close proximity to the RNA". How many structures were evaluated?

- In the part on IDRs, the manuscript gets a bit speculative and lengthy. I think this part should be shortened.

- I think Figure 4c is not cited in the manuscript.

NCOMMS-18-19844
Panhale et al
Point by point response to reviewers

Summary

Here is a brief summary of the new experiments added upon revision.

- We have used formaldehyde-based crosslinking to identify RNA binding domains. This adds a new perspective and novelty to the manuscript, for the first time, describing FA based RBDs.
- In an effort to independently validate new RNA binding proteins we used an approach based on a different chemistry (Silica based RNA isolation) to purify all large RNA binding proteins. With this strategy, we successfully validated 5 proteins involved in oxidative phosphorylation, mitochondrial RNA transport and DNA repair.
- Moreover, we also used an orthogonal approach (PNK assay) to verify 9 RNA binding proteins involved in processes like cell signalling and metabolism.
- Based on reviewer 2's suggestion we suggest an alternative title for our method: Protein Domain Isolation by RNA pulldown (ProDIRP). However, our preference would be to retain the original title.

Upon revision, we have added two new figures, Supplementary Fig. 10, Supplementary Fig. 13, Supplementary Table 13 as well as new panels Fig. 4f and Supplementary Fig. 5h. We have also incorporated two new sections on the identification of RBDs by using FA and validation of newly identified RBPs. We have also made substantial improvements to the clarity of the text.

Reviewer #1

In this manuscript, Panhale et al characterize RNA-protein interactions in human and Drosophila cells. They provide extensive data sets of global interactomes, identify direct interact sites, define multiple disordered regions as RNA-interaction domains, and validate several of them, and they have looked into evolutionary conservation. The work was performed at a high technical level, including all relevant (negative) controls.

We thank the reviewer for the supportive comment.

My only major comment is that the authors should do more justice to prior work, since not everything they show is as novel as they make it seem
In particular:

1.1. RNA interactome capture has been performed in Drosophila before – even in vivo (embryos), in contrast to the S2 cell line used here. Technically, the authors refer to the paper (Sysoev et al), however they do not spend a word on it. This should be better exposed in a revised manuscript.

As per reviewer's request we have included more detailed information regarding those reports in our updated manuscript [please see pages 5 and 6].

It is important to note that the two previous pioneering studies were carried out on samples derived from early embryonic stages (0-2 hrs^{1,2} and 4.5-5.5 hrs¹) whereas S2 DRSC cells (employed in our work) were originally isolated from late (20-24 hrs) embryonic stages³. Since both the proteome and RBPome are known to change during development^{1,4} we anticipate dramatic differences between RBPs uncovered by previous studies and our work. Indeed we found this to be the case (see Fig. 1e). Hence, our systematically compiled S2 UV RBPome holds great potential to substantially increase the number of bona fide *Drosophila* RBPs.

We would also like to point out that both of the previous studies did not perform formaldehyde dependent interactome or RBD analyses. Generally, all RNA interactome capture based reports till now arbitrarily consider either lower or higher stringency cut-offs for RBPome definition, thereby ignoring the fact that higher stringency cut-offs might eventually exclude proteins, which bind RNA with lower stoichiometry. In contrast, we have taken a comprehensive approach allowing us to draw a complete "topographical map" of the *Drosophila* RNA protein interaction landscape visualizing both network architecture and domain RNA interaction at single amino acid resolution. For the review process (not included in the manuscript) we have now compared the two previously published interactomes with our stringent (RNA binding domain definition filtered) ADJ-RBPome (Venn diagram, see Revision Fig. 1). In summary, our work nicely overlaps with the previous publications^{1,2}. Notably, the overlap between our ADJ-RBPome and each of the early embryo RBPomes is as large as the overlap between the previously compiled "early fly" RBPomes.

Revision Fig. 1: Venn Diagram illustrating the overlap between the RBPomes from Wessels et al. 2016, Sysoev et al. 2016 and the stringent ADJ-RBPome derived from the CAPRI protocol.

1.2. Although CAPRI is presented as a novel method, it heavily leans on RBDmap that was developed by others recently. Also here, technically the paper is cited (ref #18), however without indicating the extent of similarity between both protocols.

The only difference between the methods is the use of a filtration step (in CAPRI) instead of a second round of poly(A)-capture (in RBDmap). Again the authors should make this clear in the text (and probably methods section) to do justice to this previous work.

We have now revised the manuscript text to address this concern raised by the reviewer [see pages 8 and 13]. Additionally, Supplementary Fig. 8a outlines the similarities and differences between our and previously published protocols. At the same time we would like to point out that there are some significant differences between RBDmap and CAPRI that are related to the fact that CAPRI and RBDmap were developed independently and that discussed in the following.

Fundamentally, CAPRI enables simultaneous RBPome, adjacent and crosslinked peptide analysis starting from the very same sample (input). A method similar to RBDmap will require three consecutive oligo-dT purification experiments. A first round, for isolation of RNA bound proteins, a second one for purifying Lys-C digested RNA bound peptides and a third one for isolating tryptic RNA crosslinked peptides. Each round of oligo-dT pulldown will require fresh expensive beads, will include many washes, and will likely lead to sample loss. This loss can be reduced by utilizing a single step oligo-dT enrichment combined with FASP. Besides and contrary to the one pot solution offered by CAPRI, other traditional methods of crosslinked RNA-peptide hetero-conjugate purification, like TiO₂ bead assisted enrichment^{5,6}, also require the combination of several consecutive purification steps. In the case of CAPRI, we introduce "RNase-FASP", which allows sample clean-up with the chaotrope urea (8 M). The latter is not compatible with oligo-dT mediated purification (destabilization of DNA:RNA helix) and is more likely to eliminate non-specific interactions (between RNA and non-crosslinked peptides). At the same time the FASP MWCO filter removes larger biomolecules (undigested nucleic acids, polysaccharides etc.) frequently interfering with nanoLC-MS performance. Thus, CAPRI grants high stringency and efficient sample clean-up, whereas at the same time minimizing sample handling steps.

Minor comments:

1.3. In Fig 4d and e the authors visualize RNA-binding proteins within complexes using concentric circles. However it remains unclear what this actually indicates, in particular if it correctly reflects the 'true' configuration placing proteins in the core or the periphery. For instance, how can this be projected on known structures, and what novel insight does it provide? Probably more evidence should be demonstrated beyond just Sf3b1.

We apologize for the misunderstanding. The concentric circles indicate the confidence with which we can predict a protein to be (directly) bound to RNA. It also reflects the resolution of defining a RNA binding domain. Although it need not necessarily reflect the hierarchy of actual complex interaction, it reflects the proximity of a protein to the RNA interface. The likelihood of a protein to form UV crosslinks in a complex is highly dependent on the proximity and stability of its interaction with RNA. We have also included other complexes like TREX complex, and cytoplasmic and mitochondrial ribosomes in our analysis (Supplementary Fig. 11).

1.4. Fig S1e: Why is rRNA so abundant in a method that is directed at poly(A)-RNA?

The Reviewer has correctly identified a minor limitation of the current oligo-dT based isolation of RNA binding proteins. Traditionally, one step isolation

by oligo-dT beads leads to 10-20% rRNA contamination (NEB manual). This could be due to rRNA continues to be associated to mRNA it is associated with or it short A-rich sequences which can be annealed to oligo-dT at 4°C. This the reason why usual mRNA-seq experiments use two step isolation wherein the RNA is eluted (~50°C for 3min) from the beads and rebound to specifically enrich mRNA. In Baltz et al. 2012⁷, the authors tried to remove rRNA by extensive washes at room temperature in a one step purification attempt. With this approach they got 2-8 % rRNA in non-irradiated cells, however they still obtained 38% rRNA in the crosslinked samples (Fig. 1e in Baltz et al. 2012⁷). This is most probably due to UV irradiation induced crosslinks formed between rRNA and mRNA. A similar high rRNA content in UV irradiated samples can be seen in Supplementary Fig. S1d. of Castello et al. 2016⁸.

In our study, we have purified the whole protein UV and FA-RBPomes at 4°C, which partially preserves the cross-hybridization of rRNA and oligo-dT. However, we see a reduced rRNA fraction in noUV elution as compared to “plus” UV fractions in CAPRI experiments (Supplementary Fig. 5b and Supplementary Fig. 6d). This is consistent with previous observations (e.g. Fig. 1e in Baltz et al. 2012⁷).

Reviewer #2

Panhale et al. describe a technique to map RNA-binding domains by identifying crosslinked and 'adjacent peptides' and in some cases protein-RNA interaction sites. The authors study the RNA-protein interactome in *Drosophila* and human representative for metazoan. The study is rather broad comparing and combining different techniques and organisms. The technique favoured here (CAPRI) was previously published by the Hentze laboratory. Unfortunately, the study is missing out in several points and, in its present form, does not meet the requirements of a high impact journal such as *Nature communications*. In fact, the only novelty of this work is the use of formaldehyde as a crosslinker which raises severe doubts for suitability.

My points of concern are listed below.

We would like to thank the reviewer for the very detailed comments and suggestions that we have taken very seriously. We tried to address all the concerns and criticism and truly believe that the revised manuscript has now substantially improved. After clarifying a couple of misunderstandings in the revised manuscript (detailed below), we hope that we can convince reviewer 2 to share our view that CAPRI is definitely distinct from the work carried out by Castello et al., 2016.

Major concern:

2.1. Most importantly, the analytical concept in the presented method/workflow is not new. The here-termed CAPRI workflow was already published by the Hentze laboratory in Castello et al 2016 *Mol Cell* and later presented in form of a protocol: Castello et al 2017 *Nat protoc*. Direct analysis of crosslinking sites by mass spectrometry has been introduced by the Urlaub laboratory (Kramer et al 2014 *Nat Methods*) and has already been applied in numerous publications of this group.

We agree that "adjacent" (ADJ) and crosslinked (XL) peptide analysis work has been carried out by the laboratories of Hentze and Krijgsveld as well as Urlaub and Kohlbacher (e.g. Castello et al 2016⁸, Kramer et al. 2014⁶). However, both protocols depend on separate enrichment steps for identifying ADJ and XL peptides. So far, studies aiming to define RNA-interacting domains either exclusively relied on custom workflows to identify crosslinked RNA-peptide hetero-conjugates or utilizing "adjacent" peptide enrichment. Here, with our independently developed protocol we address this problem by enabling the simultaneous ADJ- and XL- peptide identification in a one pot approach, harnessing simple and efficient size-based selection properties of FASP filters that represent a mainstay in common proteomics sample preparation. The latter allow high stringency washes at 8M urea (expected to remove peptides non-specifically sticking to RNA) and minimize sample loss (low non-specific adsorption in presence of 8 M urea).

Furthermore, utilizing the commercially available software PEAKS we developed a de novo sequence tag assisted crosslinked peptide analysis approach (employing a defined number of PTMs). The latter first attempts to

(partially) interpret all MS/MS spectra for peptide sequence assignment prior to the execution of the database search engine (PEAKS-DB) step. In contrast, the RNPxl⁶ and MSFragger⁹ pipelines rely on classical database search algorithms that are (in their first step) entirely based on matching experimental MS1 peptide ion masses with corresponding theoretical masses of peptides (generated in silico from a protein sequence database). In this context the number of theoretical peptide ion masses exponentially increases with the number of variable (RNA) PTMs (RNPxl). In our de novo assisted approach PEAKS first assigns MS/MS peptide spectrum matches taking into account the “base” (nucleobases) remaining covalently attached to the internal peptide ion fragments. As described in detail in the supplementary information this is crucial because the crosslinked RNA is metastable and frequently decomposes during the process of MS/MS spectrum generation (in the gas phase of the mass spectrometer). Moreover, due to our unique RNA-PTM definitions, we not only identified new types of crosslinks, but also validated many crosslinks in published RNA-protein complex PDB structures. Even though our strategy is different, we would like to stress here that we do not claim (in the manuscript) that our approach is generally superior to RNPxl or MSFragger because this would require an extensive and systematic comparison of these tools, which is not the scope of this manuscript. Nevertheless, we feel that the XL peptide spectra are more intuitive to be validated by general MS users as the internal RNA-peptide hetero-conjugate ion series are automatically annotated by PEAKS. Thus manual annotation is currently required for assigning C-H⁺, A-H⁺ and G-H⁺ (protonated base) ions in the low m/z region as well as those peaks that are generated by the dissociation of the RNA PTM (partial neutral losses). This concerns for instance the “stepwise” decomposition of the covalently attached RNA molecule at the level of the unfragmented peptide parent ion. Aside from this, our data analysis pipeline also makes use of an in-house script that removes the very low amount (1-4%) of PEAKS MS2 spectra annotations, which are obviously false positives because they contain “wrong” (chemically nonsense) combinations of either “base” and/or “rest” (for details please see supplementary methods).

Overall, we have not only independently validated previously published methodologies that have so far been exclusively employed by two highly specialized laboratories but succeeded in democratizing RBD mapping for the benefit of the RNA biology community. Therefore, we strongly believe that we provide a novel and a very useful approach to study RNA-protein interactions which will be much appreciated by the community.

2.2. The authors do, however, introduce FA crosslinking for the previously developed workflow. Even though FA crosslinking has been introduced in older studies for protein-RNA crosslinking this technique has not consolidated over the last decades. There are several reasons why FA crosslinking is not suited for these experiments. My next points will clarify this.

FA crosslinking is rather unspecific. Not only primary amines of Lysines and hydroxyl groups (as stated in this manuscript) are reactive towards formaldehyde but also side chains of cysteine, tyrosine, tryptophan, asparagine, glutamine, arginine, histidine. The works by the Kast laboratory describe this in detail (see e.g. Sutherland et al 2008 J Mass Spectrom for a detailed review). Therefore, protein crosslinking is very likely and a high number of protein interactions will be

captured (not only a 'second layer' but rather a complete interaction network). Consequently it is not surprising that a higher number of proteins will be identified from FA crosslinked sample. However, these results should be evaluated carefully as identified proteins might not directly interact with RNA but rather represent interactions partners in a network (such as a 'pull down').

We thank the reviewer for pointing out that the "zero length" (short distance: 2.3 - 2.8 Å) crosslinking agent FA can undergo diverse chemical reactions. However, FA's reputation of being rather unspecific is not deserved, because otherwise any established technology taking advantage of FA to study chromatin/DNA-protein interactions (e.g. ChIP, Bio-TAP-XL¹⁰, PiCh¹¹, RIME¹², ChIP-SICAP¹³, and CAPTURE¹⁴) would in its extreme case randomly pull-down the entire nuclear proteome, which is clearly not the case. Of note, in the chromatin field it is well established that lysines constitute the predominant sites of methylene bridge formation between histones and amino- or imino-groups of nucleobases¹⁵. In addition, in the case of ChIP assays it is well known that only a combination of FA and other (long range) homobifunctional crosslinkers (e.g. EGS, DSG) can efficiently capture indirect DNA-proximal protein-DNA interactions. Moreover, the work of Kast and other colleagues (reviewed in the publication suggested by reviewer 2: Sutherland, B.W. et al 2008¹⁶ clearly states that under the FA crosslinking conditions employed in our manuscript (relatively short incubation time of 10 min, FA concentration = 0.1% (<=1%)), "the major reactive sites ... are in fact largely limited to lysine, tryptophan side chains, and the amino termini" of proteins (driven through the immonium cations). In vivo, the lowered effective FA concentration inside living cells and the structural constraints dictated by 3D protein structure itself as well as by formation of multiprotein complex assemblies would further limit the chemical complexity of the crosslinking reactions (reviewed in Hoffman, E.A. et al., 2015¹⁷). We would also like to point out that we have always utilized fresh ampules of monomeric methanol-free formaldehyde (Thermo Scientific Pierce™ #28906) in our experiments, further omitting diverse FA chemical reactions enhanced by the presence of polymerised (condensated) FA commonly used in immunohistochemistry or formalin fixation of tissues.

Besides, the combination of FA crosslinking with AP-MS ("QTAX") pioneered by the labs of P. Kaiser and L. Huang^{18,19} has also been successfully applied to the interrogation of protein complexes by us and others²⁰⁻²².

Furthermore, in the following explanation we would like to outline why the use of FA *in combination with* UV will be beneficial for accurate mapping of protein RNA complexes. The first attempts applying FA crosslinking with the aim to study the topology of ribonucleoprotein complexes date back to the 1970s as described by the pioneering studies of Brimacombe and colleagues on the *E. coli* ribosome²³. Nevertheless and opposite to the chromatin field, FA mediated RNA-protein crosslinking has for a long time led a niche existence, which can mainly be attributed to the fact that the crosslinks are reversible (especially at ambient or higher temperature) and that UV crosslinking works orders of magnitude better for (single-stranded) RNA than DNA. But 254 nm UV crosslinking is also known to stabilize adventitious binding events and is subject to biases (pyrimidines are more photoactive than purines; double stranded RNA is way less photoactive; reactivity among amino acid side chains differs) that could most possibly be counterbalanced by the properties of FA^{24,25}. Notably, even though it

is commonly accepted that the specificity of UV crosslinking (concerning single nucleotide resolution) is initially mediated by the process of photoactivation of nucleobases, the detailed chemical structures of the amino acid side chain-ribonucleobase crosslinked molecules are (at least to our knowledge) vastly unknown until now.

Hence, particularly in combination with antisense oligo mediated purification of RNP complexes, the current literature shows nice methodological examples demonstrating the specificity of FA crosslinking for detecting short range RNA-protein interactions: RIPIT-Seq²⁶, CHIRP²⁷, CHIRP-MS²⁸ and CHART²⁹ protocols. Consequently, by using dCHIRP and in close collaboration with the Howard Chang lab (employing 3% FA for 30 min crosslinking) we were able to show that roX1 architecture contains distinct modules interacting with the protein members of the complex³⁰. Importantly, we were also able to verify these results by orthogonal (UV) CLIP-seq data.

Having said that FA also “freezes” protein-protein interactions we have been careful and have thus categorized protein constituents of the FA-RBPome as either secondary or direct but “lower confidence” RNA interactors as compared to the UV-RBPome, ADJ-RBPome as well as the XL-RBPome. Nonetheless, ~85% of the UV-RBPome is recaptured in the FA-RBPome (Supplementary Fig. 2d) and the proteins identified in both the UV- and the FA-RBPomes show indeed a very good correlation of raw intensities (Fig. 1d, Revision Fig. 3 a,b).

For Reviewer 3 we compared those *Drosophila* FA-RBPome specific proteins (1037 proteins) which possess a human ortholog (1009 proteins) to known experimental (UV) or bioinformatically available data. Interestingly, of the 1009 FA-RBPome specific *Drosophila* proteins with human orthologs, 725 have either already been shown to bind RNA (recent UV RBPome reports) or are annotated as RNA related proteins (Interpro RBDs, “RNA” in GO terms). The remaining 284 FA-specific proteins are novel and most likely reflect indirect “piggyback” binders (Revision Fig. 2).

In summary, all these results clearly indicate that FA can definitely capture direct RNA binders.

Revision Fig 2: Clustered heatmap illustrating the presence (dark purple) or absence of FA-RBPome specific proteins (bottom row) in other RNA-related protein datasets. Legend (from top to bottom): proteins harboring known Interpro RBDs, proteins containing "RNA" in their GO "Molecular Functions" (MF) or GO "Biological Processes" (BP), proteins present in other *Drosophila* embryo UV RBPomes, *Drosophila* proteins possessing human orthologs from Mullari et al., orthologous proteins from the recently published OOPS³¹ and XRNAX³² methods and orthologs from the comprehensive Hentze lab review³³.

2.3. The latter two points might explain the observation of 'novel pathways' which have not been reported to be involved in RNA-binding or metabolism before. Particularly concerning are endocytosis and Valine-Leucine-Isoleucine degradation which should not be included in the RBPome and might represent false-positives.

Generally, there is always the possibility of false discovery in large scale datasets and this is true both for the previously published UV-RBPomes^{25,33} and the FA-RBPome presented here. Nonetheless, we used a very careful wording and described the proteins identified in the FA interactome to have lower confidence for being engaged in direct RNA interactions. However, we would like to stress that proteins implicated in valine-leucine-isoleucine degradation or endocytosis were also found in our UV-RBPome and are not FA specific.

We are surprised about the criticism regarding metabolic enzymes. Metabolic pathways have been reported in other RBPomes (particularly by the extensive work performed in the Hentze laboratory)³³. For both biological (cell type, cell state, tissue specificity) and technical (undersampling problem of proteomic methods) reasons we cannot rule out that metabolic enzymes crosstalk with RNA. Of note, we have validated candidate RNA proteins belonging to metabolic pathways among them enzymes involved in valine-leucine-isoleucine

degradation (Fig. 4f and Supplementary Fig. 4b, Supplementary Fig. 13e). For details please see our reply to your question 2.6.

Just like the reviewer we were also puzzled by the presence of proteins involved in the process of endocytosis in our RBPomes. There are various reports describing the association of RNA/polysomes with membrane vesicles and the process of endocytosis (reviewed in Jansen, R.P. et al. 2014³⁴). Interestingly, it has also been recently shown that RNA granules associate with endosomes along the axons of retinal ganglion cells³⁵) because mRNA and translating ribosomes associate with late Rab7a endosomes in these axons. Strikingly, it has been demonstrated that *Drosophila* S2 cells possess the ability to take up RNA by receptor-mediated endocytosis³⁶. A similar pathway was described for exogenous mRNA uptake by a diversity (HEK, HeLa, dendritic cells, mouse fibroblasts) of mammalian cells³⁷. Importantly, we were able to experimentally verify proteins associated with the endocytosis pathway (Supplementary Fig. 4f).

Admittedly, similar to all RBPome resource papers, substantial orthogonal follow-up studies (UV-CLIP etc.) will be required that are out of the scope of this manuscript.

2.4. FA crosslinking is reversible by heat. See also the works by Kast and coworkers for details. The authors do not discuss this or consider this during their analysis. FA crosslinked samples are first heated at 56°C (which might be acceptable for FA crosslinking), then separated by NuPAGE gel electrophoresis (which usually involves heating at 70°C, details not given in the methods section) and then analysis using a Q exactive mass spectrometer (which requires high source temperatures of at least >200°C, details not given in the methods section). FA crosslinked peptides including protein-RNA interaction sites should therefore not be identified using these experimental conditions.

We apologize for this misunderstanding and completely agree with the reviewer that FA crosslinks are reversible and heat-sensitive. In the submitted manuscript we did not present any FA based RBD analysis, which has already been carried out in vitro (RCAP method) in a case study concerning the hepatitis C virus RNA-dependent RNA polymerase (reviewed in Vaughan, R. et al., 2012³⁸). Therefore, RNA-peptide hetero-conjugates stabilized via methylene bridges were not introduced into the mass spectrometer at any given time. Recently, the half life of FA protein-DNA crosslinks was determined to be around 179 h at 4°C³⁹. Therefore, our GeLC-MS catalogued whole protein FA-RBPome was captured on oligo-dT beads never exceeding 4°C. Once eluted from the beads (subsequent to high stringency wash steps) the eluates were treated with RNases at 37°C for 1 hr and later at 52°C for an additional 30 min (causing FA decrosslinking as well). The proteins were then precipitated using TCA. Remaining crosslinks were further de-crosslinked during 10 min denaturation at 95°C in 1X LDS (NuPAGE loading buffer).

During revision we have tested the possibility of identifying RNA binding domains by using FA crosslinking (*Drosophila* S2 cells) in a small scale study as outlined in the following sections and included them in the revised manuscript (Supplementary Figure 7). Briefly, our data show that FA-RBD mapping is in principle possible but would need further experimental improvements in order to

increase the depth of analysis. The following new section was thus added to the revised manuscript [Page 13]:

“Identification of RNA binding domains via FA mediated crosslinking

*FA (formaldehyde) is a small molecule “zero-length” crosslinker with the capacity to quickly diffuse through living cells thereby freezing biomolecular interactions and preventing intercompartment mixing⁴⁰. Like UV irradiation, FA crosslinking is also used to study RNA protein interactions via RCAP⁴¹, RIPiT-seq²⁶ or CHIRP-MS²⁸. However a large scale study testing its applicability to compile comprehensive RBPomes or to map RBDs has not been performed until now, albeit FA and UV crosslinking can be considered as complementary approaches. While UV is biased towards preferential crosslinking of pyrimidines in single stranded RNA (reviewed in Moore, K.S. and ‘t Hoen, P.A.C., 2019⁴²), FA favours reactions with purines and is also able to target RNA double helices⁴¹. Moreover, it is believed that UV can even photoactivate ribonucleobases buried in the core of protein structures. To the contrary, FA mainly freezes molecular interactions taking place at surface exposed residues, at least under the conditions employed for studying RNA-protein interactions. Nevertheless, FA has some limitations for studying RNA protein interactions, namely the additional formation of intra- and inter-protein crosslinks as well as the metastable nature of the covalent crosslink bonds. Hence, we asked if we could capture the FA crosslinked peptides by performing on-bead proteolysis of oligo-dT immobilized FA crosslinked RBPs with Lys-C and trypsin. The RNA was eluted and RNA-peptide bonds were de-crosslinked by heat after which RNA was degraded. The peptides were further digested with trypsin and analysed with mass spectrometry (Supplementary Fig. 10a,b, see methods for details). From three biological replicates, a total of 554 peptides (FA-dom-peptides) mapping to 162 proteins (FA-dom-RBPome) using an FDR of 0.15. Each peptide was detected in at least two biological replicates with more than 8 fold higher intensity in the crosslinked samples (Supplementary Fig. 10c). These proteins were also enriched for classical RBPs (Supplementary Fig. 10d) and overlapped significantly with other *Drosophila* RBPomes (Supplementary Fig. 10e,f). Of the 554 FA-dom-peptides, 225 (40.6%) peptides mapped to classical RNA binding domains (Fig. 2g). We also observed that the majority (57.4%) of the FA-dom-peptides overlapped with ADJ-peptides and about a third (36.9%) of the UV-XL-peptides overlapped with the FA-dom-peptides. Some examples of these overlaps can be seen in Supplementary Fig. 6a, 9c,d,e. As a proof of principle, these overlaps clearly suggest that FA-dom-peptides can also be used to identify RNA binding domains. As the number of peptides uncovered by CAPRI was higher than in the FA-dom-RBPome, we opted to focus on CAPRI peptides for all subsequent analyses“.*

Supplementary Figure 10

Supplementary Fig. 10: FA domain peptides analysis.

- a) Simplified schematic representation of RNA binding domain analysis by using formaldehyde. Briefly, 0.1% formaldehyde crosslinked RNA binding proteins were captured by using oligo-dT beads. The proteins were partially digested by using Lys-C and trypsin. After thorough washing RNA along with the crosslinked peptides were eluted by heat. The RNA peptide crosslinks were reversed by heating and later the nucleic acids were degraded by nucleases. The peptides were further digested by trypsin and cleaned-up by SP3 purification⁴³.

- b) RNA profiles from Drosophila cells analysed by Bioanalyzer and compared to input RNA (equal amount of RNA was loaded in all lanes).
- c) Clustered heatmap of peptide raw intensities (log2) measured in three biological replicates. The selected FA-dom-peptides are depicted in the last lane with maximum intensity.
- d) Enriched Gene Ontology terms (Molecular Functions) from FA-dom peptide protein interactome (FA-dom-RBPome) in Drosophila.
- e) Venn diagram representing the overlap of proteins identified in ADJ-RBPome, UV-XL-RBPome and FA-dom-RBPome.
- f) Venn diagram showing the overlap of FA-dom-RBPome with the full protein interactomes UV- and FA-RBPomes.
- g) Clustered heatmap representing the distribution of FA-dom-peptides overlapping with Pfam, RBD-Pfam and IDR. Different categories of peptides are annotated on the left of each panel.
- h) Bar graph representing the overlaps of XL-peptides with or without extension on FA-dom-peptides (left). Bar graph showing overlaps of FA-dom-peptides with or without extension on ADJ-peptides (right).

Finally, the description of this workflow was added to the methods section of the revised manuscript [Page 29].

“FA-dom-peptides isolation protocol

FA crosslinked RBPs were purified from 4.4×10^6 S2 DRSC cells with oligo-dT beads as described before at 4 °C. Once the RNA had been captured on beads, the proteins were digested. To pursue this the beads were transferred to a 1.5 ml microcentrifuge tube and washed with 1 ml OBD buffer (50 mM Tris-Cl pH 7.8, 200 mM LiCl, 1mM EDTA). The on-bead digestion was performed at 25 °C for 1 hr in 400 µl of OBD buffer (rotating at 4 rpm) by addition of 1 µg Lys-C and 1 µg of trypsin. After 1 hr the supernatant was removed and replaced with 400 µl of OBD buffer for another round of digestion with 1µg trypsin for 2 hrs. Next, the beads were separated and the supernatant containing released peptides was removed. The beads were transferred to a 5 ml protein low binding tube and washed once with 4 ml OLB, once with 4 ml WB2, 3 times with 4 ml WB3 and finally with prechilled 1 ml 200 mM ammonium bicarbonate (ABC) buffer for 2 min each. The RNA was eluted with 200 µl of 50 mM ABC buffer at 56 °C for 3 min and again with 100 µl of 50 mM ABC buffer at 56 °C for 3 min. The eluates were cleared of remnant beads by using a magnet and spinning at 2000xg for 5 min. The RNA peptide bonds were de-crosslinked by incubating the eluates at 65 °C, 1100 rpm for 90 min. We took an aliquot of RNA for Bioanalyzer capillary electrophoresis. The remaining RNA was degraded with a mixture of 0.5 µl of Benzonase, 0.5 µl of RNase A and 0.5 µl of RNase I for 90 mins at 37 °C at 1100 rpm and for another 30 min at 52 °C at 1100 rpm. The peptides were further digested with 0.5 µg trypsin and cleaned-up for LC-MS analysis by SP3 purification⁴³.”

2.5. The CAPRI workflow includes pre-digestion with Lys-C. Followed by tryptic digestion so-called 'adjacent peptides' are generated. The authors use these peptides to locate RNA-binding domains. However, they do not discuss or consider incomplete digestion. When a protein is digested incompletely or in the case of missing Lys-C cleavage sites this localisation is misleading and most likely reveals false positives. Importantly, FA strongly reacts with and thereby modifies Lysine residues so that these can no longer be cleaved by Lys-C and/or trypsin. As stated above this is a misconception of the workflow and is prone to false positives.

For UV crosslinked CAPRI peptides we have performed experiments to verify missed cleavages. We find a 12% missed cleavage rate (for Drosophila

proteins) in on-bead digested supernatant peptides (separated from the captured RNA-polypeptide conjugates) as well as the Lys-C digested flow through peptides obtained by the RNA-FASP step (Supplementary Fig. 5h). This number is very close to the Lys-C missed cleavage frequency determined for the *E. coli* proteome⁴⁴. Nevertheless, one has to keep in mind that missed cleavage frequency might be subject to organismal bias as it is known for Promega trypsin, representing one of the most frequently used reagents in bottom-up proteomics (<https://www.promegaconnections.com/enhanced-protein-mass-spectrometry-analysis-with-trypsinlys-c-mix/#more-18660>). In the context of UV crosslinked RNA-protein complexes, missed cleavage frequency could be enhanced by both steric hindrance (protection of peptide Lys-C sites by RNA; interaction of lysine side chain with negatively charged phosphate backbone) and the higher frequency of lysines (being present) in RNA binding regions. In case of the latter, for Lys-C it has been shown that any additional lysine residue N- or C-terminal to the cleavage site in a window from -7 to +7 slightly impairs Lys-C cleavage efficiency⁴⁵.

In our data analysis pipeline, for tryptic adjacent peptides containing missed cleavages we simply extend the peptides *in silico* to the next "K" site (Lys-C cleavage site). This of course also includes tryptic peptides harboring an internal "K".

Supplementary Fig. 5h: Missed cleavage observed in Lys-C cleaved peptides released on FASP filters and collected as flow through.

Furthermore, we have corrected and revised the RNA-FASP protocol in the methods section of the paper to precisely describe the adjacent peptide sample preparation, which was also employed for the estimation of missed cleavages. Therefore, we have added the following sentence [Page 28]:

"This was followed by a wash with 100 ul of UB buffer (50mM Tris-HCl, pH 8.0, 4M urea). Lys-C (0.5 ug) was added to the filter in 100 ul UB buffer for another round of digestion and incubated at 25 °C for 6 hrs for Drosophila S2 cells (10 hrs for HEK cells). After digestion the released peptides were washed away with a 100ul 0.5 M NaCl wash. To estimate the efficiency of Lys-C digestion, these peptides were analysed by Mass spectrometry".

This key part of our protocol ensures that the probability of contaminating the samples with peptides not belonging to an RBD is kept low. We are

convinced that the two-step enrichment regimen for adjacent peptides; namely on-bead Lys-C proteolysis (plus wash steps) followed by RNA-FASP Lys-C digestion (including 8 M urea wash cycles), reduces the amount of unspecific peptides that are not covalently attached to RNA to negligible levels. It seems likely that reviewer 2 raised concerns about false positive identifications of RNA binding proteins in our approach due to our omission of this detailed information in the methods section of the original submission. Again, we are very thankful to the comments of the reviewer for pointing out that the degree of missed cleavages has an impact on the resolution of RBD mapping.

2.6. The identification of RNA-binding sites in the complexes of oxidative phosphorylation (electron transport chain complexes) is most unlikely. RNA-binding has never been discussed for any of these complexes and does not have any biological meaning. For instance, the gamma subunit of the ATP synthase is connecting the head and the base of the ATP synthase. Due to its position in the multi-subunit complex RNA-binding is sterically impossible (in addition to a biological meaning). The same is true for subunit I of Cytochrome c oxidase. This particular subunit is completely embedded in the membrane and surrounded by additional protein subunits.

This study is missing independent (in vivo) experiments for the validation of the novel RNA-binding proteins and their domains.

In order to address both points we have applied two different methodologies to validate the candidate RBPs discovered in the FA-, UV- and ADJ-RBPomes. This is described in a new section "Validation of newly identified RBPs" in the main text and new figures Supplementary Fig. 13 and Fig. 4f reproduced below [Page 15-16].

"Validation of newly identified RBPs"

We validated a selection of proteins representing diverse functional classes using two independent techniques in human cells. First, we developed a technique which can capture all large RNA (>200 nucleotides) in a cell by adopting commercially available silica solid phase extraction (similar to the recently published 2C technique⁴⁶). Briefly, UV crosslinked cells were resuspended in guanidine thiocyanate containing buffer, which denatures non-covalent interactions (Supplementary Fig. 13a, see methods for details). Next, the solution was passed through a genomic DNA elimination column. After adding ethanol to the flowthrough, the solution was passed through a RNA binding column. Following stringent washes, the RNA along with its covalently attached proteins was eluted by nuclease free water. The RNA and proteins profiles were analysed by Bioanalyzer and silver staining, respectively (Supplementary Fig. 13b,c). The technique was validated by performing Western blots with positive controls (classical RBPs: ADAR, DHX9, HNRNPM1, HUR3A2, POLRMT; non-classical RBP: GAPDH) and histone H3 as a negative control (Supplementary Fig. 13d). Using this, we further validated new RBPs, some of which have also been reported in previous interactome capture experiments³³. We verified proteins involved in oxidative phosphorylation (OXPHOS Complex I member ATP Synthase F1 Subunit Alpha (ATP5A1) and Complex II member Succinate Dehydrogenase Complex Flavoprotein Subunit A (SDHA)), DNA repair (DNA mis-match repair protein mutS homolog 6 (MSH6), and X-Ray Repair Cross Complementing 5 (XRCC5)^{33,47}) and a mitochondrial tRNA/protein import protein (Mitochondrial import receptor subunit (TOMM20)) (Supplementary Fig. 13e).

Second, in order to validate proteins with an orthogonal approach, we utilized PNK assays^{33,48}. Briefly, affinity tagged proteins were purified from UV crosslinked and non-crosslinked cells in denaturing conditions. RNA bound to proteins was labelled with radioactive ³²P using the PNK enzyme. The proteins were separated by size (SDS-PAGE) and transferred to a membrane on which the RNA binding activity is identified by autoradiography (detecting radioactivity at the respective molecular size) (Fig. 4f). We first validated the assay using three known RBPs: SRRT, PPIE and RACK1 (the first two carrying RRM and the third carrying WD40 domains). Next, we applied the assay to SAP18, a splicing protein⁴⁹ which until now has not been confirmed to directly bind RNA (Fig 4f). Third, we confirmed two proteins involved in cellular signaling as new RBPs, namely PEBP1/RKIP and CSNKD1. PEBP1 (Phosphatidylethanolamine Binding Protein 1) is a well established tumour suppressor protein involved in a multitude of signaling pathways such as Raf/MEK/ERK, NFkB, PI3K/Akt/mTOR, p38, Notch and Wnt⁵⁰. Notably, for PEBP1 the radioactive smear observed in the MW range above the (expected) protein size is specific to radiolabeled RNA as it is sensitive to high amounts of RNase treatment (Fig 4f). CSNKD1 (Casein Kinase 1 Delta) represents another protein, engaged in regulation of multiple pathways including Wnt signaling, DNA repair and circadian rhythms⁵¹. Next, we additionally validated the cytoskeletal ERM family proteins ezrin (EZR), radixin (RDX) and moesin (MSN), which were predicted to interact with RNA in previous RBPomes^{8,33}.

Interestingly, we were also able to verify RNA binding of three metabolic enzymes. Two of them, ALDH6A1 (Aldehyde dehydrogenase 6 family, member A1) and MCAT (Malonyl-CoA-Acyl Carrier Protein Transacylase) are newly identified RBPs with RNA binding regions defined by our ADJ-RBPomes (and also predicted in previous RBPomes³³). Interestingly, the methylmalonate semialdehyde dehydrogenase ALDH6A1 plays a role in both the valine, leucine, isoleucine degradation (Supplementary Fig. 4b) and pyrimidine catabolic pathways, while MCAT is involved in mitochondrial fatty acid beta oxidation. The short/branched chain acyl-CoA dehydrogenase CG3902 was predicted to bind RNA based on our FA-RBPome analyses. It is a member of both the fatty acid and valine, leucine, isoleucine degradation pathways (Supplementary Fig. 4b). We cloned its human ortholog ACADSB and validated its ability to directly bind RNA using the PNK assay (Fig. 4f). This finding demonstrates that the use of FA crosslinking can extend the discovery potential of comprehensive RNA interactome capture.“

Supplementary Figure 13

Supplementary Fig. 13: Verification of new RNA binding proteins by Protein Isolation by large RNA Pulldown (PIRP).

- Workflow for isolating large RNA binding proteins. Briefly, UV irradiated and non-irradiated cells were resuspended in RLTplus buffer and passed through genomic DNA binding column. Ethanol was added to the flowthrough upto 35%. The mixture was loaded on an RNA binding Midi column. The column was washed with RW1, RPE buffers and then dried by centrifugation. The RNA along with bound proteins was eluted from the column by using hot nuclease free water.
- Bioanalyzer profiles of the eluted large RNA from crosslinked and non-crosslinked cells was analysed along with total RNA from the same cells.
- Silver stained protein profiles representing all large RNA bound proteins.
- Western blots used to validate the LRIC protocol with classical RBPs (ADAR, DHX9, HNRNPM1, HUR3A2, POLRMT and non-classical RBPs like GAPDH).

- e) Validation of proteins involved in oxidative phosphorylation (ATP5A, SDHA), protein involved in tRNA import and protein import porin (TOM20) and proteins involved in DNA repair (MSH6 and XRCC5).

Fig. 4f: Validation of RBPs using PNK assay by affinity purification of Flag-His-Biotin-His (Fl-HBH) tagged proteins which are stably expressed in Flp Trex HEK293T cells. RNA-protein hetero-conjugates are detected by autoradiography of P32 labelled RNA (top labeled with red asterisk) while proteins are detected by Western blot (bottom) using anti Flag antibody (labeled with asterisk). From left to right: the assay was performed on the following class of proteins: (i) RNA related proteins comprising the classical RRM domain containing RBPs Serrate RNA Effector Molecule (SRRT) and Peptidylprolyl Isomerase E (PPIE) as well as Receptor For Activated C Kinase 1 (RACK1) harboring a WD40 domain and SAP18, a new direct RNA binder); (ii) Cell signalling proteins Phosphatidylethanolamine Binding Protein 1 (PEBP1) and casein kinase 1 delta (CSNK1D); (iii) cytoskeleton ERM family proteins Radaxin (RDX), Erizin (ERZ), and Moesin (MSN); and finally (iv) metabolic enzymes Aldehyde Dehydrogenase 6 Family Member A1 (ALDH6A1), Malonyl-CoA-acyl carrier protein Transacylase (MCAT) and short/branched chain acyl-CoA dehydrogenase (ACADSB). For PEBP1 and RDX additional controls with high amounts of RNase I treatment (100 Units) are included along with the standard (1 Unit) treatment employed in all experiments.

In addition, please consider that the description of the experimental workflow depicted in Suppl. Fig. 13a was added to the methods part of the revised manuscript [Page 40-41].

“Protein Interactome of large RNA

We have developed a large RNA (RNA length >200 nucleotides) interactome capture protocol which shares similarities to recently published 2C protocol⁴⁶. The procedure uses silica based purification of large RNA (RNA longer than 200 nucleotides). For large scale experiments (25 million HEK293T cells equivalent to one 15 cm² plate) the same procedure was performed using (RNAeasy Midi kit Cat No. 75144) with some modifications. For the large scale interactome capture 25 million UV crosslinked (254 nm, 200 mJ/cm²) and non-crosslinked HEK293T cells (a 15 cm confluent plate) were resuspended in 3.5 ml RLTplus Buffer (Qiagen Cat No. 1053393, probably contains ~5 M guanidine thiocyanate and proprietary detergents) with 35 µl of 14.3 M beta mercaptoethanol. The samples were homogenized with a rotor stator homogeniser at 2000 rpm for 30 sec. The lysate was loaded on the midi genomic DNA elimination column (Enzymax EZC222) and spun through at 3000xg for 5 min. An equal volume of 70% ethanol was added to the flowthrough and loaded onto a midi RNA column (RNAeasy Midi kit Cat No. 75144). The flowthrough was collected after centrifugation at 3000xg, 3 min and stored at RT for a second iteration of RNA isolation. The column was washed once with 3.5 ml of RW1 wash buffer (Proprietary

composition: contains guanidine salts) and twice with 2.5 ml RPE wash buffer from RNAeasy Midi kit with centrifugation at 3000xg, 3 min. The columns were later dried by additional centrifugation at 3000xg 2 min. The RNA along with the UV crosslinked RNA-protein complexes were eluted by adding 500 µl of preheated (75 °C) nuclease-free water to the columns and incubating for 1 min at room temperature. The eluate was collected as flowthrough after centrifugation at 3000xg for 3 min. The column was reused to isolate RNA from the flowthrough fraction (kept aside beforehand). The eluted RNA and protein content was quantified by using Qubit fluorometric quantification. RNA was also quality controlled by Bioanalyzer pico chip CE. For protein analysis, RNA was degraded by addition of 1/10th volume of 10X RNA digestion buffer (100 mM Tris-HCl pH 7.5, 1.5 M NaCl, 0.5% NP-40) and 1 µl of a cocktail of RNase A and benzonase (1 µl benzonase, 25 U, Novagen + 1 µl RNases A, Thermo Scientific, 10µg/µl + 300 µl of 1X digestion buffer) and incubated at 37 °C for 12 hrs. Proteins were subsequently visualized by both silver staining and Western blot analysis. For small scale capture experiments (1-10 million cells) the above procedure can be performed employing Qiagen AllPrep DNA/RNA mini kit 80204.”

2.7. The authors used the PEAKS software for assignment of crosslinked amino acids after sequencing of peptides that are still crosslinked to RNA. According to the supplementary data the database search took several days which is not worthwhile. Furthermore, the authors claim in the introduction that currently available software tools for identification of proteins and amino acids that are crosslinked to RNA require manual inspection of the mass spectra. However, the authors also manually evaluated their results when using PEAKS. The prolonged computation time and the fact that mass spectra needed to be manually validated make an advantage of the data evaluation strategy introduced here questionable when compared with existing software tools such as MSFragger or RNPxl.

We agree with the reviewer that the generation of search strategy 3 (SS3) data presented in the paper is time consuming (on a standard Windows 7 desktop computer). However, we also explain in detail in the supplemental data, that we additionally tested two other search strategies (SS1 and SS2) that offer substantially higher speed and do not compromise a lot in regard to detection sensitivity. Furthermore, SS1 is as fast as any normal database search (for non-modified and modified peptides bearing a few standard PTMs) and only retrieves those RNA-crosslinked peptides identified with the highest confidence by SS3 (thereby minimizing manual inspection to almost zero). As a practical solution we are utilizing SS1 for the setup/optimization of experiments and once complete replicate MS data are acquired we process them with SS3. Therefore, the comparison on the speed and precision of the PEAKS analysis with MSFragger and/or RNPxl is currently not within our scope. Also, it can easily be envisioned that in the near future SS3 can be boosted by first filtering out all acquired (standard peptide ion) spectra, which match to “normal” non-crosslinked peptides. This will however require PEAKS Bioinform. Inc. to modify some proprietary code in their software. We are in contact with them.

2.8. Upon manual annotation of their crosslink spectra the authors claim that they identified 'novel' crosslinks in the MS2 spectra where the N-glycosidic bond

of the crosslinked RNA moiety has been cleaved. Similar spectra annotation has been performed and published by the Urlaub laboratory in previous studies.

We again would like to apologise for a clear misunderstanding. The fact that the chemical dissociation of the bond between the ribose and the nucleobase happens during MS2-sequencing (collision induced dissociation) is of course known⁵². Such ions in the MS2 spectra are used for the interpretation of RNA-peptide hetero-conjugate sequence. We claim to identify ionized molecules (at MS1 spectra level), which already underwent N-glycosidic bond cleavage. The N-glycosidic bond cleavage observed by us may occur to a certain extent during RNA-peptide ion transfer into the mass spectrometer ("in source dissociation"; the Q Exactive series S-lens ion funnels are RF devices) at pH 2.3 and 270°C. Thus, we need to define the modifications of "bases only" for the crosslinked peptides. We could not find any publication which reports such a modification. The reason for this might simply be that "base-only" was previously never considered as a variable modification for database search engines.

2.9. This study mostly targets *Drosophila* and in the second part (CAPRI method) compares the results with human HEK cells. The authors use these two species as representatives for metazoan. However, this conclusion is probably over-interpreted and additional species should be selected when discussing the 'metazoan RBPome'.

The adjacent peptide analysis until now has been reported only in human and mouse cells. We have used *Drosophila* as an example and future studies could include other representative metazoans. We have now included this point in the discussion. We feel that addition of other species is beyond the scope of this manuscript.

2.10. The methods section does not allow to reproduce the results. There are too many sections and it is not clear which sections need to be performed for which experiment. For instance, the section 'Interactome capture' is completely unnecessary.

Sections on data analysis (e.g. database search and sorting database search results) should be combined and not spread out over several sections. Large parts of the method sections are repetitive (e.g. on-bead digestion).

We have updated the methods section to be more detailed and clear. We will be happy to help the reviewer with reproduction of any of the methods by providing ready to use lab protocols.

To further clarify this issue, the interactome capture section is included to give details for reproducibility of data, especially for FA-RBPome capture.

2.11. The described workflow with different techniques including comparison is rather complicated and time-consuming. I doubt that this workflow in the described form is a technical improvement which will be used by the community.

We respectfully disagree with this comment.

When the FASP protocol for protein digestion was introduced it very much simplified the sample preparation of proteins, because it achieved complete denaturation, buffer exchange and salt removal for high recovery of peptides. The only difference in this protocol is that for CAPRI we use the (size of the) RNA moiety for retaining the peptide-RNA hetero-conjugate on top of the MWCO filter. This modification of standard FASP was subsequently termed "RNA-FASP". As described above, this greatly simplifies the sample preparation and makes it more reproducible, avoiding additional enrichment steps for capturing RNA crosslinked peptides. We would like to point out the compatibility of the RNA-FASP protocol with any other upstream processing protocols for deciphering RBPomes or conducting RPD mapping (similar to the ones already published). Advantages of the protocol for the CAPRI technology are as follows: (i) high stringency washes and (ii) one-time bead usage in the domain protocol. In order to be able to perform the same type of experiment with the RBDmap procedure we would need three steps of oligo-dT bead binding events (one for RBPome isolation, a second one for Lys-C digested peptide isolation and a third one for the release of adjacent peptides via trypsin proteolysis followed by elution of RNA crosslinked peptides). In addition, utilization of high stringency solutions including 8M urea are not compatible with RBDmap.

2.12. The figures are overloaded and not entirely clear. When printed at final size some of the text will not be readable. Figure 4d is not readable at current size and resolution. Furthermore, the authors fail to illustrate in any figure which crosslinks derived from UV and which from FA. This is misleading.

Thank you for pointing this out. Upon revision, we have made the font sizes of Figure 4d larger and increased the font sizes in other figures wherever possible. We have also checked the images by printing them. In the original manuscript, we did not show any crosslinked peptides derived from the FA crosslinking. However, in the revised manuscript we have included peptides identified by formaldehyde mediated RBD identification as explained in point 2.4. above.

2.13. I wonder whether the authors are aware of the current literature in the field of RNA-binding proteins. Particularly the parts on FA crosslinking and UV crosslinking are incompletely referenced.

We have tried to include more references in the paper wherever possible and now hope the reviewer will be satisfied with these.

General comments:

2.14. The name CAPRI is not a good choice for a new technique as this abbreviation is already used in the structural biology community which is also addressed here. CAPRI stands for "Critical Assessment of PRediction of Interactions" and is a community-wide experiment on the evaluation of protein-protein docking for structure prediction. I suggest a different name which will not be confused in the structural biology community.

We apologize for the fact that it escaped our attention that CAPRI is used in the structural biology field. However, since our work will be relevant for the OMICS world and the RNA biology community we prefer to keep the name CAPRI. Nonetheless, we suggest a possible alternative title: "ProDIRP:Protein Domain Isolation by RNA Pulldown".

2.15. The manuscript is written in a lousy manner. Some parts are very lengthy without providing sufficient information and clarity. The referencing is in large parts incomplete. Some of the results sections are rather a discussion. There are many abbreviations that are introduced, however, these are not used afterwards (even in the same paragraph). Some terms are spelled differently throughout the entire manuscript. See detailed comments below.

We apologise for the inconsistencies in the spellings. We have taken care of this upon revision.

Minor comments:

2.16. The term 'cross-linking, cross-linked, ...' is spelled differently throughout the manuscript (cross-link, crosslink, XL, ...).

We thank the reviewer for pointing out the inconsistencies. We have now used "crosslink" everywhere. However, "XL" is being used in the nomenclature for peptides selected after PEAKS analysis: "XL-peptides".

2.17. I suggest to use a different term than 'rest' for the specific crosslink PTM.

We definitely would like to use a different one, however the name is for now included in the PEAKS software analysis files. Changing the name in the analysed files is currently not possible for us. In the future, better nomenclature could be used on the availability of a PEAKS "annotation tool" (ongoing collaboration with Bionfor. Inc.). Also the nomenclature "rest" was intended to make it clear that we are dealing here with a neutral loss rest (MS/MS fragmentation induced) and not with a standard modification (like e.g. acetylation), which stays attached to the modified amino acid residue during MS/MS "sequencing". Standard PTM here means a modification that does not change its chemical composition between the two processes of peptide ion generation (ionization) and MS/MS fragmentation.

2.18. The abbreviation FA is several times introduced (e.g. twice on page 23) but not used and used in the text. This is just an example, there are several other abbreviations that are introduced and afterwards used or not used.

FA repetition is corrected in the main text now. Many abbreviations are used in text in order to refer to the images, wherein they are used multiple times. With more than six different kinds of peptides referred to in the text.

2.19. There are many typos, missing punctuation and inconsistencies.

We have corrected all the typos that we could find.

2.20. What exactly is meant by 'trituration' (p. 24)? I suggest to use the term 'grinding'.

The term trituration was introduced in the context of using a pipette to lyse/homogenise/mix a frozen cell pellet in the lysis buffer solution. However, this is not the traditional use of the word. We have added the description of the process instead of the word in the methods section. Grinding may not be an appropriate term here as it is usually used for manual or motor pestle tissue "grinding".

2.21. The abbreviation MeCNI (e.g. p. 27) for ACN is rather unusual.

More and more people in the proteomics community are using MeCN (Choudhary lab, Mann lab) but we have changed it to ACN as suggested by the reviewer.

2.22. The section 'background of crosslinked peptide analysis' in the methods section is entirely redundant. In addition, this section (as well as the introduction and some other sections) is poorly referenced...

The background is provided specifically for researchers not working in the UV RNA-protein crosslinking field in order to make it easy for them to understand the way RNA-PTMs are defined. We have tried to include more references in the manuscript wherever possible.

In summary, the authors present a method that has been already published and extensively applied in others studies, mainly from the Hentze and Urlaub laboratories. The 'novelty' of introducing formaldehyde as RNA-protein crosslinker and applying the already described workflow from the Hentze group is in fact a clear misconception due to the comments listed above. Regrettably I have to recommend rejection of this manuscript.

We hope that we have addressed and clarified the major concerns raised by Reviewer 2. Particularly, we have put special emphasis on the advantages of the chosen concept over previously published methods and pointed out that CAPRI considerably simplifies RBP and RBD discovery making it accessible to a broad community of research groups that have access to a MS/proteomics facility. Nonetheless, we are thankful to the reviewer for his/her extensive and thorough evaluation of the manuscript.

Reviewer #3

Panhale et al. present a comprehensive strategy to identify RNA interacting proteins. In a first step, they perform an mRNA interactome capture using both UV and formaldehyde crosslinking. The latter allows them to identify components of RNPs that do not directly contact RNA which is an interesting extension to facilitate the characterisation of RNP complexes. Furthermore, they develop CAPRI which combines two previously known approaches to detect RNA-binding domains (RBDs). They use this to identify a large number of known and novel RBDs in human and *Drosophila* cell culture. One of their discoveries is that they observe a significant fraction of intrinsically disordered regions in contact with RNA, some of which are evolutionarily conserved between human and *Drosophila*.

The data of this study represents a valuable resource that will be useful for RNA biologists. The authors present their findings in a very well-written manuscript with clear figures.

We thank the reviewer for his/her supportive comments.

Major comments:

3.1. The authors claim that the FA-RBPome allows to discover a secondary interaction layer. However, they currently do not provide any independent evidence or experimental validations for these additional interactions. E.g. are these proteins from metabolism, endocytosis or DNA replication co-localising with any type of RNA granules? Also, in order to evaluate the benefit of the FA-RBPome, it would be important to separately analyse the subset which is not present in the UV-RBPome.

Upon revision, we have addressed the reviewer's comment by separately analysing the proteins that are exclusively identified by FA as well as by conducting independent and orthogonal verification experiments for newly discovered RNA binding proteins (Fig. 4f, Supplementary Fig. 13).

In order to achieve this, we selected *Drosophila* FA-RBPome specific proteins (1037 proteins) which possess a human ortholog (1009 proteins). Next, we asked how many of the latter proteins have already been identified by previous UV interactome capture studies (carried out in human cells)³³. Moreover, we extracted information concerning the presence of classical RNA binding domains in these proteins from the Interpro database and nucleic acid related GO term annotations. Interestingly, of the 1009 FA-RBPome specific *Drosophila* proteins with human orthologs, 725 have either already been shown to bind RNA (previous UV RBPome reports) or are annotated as RNA related proteins (Interpro RBDs, "RNA" in GO terms). The remaining 284 FA specific proteins are novel and most likely reflect indirect "piggyback" binders (Revision Fig. 2).

Revision Fig. 2: Clustered heatmap illustrating the presence (dark purple) or absence of FA-RBPome specific proteins (bottom row) in other RNA-related protein datasets. Legend (from top to bottom): proteins harboring known Interpro RBDs, proteins containing "RNA" in their GO "Molecular Functions" (MF) or GO "Biological Processes" (BP), proteins present in other *Drosophila* embryo UV RBPomes, *Drosophila* proteins possessing human orthologs from Mullari et al., orthologous proteins from the recently published OOPS³¹ and XRNAX³² methods and orthologs from the comprehensive Hentze lab review³³.

We have validated many RBPs using two independent protocols. For details please see reply to the second Reviewer (point 2.6.). Briefly, we utilized an independently developed large RNA interactome capture protocol (Supplementary Fig. 13a-d) similar to the 2C protocol⁴⁶ to validate proteins involved in oxidative phosphorylation and DNA repair (Supplementary Fig. 13e). We have also independently verified 9 human orthologs of novel RBPs discovered in the *Drosophila* UV-RBPome using the PNK assay (Fig. 4f). The proteins verified are involved in signal transduction and metabolic pathways or belong to the family of cytoskeletal proteins. Here we would like to specifically highlight a novel *Drosophila* FA-RBPome-specific RNA associated protein, a short/branched chain acyl-CoA dehydrogenase (CG3902). Its involved in fatty acid degradation and valine, leucine, isoleucine degradation pathway (Supplementary Fig. 4b). We succeeded to validate the direct RNA binding of its human ortholog ACADSB via UV (PNK assay).

We additionally attempted to validate several novel RBPs uncovered in the *Drosophila* FA-RBPome by PNK assay (Fig. 4f). Unfortunately, this is currently not possible due to the vast extent of de-crosslinking taking place during the process of sample preparation for immunoblot analysis. We are in the process of optimising a formaldehyde based CLIP seq similar to RIPIT-seq²⁶. This is however, currently beyond the scope of this manuscript.

In the case of endocytic pathways, it has been recently shown that RNA granules associate with endosomes along the axons of retinal ganglion cells³⁵. In

these, the mRNA and translating ribosomes get associated with late Rab7a endosomes.

3.2. At present, it is not clear whether there really is a synergistic value of CAPRI. It seems that two already known technologies were run at the same time, but I am not sure about the added value of this combination, especially since the authors simply merge both datasets into a combined list of CAPRI peptides.

We do agree that both adjacent and crosslinked peptide identification experiments have been described before, but as separate techniques. However, here, we have combined the two by introducing a single pot analysis (please see also point 1.2 and 1.3 in response to reviewer 1). This comes with the additional benefit of verifying the validity of the adjacent peptides concept (for RBD mapping) as we can demonstrate that many crosslinked peptides are nicely overlapping with the in silico extended adjacent peptides. Of note, we apply "RNA-FASP" (employing standard FASP filters) for both the enrichment and clean-up of the crosslinked RNA peptide hetero-conjugates. As mentioned in our reply to reviewer 1 and 2, "RNA-FASP" liberates crosslinked peptides from RNA. At the same time undigested nucleic acids and proteins as well as other high molecular weight biomolecules are still retained by the filter. Hence, an additional sample processing step like TiO₂-based enrichment of crosslinked peptides can be entirely omitted. The efficiency and reproducibility of single stranded RNA capture and release from FASP filters was thus the key invention for developing CAPRI.

Minor comments:

3.3. Since evolutionary conservation is only a minor point in this study, I would mention it in the abstract but not in the title.

We respectfully disagree with the reviewer. We believe that evolutionary conservation on new RNA binding domains is an important conclusion of the paper. Importantly, the independent validation experiments described above and in the response to reviewer 2 have entirely been conducted on human orthologs of "new" Drosophila RBPs.

3.4. Figure 1d: The correlation of intensities used in this figure is not meaningful, as the intensities can be dominated in both cases from high background values. I assume that a comparison of FA-RBPome against total proteome could show a similar correlation. Therefore, enrichment should be used rather than intensities.

The aim of Fig. 1d is to show that proteins which are highly abundant in the UV-RBPome (because they are abundant RBPs and/or UV-crosslink very well) would most likely be highly abundant constituents of the FA-RBPome as well. We consider this result important because it nicely demonstrates that, for proteins detected by both RBP capture workflows, major crosslinking method related systemic biases do not exist (Revision Fig. 3).

Revision Fig. 3: Pearson correlation (PC) scatter plots of total protein intensities observed in the UV- and FA-RBPome. (a) PC plots comparing each of the biological replicates for UV and FA captures separately. (b) PC plots for proteins comparing UV with FA biological replicates, respectively. (c) PC plot visualizing “crosslinking versus control” log₂ fold changes for formaldehyde (FA/noFA) compared to UV (UV/noUV) RBPs. PC values comparing biological replicates belonging to the same crosslinking method are very good (>0.9). PC values comparing biological replicates between FA and UV crosslinking datasets are decent (>0.7).

3.5. page 6: "The overall coverage of RNA-related proteins was higher in the FA-RBPome compared to the UV-RBPome (Fig. 1f)." A comparison of absolute numbers is not meaningful, given that the detected FA-RBPome is substantially larger than the UV-RBPome. Using fractions instead would allow to evaluate the relative enrichment of each dataset in the different categories. It would also be interesting to show the same analysis separately for the UV-RBPome-overlapping and unique proteins from the FA-RBPome.

We agree with the reviewer that enrichment would be meaningful to consider in case if we were comparing the specificity of the two crosslinking methods. However, we wanted to show that FA recovers more classical RBPs than UV itself (Fig. 1f). This is important for any future experiments designed to detecting changes in RNA protein binding in living cells. Supplementary figure 3a together with Revision Fig. 4a and b (below) reveal the enrichment of RNA associated GO molecular functions in UV and FA-RBPomes, respectively. In Revision Fig 4c it becomes apparent that RNA related GO molecular functions are highest enriched in the Drosophila ADJ-RBPome, followed by the UV-RBPome and then the FA-RBPome. The properties of FA-RBPome-specific proteins can be seen in Revision Fig. 3.

Revision Fig. 4: Enriched Gene Ontology Molecular Functions for UV-RBPome (a) and FA-RBPome (b) shown as a fraction of interactome (X-axis) and colored as fraction of coverage for each of the terms in the Drosophila proteome. The Benjamini-Hochberg corrected p-values are shown on the Y-axis for the respective GO terms. c) Bar chart depicting a comparison of the fold enrichment in the top FA-RBPome GO MF terms in each of the FA-, UV- and ADJ-RBPomes in Drosophila. The Benjamini-Hochberg corrected p-values for FA-RBPome are reported on the Y-axis for the respective GO terms.

3.6. page 7: "However, more than 65% of the identified Drosophila proteins in both UV- and FA-RBPomes did not harbour any of these RBDs (Fig. 1g)." This statement does not fit to the values shown in Figure 1g (i.e. 61% for UV-RBPome).

We would like to thank the reviewer for this comment. Indeed, we apologise for this typo. The correct statement here would be more than 60% (61.04% in UV-RBPome and 69.74% for FA-RBPome).

3.7. It would be helpful to report the results with higher precision. E.g. on page 10, "In each of the structures, the amino acid at the site of crosslink was in close proximity to the RNA". How many structures were evaluated?

We thank the reviewer for pointing this out. After evaluating the structures again we have made our statement more precise in the main text [Page 10].

"To this end, we visualised the peptides/amino acids on 7 available RNA-protein crystal structures (Supplementary Table 6, Fig. 3d, e Supplementary Fig. 7d,e). All mapped XL-peptides were located close to RNA in 3D space. Of the 17 crosslink sites that we mapped in these structures (Fig. 3d and Supplementary Table), 15 were in close proximity to RNA."

3.8. In the part on IDRs, the manuscript gets a bit speculative and lengthy. I think this part should be shortened.

We have shorten the text in the results section of IDRs.

3.9. I think Figure 4c is not cited in the manuscript.

We thank the reviewer for spotting this error. The sentence referring to this panel had inadvertently been deleted in the previous version of the manuscript and has now been restored (see page 14). The figure shows a Venn diagram comparing the *Drosophila* UV-, ADJ- and XL-RBPomes.

References

1. Sysoev, V. O. *et al.* Global changes of the RNA-bound proteome during the maternal-to-zygotic transition in *Drosophila*. *Nat. Commun.* **7**, 12128 (2016).
2. Wessels, H.-H. *et al.* The mRNA-bound proteome of the early fly embryo. *Genome Res.* **26**, 1000–1009 (2016).
3. Schneider, I. Cell lines derived from late embryonic stages of *Drosophila melanogaster*. *J. Embryol. Exp. Morphol.* **27**, 353–365 (1972).
4. Casas-Vila, N. *et al.* The developmental proteome of *Drosophila melanogaster*. *Genome Res.* **27**, 1273–1285 (2017).
5. Engholm-Keller, K. & Larsen, M. R. Titanium dioxide as chemo-affinity chromatographic sorbent of biomolecular compounds--applications in acidic

- modification-specific proteomics. *J. Proteomics* **75**, 317–328 (2011).
6. Kramer, K. *et al.* Photo-cross-linking and high-resolution mass spectrometry for assignment of RNA-binding sites in RNA-binding proteins. *Nat. Methods* **11**, 1064–1070 (2014).
 7. Baltz, A. G. *et al.* The mRNA-bound proteome and its global occupancy profile on protein-coding transcripts. *Mol. Cell* **46**, 674–690 (2012).
 8. Castello, A. *et al.* Comprehensive Identification of RNA-Binding Domains in Human Cells. *Mol. Cell* **63**, 696–710 (2016).
 9. Kong, A. T., Leprevost, F. V., Avtonomov, D. M., Mellacheruvu, D. & Nesvizhskii, A. I. MSFragger: ultrafast and comprehensive peptide identification in mass spectrometry-based proteomics. *Nat. Methods* **14**, 513–520 (2017).
 10. Alekseyenko, A. A., Gorchakov, A. A., Kharchenko, P. V. & Kuroda, M. I. Reciprocal interactions of human C10orf12 and C17orf96 with PRC2 revealed by BioTAP-XL cross-linking and affinity purification. *Proc. Natl. Acad. Sci. U. S. A.* **111**, 2488–2493 (2014).
 11. Déjardin, J. & Kingston, R. E. Purification of proteins associated with specific genomic Loci. *Cell* **136**, 175–186 (2009).
 12. Mohammed, H. *et al.* Rapid immunoprecipitation mass spectrometry of endogenous proteins (RIME) for analysis of chromatin complexes. *Nat. Protoc.* **11**, 316–326 (2016).
 13. Rafiee, M.-R., Girardot, C., Sigismondo, G. & Krijgsveld, J. Expanding the Circuitry of Pluripotency by Selective Isolation of Chromatin-Associated Proteins. *Mol. Cell* **64**, 624–635 (2016).
 14. Liu, X. *et al.* In Situ Capture of Chromatin Interactions by Biotinylated dCas9. *Cell* **170**, 1028–1043.e19 (2017).
 15. Jackson, V. Studies on histone organization in the nucleosome using

- formaldehyde as a reversible cross-linking agent. *Cell* **15**, 945–954 (1978).
16. Sutherland, B. W., Toews, J. & Kast, J. Utility of formaldehyde cross-linking and mass spectrometry in the study of protein-protein interactions. *J. Mass Spectrom.* **43**, 699–715 (2008).
 17. Hoffman, E. A., Frey, B. L., Smith, L. M. & Auble, D. T. Formaldehyde crosslinking: a tool for the study of chromatin complexes. *J. Biol. Chem.* **290**, 26404–26411 (2015).
 18. Guerrero, C., Tagwerker, C., Kaiser, P. & Huang, L. An integrated mass spectrometry-based proteomic approach quantitative analysis of tandem affinity-purified in vivo cross-linked protein complexes (qtax) to decipher the 26 s proteasome-interacting network. *Mol. Cell. Proteomics* **5**, 366–378 (2006).
 19. Tagwerker, C. *et al.* A tandem affinity tag for two-step purification under fully denaturing conditions: application in ubiquitin profiling and protein complex identification combined with in vivocross-linking. *Mol. Cell. Proteomics* **5**, 737–748 (2006).
 20. Aktaş, T. *et al.* DHX9 suppresses RNA processing defects originating from the Alu invasion of the human genome. *Nature* **544**, 115–119 (2017).
 21. Schmitt-Ulms, G. *et al.* Time-controlled transcadiac perfusion cross-linking for the study of protein interactions in complex tissues. *Nat. Biotechnol.* **22**, 724–731 (2004).
 22. Larance, M. *et al.* Global Membrane Protein Interactome Analysis using In vivo Crosslinking and Mass Spectrometry-based Protein Correlation Profiling. *Mol. Cell. Proteomics* **15**, 2476–2490 (2016).
 23. Möller, K., Rinke, J., Ross, A., Buddle, G. & Brimacombe, R. The use of formaldehyde in RNA-protein cross-linking studies with ribosomal subunits from *Escherichia coli*. *Eur. J. Biochem.* **76**, 175–187 (1977).

24. Wheeler, E. C., Van Nostrand, E. L. & Yeo, G. W. Advances and challenges in the detection of transcriptome-wide protein-RNA interactions. *Wiley Interdiscip. Rev. RNA* **9**, (2018).
25. Friedersdorf, M. B. & Keene, J. D. Advancing the functional utility of PAR-CLIP by quantifying background binding to mRNAs and lncRNAs. *Genome Biol.* **15**, R2 (2014).
26. Singh, G., Ricci, E. P. & Moore, M. J. RIPit-Seq: a high-throughput approach for footprinting RNA:protein complexes. *Methods* **65**, 320–332 (2014).
27. Chu, C., Qu, K., Zhong, F. L., Artandi, S. E. & Chang, H. Y. Genomic maps of long noncoding RNA occupancy reveal principles of RNA-chromatin interactions. *Mol. Cell* **44**, 667–678 (2011).
28. Chu, C. *et al.* Systematic discovery of Xist RNA binding proteins. *Cell* **161**, 404–416 (2015).
29. Simon, M. D. Capture Hybridization Analysis of RNA Targets (CHART). in *Current Protocols in Molecular Biology* (2013).
30. Quinn, J. J. *et al.* Revealing long noncoding RNA architecture and functions using domain-specific chromatin isolation by RNA purification. *Nat. Biotechnol.* **32**, 933–940 (2014).
31. Queiroz, R. M. L. *et al.* Comprehensive identification of RNA-protein interactions in any organism using orthogonal organic phase separation (OOPS). *Nat. Biotechnol.* (2019). doi:10.1038/s41587-018-0001-2
32. Trendel, J. *et al.* The Human RNA-Binding Proteome and Its Dynamics during Translational Arrest. *Cell* (2018). doi:10.1016/j.cell.2018.11.004
33. Hentze, M. W., Castello, A., Schwarzl, T. & Preiss, T. A brave new world of RNA-binding proteins. *Nat. Rev. Mol. Cell Biol.* **19**, 327–341 (2018).
34. Jansen, R.-P., Niessing, D., Baumann, S. & Feldbrügge, M. mRNA transport

- meets membrane traffic. *Trends Genet.* **30**, 408–417 (2014).
35. Cioni, J.-M. *et al.* Late Endosomes Act as mRNA Translation Platforms and Sustain Mitochondria in Axons. *Cell* (2018). doi:10.1016/j.cell.2018.11.030
 36. Ulvila, J. *et al.* Double-stranded RNA Is Internalized by Scavenger Receptor-mediated Endocytosis in Drosophila S2 Cells. *J. Biol. Chem.* **281**, 14370–14375 (2006).
 37. Lorenz, C. *et al.* Protein expression from exogenous mRNA: uptake by receptor-mediated endocytosis and trafficking via the lysosomal pathway. *RNA Biol.* **8**, 627–636 (2011).
 38. Kao, C., Vaughan, Running & Qi. Mapping protein–RNA interactions. *VAA729* (2012).
 39. Kennedy-Darling, J. & Smith, L. M. Measuring the Formaldehyde Protein–DNA Cross-Link Reversal Rate. *Anal. Chem.* **86**, 5678–5681 (2014).
 40. Srinivasa, S., Ding, X. & Kast, J. Formaldehyde cross-linking and structural proteomics: Bridging the gap. *Methods* **89**, 91–98 (2015).
 41. Kao, C., Vaughan, Running & Qi. Mapping protein–RNA interactions. *Virus Adaptation and Treatment* 29 (2012).
 42. Moore, K. S. & 't Hoen, P. A. C. Computational approaches for the analysis of RNA-protein interactions: A primer for biologists. *J. Biol. Chem.* **294**, 1–9 (2019).
 43. Hughes, C. S. *et al.* Ultrasensitive proteome analysis using paramagnetic bead technology. *Mol. Syst. Biol.* **10**, 757 (2014).
 44. Chiva, C., Ortega, M. & Sabidó, E. Influence of the digestion technique, protease, and missed cleavage peptides in protein quantitation. *J. Proteome Res.* **13**, 3979–3986 (2014).
 45. Gershon, P. D. Cleaved and missed sites for trypsin, lys-C, and lys-N can be

- predicted with high confidence on the basis of sequence context. *J. Proteome Res.* **13**, 702–709 (2014).
46. Asencio, C., Chatterjee, A. & Hentze, M. W. Silica-based solid-phase extraction of cross-linked nucleic acid-bound proteins. *Life Sci Alliance* **1**, e201800088 (2018).
 47. Conrad, T. *et al.* Serial interactome capture of the human cell nucleus. *Nat. Commun.* **7**, 11212 (2016).
 48. Maticzka, D., Ilik, I. A., Aktas, T., Backofen, R. & Akhtar, A. uvCLAP is a fast and non-radioactive method to identify in vivo targets of RNA-binding proteins. *Nat. Commun.* **9**, 1142 (2018).
 49. Singh, K. K. *et al.* Human SAP18 mediates assembly of a splicing regulatory multiprotein complex via its ubiquitin-like fold. *RNA* **16**, 2442–2454 (2010).
 50. Zaravinos, A., Bonavida, B., Chatzaki, E. & Baritaki, S. RKIP: A Key Regulator in Tumor Metastasis Initiation and Resistance to Apoptosis: Therapeutic Targeting and Impact. *Cancers* **10**, (2018).
 51. Schitteck, B. & Sinnberg, T. Biological functions of casein kinase 1 isoforms and putative roles in tumorigenesis. *Mol. Cancer* **13**, 231 (2014).
 52. Lenz, C., Kühn-Hölsken, E. & Urlaub, H. Detection of protein-RNA crosslinks by NanoLC-ESI-MS/MS using precursor ion scanning and multiple reaction monitoring (MRM) experiments. *J. Am. Soc. Mass Spectrom.* **18**, 869–881 (2007).

Reviewers' comments:

Reviewer #1 (Remarks to the Author):

The authors have satisfactorily addressed my concerns. Yet, overseeing the questions raised by the other referees and the responses to them, I would like to see clarified how FA specifically crosslinks proteins to RNA as opposed to DNA. The reason for bringing this up is that FA-mediated crosslinking of RNA is likely the most debatable aspect of the study. One reason is that FA has been used in a range of chromatin-targeted methods to crosslink proteins to DNA. I appreciate that the conditions used here are 'mild' (i.e. low FA concentration for a short period of time), however the possibility of DNA-protein crosslinking cannot be ignored. It would thus be extremely useful if the authors can demonstrate absence of DNA in their preparations to exclude the possibility that larger number of identified proteins after FA-crosslinking are due to contamination with DNA. This will help to avoid confusion in the field, and to spread the use of CAPRI as a robust and specific methodology to determine protein-RNA interactions.

Reviewer #2 (Remarks to the Author):

The manuscript of Panhale et al. was revised according to the reviewer's comments, including some additional experiments. While I appreciate the detailed response to my concerns raised, I have to say that the response letter is more silver-tongued than actually addressing the concerns. There are still major doubts which have not been dispelled. My major points of concern are:

- The novelty/meaning of the technique is not justified. As also pointed out by reviewer 3, the method (which even obtained its own name) is a combination of two already existing techniques. The application of FA crosslinking in the context of this technique is also not clear. The authors perform FA- and UV-cross-linking, while only UV-crosslinking is part of the actual technique (i.e. CAPRI, see pages 7 and 8). It is not clear why FA-crosslinking is employed at all as it is not discussed in the CAPRI technology (pages 7/8). In addition, it is not clear how the four data sets FA-RBPome, UV-RBPome, ADJ-RBPome and XL-RBPome were originated – the latter should be the same as the FA- and UV-RBPome.
- The authors claim that they sufficiently validated their method and verified novel protein-RNA interactions. However, the validation of the technique is rather poorly addressed. For instance, the structures shown in Figure 3d 3e are not clearly visible and only one protein is used for

validation. The authors could have used the multitude of highly resolved structures of spliceosomal subcomplexes (these are: spliceosomal B, C, tri-snRNP, Bact, step I complexes and others) to compare their data (both, xl-peptides and ADJ peptides) with the structures of the proteins in these complexes. I further do not agree with the verification of the newly identified RBPs. For instance, the authors identified novel RBPs in the respiratory chain complexes. Although these proteins are discussed more in detail than others, it appears that the authors do not have detailed knowledge on the proteins and refuse to acquire some basic knowledge. First, there is no biological meaning why the proteins discussed here should bind. They further discuss binding of RNA to the gamma subunit of the ATP synthase (see Supplementary Figure 12), however, their validation reveals binding of the alpha subunit. (Please note, that the Western Blot shown in Supplementary Figure 13e is of poor quality. According to the information provided in the methods section, there is no ATP synthase-specific antibody which used.) In addition, they wrongly state in their manuscript that subunit alpha of the F1 ATP synthase is a member of complex I of the OXPHOS complexes. In fact, ATP synthase is complex V. (Minor point: the authors also discuss cytochrome oxidase – the correct name is cytochrome c oxidase (COX).) Comparing these results with the data presented in Supplementary Figure 12 it becomes apparent that all these interactions are revealed by FA-crosslinking. As discussed below (and in my previous report), these might be false-positives and should carefully checked.

- The authors further claim that their technique outperforms previously published techniques in that it does not require manual spectra annotation. However, on page 10 of their revised manuscript they state that “the spectra were manually curated to produce a final set of crosslinked peptides...”. It also wrongly stated that previously published studies require manual peptide sequence analysis. In the here cited studies manual inspection of the mass spectra was employed for validation of the data analysis workflow.

- The authors further seem to ignore the facts that FA-crosslinks are reversible. Even though crosslinking of the sample could be proven by SDS-PAGE the covalent linkages are most likely disrupted during the ionization process (the authors claim that the transfer capillary is heated to 270 °C). Previous studies using FA-crosslinking mostly used Q-ToF mass spectrometers (or similar) which do not employ a heated transfer capillary (the instrument set-up is different which does not require these temperatures in the ion source). In addition, FA-crosslinking induces protein-crosslinking (as mentioned by the authors in their revised manuscript) which might be the reason for the higher number of identified RNA-binding proteins, i.e. using FA-crosslinking will rather pull down an entire interaction network than the RNA-binding interactome. In this respect the authors cannot conclude at all whether the cross-link has derived from RNA or protein-protein interaction.

- The over-interpretation of the results and the lack of knowledge of the current literature becomes obvious through the presented network of the spliceosomal complex. I doubt that all listed proteins directly bind RNA. There are several proteins that have never been reported in the context of the spliceosome (e.g. Rab19) and are known to function in very different cellular processes (e.g.

membrane trafficking). The color coding and, as a consequence, the over-representation of proteins identified to be involved in RNA interaction is misleading since only a small number of proteins have been identified in the XL- and ADJ-proteomes (which likely represent direct RNA-binding proteins). Other proteins listed are simply identified in the pull-down experiments without any further specification (i.e. XL-site of adjacent site).

- Indeed, it seems that the authors are not aware of the current literature or simply ignore it. A very striking example of this fact is that they claim in the figure legend of Figure 4 that a novel surface interaction with RNA has been identified in the SF3b protein complex. The SF3b complex has been crystallized and the protein-protein and protein-RNA interaction sites (both based on cross-linking experiments) including the repeating “HEAT domains” (here termed “Armadillo-fold”) have been identified by chemical and UV-cross-linking, respectively and described in a very detailed manner in this work (Cretu et al., 2016). Moreover, cryo-EM structures of these proteins within spliceosomes are available showing the interaction of these domain with RNAs. In addition, the authors claim in their manuscript and in the point-by-point response that they identified “new types of protein-RNA cross-links” by MS. This is again wrongly stated. All of the described “new types” of cross-links which have been observed in the MS2 spectra have already been described and documented in several previously published studies that focus on the MS-identification of UV-based protein-RNA cross-linking sites (together with their extensive spectra annotation alongside a detailed repository of supplementary data with).

In summary, I have to conclude that I have severe doubts about the methodology and the data presented here. I can therefore not recommend publication of this manuscript.

Reviewer #3 (Remarks to the Author):

I thank the authors for their comprehensive response. Based on the response and the revision I recommend the manuscript for publication.

NCOMMS-18-19844A
Panhale et al
Point by point response to reviewers

Summary

Here is a brief summary of the major changes and experiments added upon revision.

- We have now made a new Fig. 1 which provides a single schematic outline of the approaches taken in the manuscript that describes CAPRI as well as the complementary approaches used in the study.
- We have performed control experiments to show the negligible extent of DNA contamination in our FA- & UV-RBPome purifications (Supplementary Fig. 1d,f).
- We have added an additional spliceosomal complex member structure (SF3B1) along with the mapped ADJ-peptides in the Fig. 5e.

Reviewer #1

The authors have satisfactorily addressed my concerns. Yet, overseeing the questions raised by the other referees and the responses to them, I would like to see clarified how FA specifically crosslinks proteins to RNA as opposed to DNA. The reason for bringing this up is that FA-mediated crosslinking of RNA is likely the most debatable aspect of the study. One reason is that FA has been used in a range of chromatin-targeted methods to crosslink proteins to DNA. I appreciate that the conditions used here are 'mild' (i.e. low FA concentration for a short period of time), however the possibility of DNA-protein crosslinking cannot be ignored. It would thus be extremely useful if the authors can demonstrate absence of DNA in their preparations to exclude the possibility that larger number of identified proteins after FA-crosslinking are due to contamination with DNA. This will help to avoid confusion in the field, and to spread the use of CAPRI as a robust and specific methodology to determine protein-RNA interactions.

We thank the reviewer for the very positive and constructive feedback! We would also like to thank the reviewer for the very helpful comment and we totally agree that FA crosslinking (at concentrations $\geq 1\%$) is a mainstay in chromatin research (e.g. ChIP-seq, Hi-C). Therefore, we are quite excited about the excellent performance of FA crosslinking in facilitating the isolation of RNP complexes at ten times lower final FA concentrations (up to 0.1% max.) as compared to chromatin-centric methodologies. We thus envision that FA crosslinking assisted RNP capture will be quickly adopted by the scientific community because of FA's widespread use in molecular biology. Besides, we would also like to indicate that successful RBPome purification is critically dependent on the specificity and efficiency of the RNA capture step. As recommended by the reviewer and in order to rule out secondary interactions mediated via putative FA-induced RNA-protein-DNA-protein networks, we have evaluated the potential of DNA contamination by using two methods: Qubit DNA and RNA specific fluorometric quantification (Qubit HS DNA and Qubit BR RNA) as well as qPCR detection of genomic DNA. As a result, we found negligible genomic DNA contamination throughout all samples used in this study (Supplementary Fig. 1 d and f). Hence, we conclude that the FA-RBPome data are not inflated with false positive RNA binding proteins.

Supplementary Fig. 1 d and f : Assessment of RNA enrichment versus genomic DNA contamination present in *Drosophila* RBPome samples. d) Nucleic acid yields (Qubit assay) in ng/μl. No UV: sample derived from non-irradiated cells; UV: sample derived from UV-irradiated cells; no FA: sample without formaldehyde crosslinking; FA: formaldehyde-crosslinked sample; FA RNase: formaldehyde-crosslinked and RNase-digested sample. Error bars represent standard deviation (s.d.) of three biological replicates. f) Fraction of genomic DNA (%) as compared to the respective mRNA amounts based on qPCR quantification of the genes (mRNA and genomic DNA) *hxk*, *rpl22* and *roX2*. Quantification was performed on equal volumes of reverse transcribed eluate (cDNA) and non-reverse transcribed eluate (genomic DNA). Error bars represent standard deviation (s.d.) of three biological replicates.

Reviewer #2

The manuscript of Panhale et al. was revised according to the reviewer's comments, including some additional experiments. While I appreciate the detailed response to my concerns raised, I have to say that the response letter is more silver-tongued than actually addressing the concerns. There are still major doubts which have not been dispelled. My major points of concern are:

2.1 The novelty/meaning of the technique is not justified. As also pointed out by reviewer 3, the method (which even obtained its own name) is a combination of two already existing techniques.

We kindly disagree with the referee and would like to emphasize again that CAPRI presents a novel method because it enables the simultaneous one-pot purification of adjacent and RNA crosslinked peptides. By introducing the novel RNA-FASP step CAPRI is able to combine existing concepts (adjacent and crosslinked peptides) with the advantage of providing a simpler and more stringent workflow (alleviating the need for stable isotope labeling based quantitative LC-MS analysis). The workflow is also compatible with other recently published protocols (e.g. TRAPP¹ and pCLAP²). Hence, we expect it to be heavily used by the RNA community.

2.2 The application of FA crosslinking in the context of this technique is also not clear. The authors perform FA- and UV-cross-linking, while only UV-crosslinking is part of the actual technique (i.e. CAPRI, see pages 7 and 8). It is not clear why FA-crosslinking is employed at all as it is not discussed in the CAPRI technology (pages 7/8). In addition, it is not clear how the four data sets FA-RBPome, UV-RBPome, ADJ-RBPome and XL-RBPome were originated – the latter should be the same as the FA- and UV-RBPome.

In this manuscript we aimed at comprehensively identifying RNA-protein interactions in *Drosophila melanogaster* (Fig. 1). FA crosslinking was thus primarily introduced to identify indirect RNA-protein interactions and direct interactions which are not amenable to UV crosslinking. By both referencing the literature and in our reply to the reviewers we have clearly stated several times that (as with any methodology) UV irradiation-mediated RNA-protein crosslinking has certain biases. FA was thus expected to complement and supplement UV based interactome capture experiments (Supplementary Fig. 2 d, 10). We believe that FA-mediated RBPome capture will prove to be particularly valuable in the field of post transcriptional gene regulation, where researchers try to understand signaling-induced remodelling of the RBPome not only at the level of direct RNA-binding proteins but also at the level of indirect RNA binders. For example indirect RNA-protein interactions have been implicated in RNA stress granule formation (reviewed in Anderson, P. and Kedersha, M., 2008³ and Wolozin, B., 2012⁴). Therefore, we expect the scientific community to adopt FA crosslinking in future interactome capture experiments to tracking changes in the RNA interactome networks in response to physiological cues. Of note, our study shows that FA captures the majority of the UV-RBPome (Fig. 2e).

We apologize that the description and definition of CAPRI in the revised manuscript was not clear to the reviewer. At the suggestion of the editor we have now added an additional figure (Fig. 1), which gives a detailed graphical summary of CAPRI and at the same time

outlines the other two techniques, namely UV-/FA-RBPome purification (Fig. 1a) and FA-based domain capture (Fig. 1c). The FA-domain capture provides information complementary to CAPRI through its use of formaldehyde.

We had originally performed FA-based RNA binding domain capture (Fig. 1c) to address Reviewer #2's concerns about the potential utility of FA in mapping RNA-protein interaction domains. Although not included as part of the current CAPRI protocol, we can clearly envision it to complement the information provided by CAPRI. We expect it to become an integral part of CAPRI 2.0 in the future.

2.3 The authors claim that they sufficiently validated their method and verified novel protein-RNA interactions. However, the validation of the technique is rather poorly addressed. For instance, the structures shown in Figure 3d 3e are not clearly visible and only one protein is used for validation. The authors could have used the multitude of highly resolved structures of spliceosomal subcomplexes (these are: spliceosomal B, C, tri-snRNP, Bact, step I complexes and others) to compare their data (both, xl-peptides and ADJ peptides) with the structures of the proteins in these complexes.

It appears that the quality of the Fig. 3d and e (now Fig. 4d and e) may have been compromised upon compression of the images during submission. We have now increased the size of both panels in the manuscript.

However, we are surprised by the request concerning the inclusion of more 3D structures for validation. In addition to Fig. 3d and e (now Fig. 4d and e), we had also presented 7 additional structures in Supplementary Table 6, which seem to have been overlooked by the reviewer. We would therefore kindly refer the reviewer to Supplementary Table 6. Furthermore, we now also show mapping of ADJ-peptides on the structure of the human spliceosomal complex member SF3B1 in Fig. 5e.

2.4 I further do not agree with the verification of the newly identified RBPs. For instance, the authors identified novel RBPs in the respiratory chain complexes. Although these proteins are discussed more in detail than others, it appears that the authors do not have detailed knowledge on the proteins and refuse to acquire some basic knowledge. First, there is no biological meaning why the proteins discussed here should bind.

We would like to argue that uncovering the biological relevance and function of the multitude of OXPHOS protein-RNA interactions is truly beyond the scope of the manuscript. Excitingly, our screen scored 34 OXPHOS proteins as RNA-binding proteins, suggesting a hitherto unknown functional role of RNA in OXPHOS biology. Of the 34 OXPHOS proteins identified in at least one of our Drosophila RBPomes, 26 proteins are also shown to bind RNA in one or more previously/recently published studies (Revision 2 Fig. 1). These interesting findings will stimulate future research in the metabolism community and deserve a more focussed separate study.

Revision 2 Fig. 1 for Reviewer2: OXPHOS proteins are recovered in our and multiple published RBPomes. The presence of *Drosophila* proteins is indicated in dark purple in the respective RBPomes. The datasets used are (from left to right): our FA-, UV- and ADJ-RBPomes from *Drosophila*, *Drosophila* orthologs of RBPs from Hentze et al 2018⁵, *Drosophila* orthologs of the human RBPome from Mullari and colleagues², *Drosophila* embryo UV RBPomes^{6,7} combined, *Drosophila* orthologs of RBPs from the recently published XRNAX⁸ and OOPS⁹ methods.

2.5 They further discuss binding of RNA to the gamma subunit of the ATP synthase (see Supplementary Figure 12), however, their validation reveals binding of the alpha subunit. (Please note, that the Western Blot shown in Supplementary Figure 13e is of poor quality. According to the information provided in the methods section, there is no ATP synthase-specific antibody which used.)

In addition, they wrongly state in their manuscript that subunit alpha of the F1 ATP synthase is a member of complex I of the OXPHOS complexes. In fact, ATP synthase is complex V. (Minor point: the authors also discuss cytochrome oxidase – the correct name is cytochrome c oxidase (COX).) Comparing these results with the data presented in Supplementary Figure 12 it becomes apparent that all these interactions are revealed by FA-crosslinking. As discussed below (and in my previous report), these might be false-positives and should carefully checked.

We admittedly did not independently verify the RNA binding of *Drosophila* ATP synthase gamma. However, we detected a UV-crosslinked peptide inside this protein (Supplementary Fig. 12b, annotated peptide in Supplementary table 7 spectrum 1), which is a gold standard method of endogenous RBP identification. We would like to emphasise that OXPHOS proteins were NOT exclusively identified by FA crosslinking (Revision 2 Fig. 1). We believe

that the sheer number of OXPHOS members identified in our interactomes speaks for the strength of these findings (newly modified Supplementary Fig. 12a).

Beyond this, we were able to validate RNA binding of two OXPHOS pathway proteins using an independent approach (Supplementary Fig. 13a). The protocol includes disruption of noncovalent RNA-protein interactions through the use of ~5M guanidine hydrochloride (Supplementary Fig. 13e). We validated Complex V member ATPase alpha (ATP5A1) and Complex II member Succinate dehydrogenase, subunit A (SDHA). Both the proteins were identified in UV- as well as FA-RBPomes. They have also been detected in interactome captures in other species (Revision 2 Fig. 1). We disagree with the judgement that the blots in Supplementary Fig. 13e are of “low quality”. We employed a widely used commercial cocktail of antibodies (OXPHOS Rodent WB Antibody Cocktail, Abcam ab110413) to detect Complex V ATP5A1, Complex III member UQCRC2, Complex IV subunit MTCO1, Complex II-SDHB and Complex I subunit NDUF88 simultaneously. The weak western blot intensity of the RNA binding proteins in the figure stems from the fact that the input amounts have been kept identical to those used in positive controls. The weaker signal intensity of the newly identified RNA-interacting metabolic enzymes is most likely related to substoichiometric RNA binding. The signal is comparable (even slightly better) to the positive control GAPDH, suggesting moonlighting RNA-binding functions similar to the ones described for the GAPDH enzyme¹⁰.

We thank the reviewer for spotting the typographical errors in the labelling of Complex V and Cytochrome c oxidase in the text. We have corrected them in the present version.

2.6 The authors further claim that their technique outperforms previously published techniques in that it does not require manual spectra annotation. However, on page 10 of their revised manuscript they state that “the spectra were manually curated to produce a final set of crosslinked peptides...”. It also wrongly stated that previously published studies require manual peptide sequence analysis. In the here cited studies manual inspection of the mass spectra was employed for validation of the data analysis workflow.

There is some misunderstanding and we regret if our wording was not precise enough. Therefore, we have rephrased the text in the revised manuscript

“ Since the number of PTMs needed to be defined for various combinations of 3 nucleotides is in the hundreds, only database search based strategies have been successful to analyse crosslinked peptides in a high throughput¹¹⁻¹³ manner. These searches also assume that a minimum of one nucleotide is left behind on the crosslinked peptide after RNase treatment. A high throughput de novo analysis could also be used along with a combination of database search. ”

In addition, we have performed manual curation of spectra in order to reduce the chance of false positive spectra mapping and identifying novel RNA binding domains. For example in human sample we manually removed 61 spectra (6.1%) of all 997 spectra. This kind of manual curation has also been recommended in the tutorial in the supplemental information of the original 2014 RNPxl paper¹². Also, we have not claimed that the crosslinked peptide

analysis performed by PEAKS DeNovo/DB+PTM “outperforms” RNPxl or MSFragger. We would just like to make the point that our study is the first one that harnesses the power of de novo sequence tag assignment for the identification of RNA crosslinked peptides. We simply tried to state that the internal fragment ion annotation is automated in PEAKS Studio 8.5. Automated annotation was not a part of the original 2014 RNPxl paper¹². However, in a more recent publication Veit et al. 2016¹¹ the authors claim to provide automated annotation in form of a plug-in node for the Thermo Fisher Proteome Discoverer software versions 2.0 and 2.1. Veit et al. 2016 additionally mention to have a manuscript in pipeline that is going to describe the RNPxl Proteome Discoverer “community node” and also explain the heuristic based annotation of internal spectral fragments.

Since we have neither utilized MSFragger¹³ nor compared it to PEAKS studio we cannot really comment on its level of automation in respect to MS2 spectra annotation (e.g. RNA marker ions, RNA PTM harboring y- or b-ions etc.). This is also because both the first publication of this software¹³ and its application in context of the XRNAX protocol⁸ do not provide any annotated MS2 spectra information. We have also already stated that the comparison of PEAKSxl with other tools is beyond the scope of this manuscript.

2.7 The authors further seem to ignore the facts that FA-crosslinks are reversible. Even though crosslinking of the sample could be proven by SDS-PAGE the covalent linkages are most likely disrupted during the ionization process (the authors claim that the transfer capillary is heated to 270°C). Previous studies using FA-crosslinking mostly used Q-ToF mass spectrometers (or similar) which do not employ a heated transfer capillary (the instrument set-up is different which does not require these temperatures in the ion source). In addition, FA-crosslinking induces protein-crosslinking (as mentioned by the authors in their revised manuscript) which might be the reason for the higher number of identified RNA-binding proteins, i.e. using FA-crosslinking will rather pull down an entire interaction network than the RNA-binding interactome. In this respect the authors cannot conclude at all whether the cross-link has derived from RNA or protein-protein interaction.

We would like to kindly reiterate that we have made NO claim about finding any FA-crosslinked heteroconjugate peptides during the MS/MS analysis. All proteins and peptides in the FA-based experiments are de-crosslinked before they are analysed by mass spectrometry.

Firstly, we would like to point out that a substantial number of “FA-specific proteins” have been reported in recent reports exclusively employing UV-based interactome captures (mostly in other species) (Fig. 2 d, Revision 1 Fig. 2 for Reviewer2).

We totally agree with the reviewer that FA may capture indirect protein-mediated interactions. We actually claim these indirect interactions to be of added value to identify proteins involved in post transcriptional gene regulation (as detailed in response to 2.1). Even if one assumes that limited FA crosslinking (0.1%) could stabilize and therefore retrieve entire (or partial) cellular protein networks, at least one of these proteins must have been in direct contact with RNA. This would imply that network-wide protein interactions are taking place when at least one of the network members is bound to RNA. Such interactions are biologically relevant.

At the risk of repeating this statement *ad infinitum*, the huge overlap in our data between the UV- and FA-RBPome and the overlap of FA-RBPome specific proteins with other published interactomes makes it very unlikely that the FA-RBPome mainly delivers false positive RNA-interacting proteins. Additionally, CAPRI will represent the right choice for scrutinizing RNA-protein interactions taking place at the closest interface between RNA and amino acid side chains.

2.8 The over-interpretation of the results and the lack of knowledge of the current literature becomes obvious through the presented network of the spliceosomal complex. I doubt that all listed proteins directly bind RNA. There are several proteins that have never been reported in the context of the spliceosome (e.g. Rab19) and are known to function in very different cellular processes (e.g. membrane trafficking). The color coding and, as a consequence, the over-representation of proteins identified to be involved in RNA interaction is misleading since only a small number of proteins have been identified in the XL- and ADJ-proteomes (which likely represent direct RNA-binding proteins). Other proteins listed are simply identified in the pull-down experiments without any further specification (i.e. XL-site of adjacent site).

We do not claim that all the proteins shown in Fig. 4d (now Fig. 5d) directly bind RNA. The image actually depicts the confidence with which a protein can be considered to directly bind RNA.

We have used the COMPLEAT database¹⁴ to annotate the spliceosomal complex members. The COMPLEAT annotation retrieves information on human (multi)protein complexes from the CORUM database. Fig. 4d (now Fig. 5d) represents the COMPLEAT complex number FC2768, which corresponds to the spliceosome C complex (<http://www.flyrnai.org/compleat/Details?requestType=browse&org=Fly&id=FC2768&search=FC2768>). COMPLEAT database has converted the CORUM human complex (ID 1181) information to Fly orthologs using DIOPT database (DRSC Integrative Ortholog Prediction Tool).

The Drosophila protein Rab19 was included in the spliceosome because of a misannotation of the Drosophila Rab19 gene as an ortholog of the human ISY1 gene (an RNA-binding splicing factor) in the DIOPT database (<http://flybase.org/reports/FBgn0015793;Tab:HumanOrthologs> (via DIOPT v7.1)). The Drosophila Rab19 protein sequence shares the highest sequence similarity with another human protein called RAB43. The RAB43 gene is also transcribed in the form of an “alternative” gene called ISY1-RAB43, which represents a run-through transcript between the two neighbouring genes ISY1 and RAB43 (Revision 2 Fig. 2 for reviewer). The ISY1 gene is indeed a RNA-binding protein involved in splicing, whereas Drosophila Rab19 and human RAB43 (the most probable ortholog of Rab19) are not. We suspect that the Rab19 gene was wrongly included in the spliceosome complex due to this error caused by the ISY1-RAB43 run-through transcript. We are very thankful to the reviewer for noticing this error and we have corrected this mistake in the spliceosomal complex in Fig. 4d (now Fig. 5d). We have also removed a pseudogene, CR31054 (<http://flybase.org/reports/FBgn0051054>), from the complex in Fig. 5d.

Revision 2 Fig. 2 for Reviewer2: Snap-shot of the human ISY1 and RAB43 gene loci reproduced from Ensembl database (http://www.ensembl.org/Homo_sapiens/Gene/Summary?g=ENSG00000261796;r=3:129087575-129161036;t=ENST00000418265). The ISY1-RAB43 run-through transcript is highlighted in green and starts from ISY1 gene on the left proceeding into the RAB43 gene on the right.

2.9 Indeed, it seems that the authors are not aware of the current literature or simply ignore it. A very striking example of this fact is that they claim in the figure legend of Figure 4 that a novel surface interaction with RNA has been identified in the SF3b protein complex. The SF3b complex has been crystallized and the protein-protein and protein-RNA interaction sites (both based on cross-linking experiments) including the repeating “HEAT domains” (here termed “Armadillo-fold”) have been identified by chemical and UV-cross-linking, respectively and described in a very detailed manner in this work (Cretu et al., 2016). Moreover, cryo-EM structures of these proteins within spliceosomes are available showing the interaction of these domain with RNAs.

Regrettably, this seems to be a matter of misunderstanding here. The novel interface which we identify in our work is the one marked by the crosslinked peptide (XL-peptide) at the extreme N-terminus of Drosophila Sf3b1 (Supplementary table 7 spectrum 26). As the reviewer states, the Armadillo-fold/HEAT-domain-based RNA interaction surface in the C-terminal portion of the protein had previously been identified by Cretu et al (2016) and we had already cited this paper in the text. We have modified the text to be more explicit. We thank the reviewer for their suggestion of mapping the ADJ-peptides onto SF3B1 protein to show their proximity to RNA. For clarification, we now not only show the structure in Fig. 5d-e, but also a comparison of our CAPRI analysis with previous data (from RBDmap¹⁵ and pCLAP²) reported for human SF3B1.

2.10 In addition, the authors claim in their manuscript and in the point-by-point response that they identified “new types of protein-RNA cross-links” by MS. This is again wrongly stated. All of the described “new types” of cross-links which have been observed in the MS2 spectra have already been described and documented in several previously published studies that focus on the MS-identification of UV-based protein-RNA cross-linking sites (together with their extensive spectra annotation alongside a detailed repository of supplementary data with).

We would like to reiterate that this new type of crosslink is observed as an independent peptide ion species in full-scan MS1 spectra. We added cartoons to show a canonical crosslink with a single nucleotide (U) covalently attached to peptide in Supplementary figure 7g and a ‘novel’ crosslink with just a (nucleo)base (uracil=U’) in Supplementary Fig. 7h. They are detected independent of each other, which means that they differ in LC retention time and are hence present in different MS1 spectra. The later (Supplementary Fig. 7h) might originate from an in-source dissociation (at the glycosidic bond) of the canonical crosslinked peptide harboring an intact nucleotide monophosphate (Supplementary Fig. 7g). None of the existing tools including RNPxl or MSFragger would currently consider these “base-only” MS1 (parent) spectra (precluding the annotation of the corresponding MS2 daughter spectra) since they assume the presence of classical crosslinks (Supplementary Fig. 7g). The above mentioned figures and two examples of novel crosslinked peptides are annotated in the Supplementary Table 5 spectra 14 and 15. Both are reproduced below for convenience. Again, we do not claim here that RNPxl or MSFragger are technically not able to identify these novel “base only” crosslinks but the software pipelines need to be “instructed” to search for them.

Snap shot of Supplementary Figure 7:

g. Illustration of RNA PTMs on a hypothetical peptide with the sequence PEPTIDE in order to depict a canonical single nucleotide crosslink of uridine 5'-monophosphate (uridine nucleotide U = U'+rest).

h. as in g for a novel crosslink with uracil base only: U'

XL-peptide with base alone

14)

Name	Sequence	RNA PTMs	M/Z	z	Position of XL
FUBP2	C(+112.03)GLVIGR	U'	415.2213	2	Via base to C

#	Immonium	b	b (2+)	Seq	y	y (2+)	#
1	188.05	216.04	108.52	C(+112.03)			7
2	30.03	273.06	137.03	G	614.40	307.70	6
3	86.10	386.15	193.57	L	557.38	279.19	5
4	72.08	485.22	243.11	V	444.29	222.65	4
5	86.10	598.30	299.65	I	345.22	173.13	3
6	30.03	655.32	328.16	R	232.14	116.57	2
7	129.11			R	175.12	88.06	1

XL-peptide with base alone

15)

Name	Sequence	RNA PTMs	M/Z	z	Position of XL
SRSF2	S(+42.01)YGRP(+151.05)PPDVEGMTSLK	G'	642.9759	3	Via base to P

#	Immonium	b	b (2+)	Seq	y	y (2+)	#
1	102.06	130.05	65.53	S(+42.01)			16
2	136.08	293.11	147.06	Y	1797.86	899.42	15
3	30.03	350.13	175.57	G	1634.81	817.90	14
4	129.10	506.23	253.62	R	1577.79	789.39	13
5	221.11	754.34	377.67	P(+151.05)	1421.68	711.34	12
6	70.07	851.39	426.20	P	1173.58	587.29	11
7	70.07	948.44	474.72	P	1076.53	538.76	10
8	88.04	1063.46	532.24	D	979.47	490.24	9
9	72.08	1162.54	581.77	V	864.45	432.72	8
10	102.05	1291.58	646.29	E	765.38	383.19	7
11	30.03	1348.60	674.80	G	636.34	318.67	6
12	104.05	1479.64	740.32	M	579.32	290.16	5
13	74.06	1580.68	790.85	T	448.28	224.64	4
14	60.04	1667.72	834.36	S	347.23	174.11	3
15	86.10	1780.81	890.90	L	260.20	130.60	2
16	101.11			K	147.11	74.06	1

Snap-shot taken from Supplementary Table 5: Spectra 14 and 15. From top to bottom: top: Table with details of the XL-peptides, middle: Spectral annotation by PEAKS and bottom: Ion match table where ions detected in the spectrum are coloured in blue (b series) and red (y series)

Reviewer #3

I thank the authors for their comprehensive response. Based on the response and the revision I recommend the manuscript for publication.

We thank the reviewer for his/her support and were pleased to read that the reviewer found our response comprehensive.

References

1. Shchepachev, V., Bresson, S., Spanos, C. & Petfalski, E. Defining the RNA Interactome by Total RNA-Associated Protein Purification. *bioRxiv* (2018).
2. Mullari, M., Lyon, D., Jensen, L. J. & Nielsen, M. L. Specifying RNA-Binding Regions in Proteins by Peptide Cross-Linking and Affinity Purification. *J. Proteome Res.* **16**, 2762–2772 (2017).
3. Anderson, P. & Kedersha, N. Stress granules: the Tao of RNA triage. *Trends Biochem. Sci.* **33**, 141–150 (2008).
4. Wolozin, B. Regulated protein aggregation: stress granules and neurodegeneration. *Mol. Neurodegener.* **7**, 56 (2012).
5. Hentze, M. W., Castello, A., Schwarzl, T. & Preiss, T. A brave new world of RNA-binding proteins. *Nat. Rev. Mol. Cell Biol.* **19**, 327–341 (2018).
6. Sysoev, V. O. *et al.* Global changes of the RNA-bound proteome during the maternal-to-zygotic transition in *Drosophila*. *Nat. Commun.* **7**, 12128 (2016).
7. Wessels, H.-H. *et al.* The mRNA-bound proteome of the early fly embryo. *Genome Res.* **26**, 1000–1009 (2016).
8. Trendel, J. *et al.* The Human RNA-Binding Proteome and Its Dynamics during Translational Arrest. *Cell* **176**, 391–403 (2018).
9. Queiroz, R. M. L. *et al.* Comprehensive identification of RNA-protein interactions in any organism using orthogonal organic phase separation (OOPS). *Nat. Biotechnol.* **37**, 169–178 (2019).
10. Garcin, E. D. GAPDH as a model non-canonical AU-rich RNA binding protein. *Semin. Cell Dev. Biol.* **86**, 162–173 (2019).
11. Veit, J. *et al.* LFQProfiler and RNPxl: Open-Source Tools for Label-Free Quantification and Protein–RNA Cross-Linking Integrated into Proteome Discoverer. *J. Proteome Res.* **15**, 3441–3448 (2016).
12. Kramer, K. *et al.* Photo-cross-linking and high-resolution mass spectrometry for assignment of RNA-binding sites in RNA-binding proteins. *Nat. Methods* **11**, 1064–1070 (2014).
13. Kong, A. T., Leprevost, F. V., Avtonomov, D. M., Mellacheruvu, D. & Nesvizhskii, A. I. MSFragger: ultrafast and comprehensive peptide identification in mass spectrometry-based proteomics. *Nat. Methods* **14**, 513–520 (2017).
14. Vinayagam, A. *et al.* Protein Complex-Based Analysis Framework for High-Throughput

Data Sets. *Science Signaling* **6**, rs5–rs5 (2013).

15. Castello, A. *et al.* Comprehensive Identification of RNA-Binding Domains in Human Cells. *Mol. Cell* **63**, 696–710 (2016).